# Message Tuning Outshines Graph Prompt Tuning:
# A Prismatic Space Perspective

**Yancheng Chen** [1 2]  **Dun Ma** [1 2]  **Shuai Zhang** [1]  **Yang Liu** [1]  **Xixun Lin** [3]  **Xiangyu Zhao** [4]  **Wenguo Yang** [5]
**Wei Chen** [6]  **Chuan Zhou** [1 7]

## Abstract

Graph Foundation Models (GFMs), built upon the *Pre-training and Adaptation* paradigm, have emerged as a research hotspot in graph learning. For GNN-based GFMs, graph prompt tuning has become the prevailing adaptation method for downstream tasks. Although recent methods explain why graph prompt tuning works, how to rigorously measure its adaptation capacity remains an open problem. Addressing this problem is critical for understanding the capability limits of graph prompt tuning and for developing more powerful adaptation methods. In this paper, we propose **P**rismatic **S**pace Theory (PS-Theory), a novel mathematical framework to quantify the capacity of adaptation methods, while focusing on establishing the upper bound for the adaptation capacity of graph prompt tuning. Building upon the proposed PS-Theory, we further introduce **M**essage **T**uning for **G**FMs (MTG), a lightweight approach that injects a small set of learnable message prototypes into each layer of the GNN backbone to adaptively guide message fusion without updating pre-trained weights. Through our PS-Theory, we prove that the adaptation capacity of MTG can exceed the theoretical upper bound of graph prompt tuning. Extensive experiments demonstrate that MTG consistently outperforms graph prompt baselines across diverse benchmark datasets, providing strong empirical support for our theoretical findings.

## 1. Introduction

Graph Foundation Models (GFMs) (Liu et al., 2025; Wang et al., 2025c), built upon the *Pre-training and Adaptation* paradigm, aim to leverage the large-scale pre-training of broad graph data to support effective adaptation across a wide range of downstream graph tasks. As an important category of GFMs, GNN-based GFMs represent a promising direction by leveraging self-supervised pre-training to acquire transferable knowledge through Graph Neural Network (GNN) backbone architectures (Wang et al., 2024; Chen et al., 2025), including Message Passing Neural Networks (MPNNs) (Gilmer et al., 2017) and Graph Transformers (GTs) (Ying et al., 2021). For pre-trained GNN-based GFMs, fine-tuning (Hu et al., 2020b; Qiu et al., 2020; Rong et al., 2020) is the most intuitive and widely adopted method for downstream task adaptation. However, as illustrated in Figure 1(a), fine-tuning[1] involves updating all model parameters, requiring a full model copy per task while demanding substantial computational resources. Furthermore, the pretext-downstream graph task gap poses a significant challenge for fine-tuning, potentially causing negative transfer in few-shot scenarios (Zhang et al., 2022; Lin et al., 2024a).

To reduce trainable parameters and mitigate negative transfer, graph prompt tuning (Fang et al., 2023; Sun et al., 2023a) has emerged as an efficient alternative to fine-tuning. It typically keeps the pre-trained model's parameters frozen and enhances the adaptability of GNN-based GFMs through input-space adaptations, such as inserting lightweight learnable tokens or subgraphs, as illustrated in Figure 1(b). This reformulates downstream tasks to resemble pre-training tasks, thereby bridging the gap between pretext and downstream graph tasks to facilitate better knowledge transfer. Recent advances in graph prompt have shown promising performance in various graph-related applications (Diao et al., 2023; Yang et al., 2023; Niu et al., 2024; Yu et al., 2025).

Owing to these empirical successes, several studies (Fang et al., 2023; Sun et al., 2023a; Wang et al., 2025a) began to analyze why graph prompt tuning works from a data op-

---

[1]Academy of Mathematics and Systems Science, Chinese Academy of Sciences [2]School of Advanced Interdisciplinary Sciences, University of Chinese Academy of Sciences [3]Institute of Information Engineering, Chinese Academy of Sciences [4]City University of Hong Kong [5]School of Mathematical Sciences, University of Chinese Academy of Sciences [6]Institute of Computing Technology, Chinese Academy of Sciences [7]School of Cyber Security, University of Chinese Academy of Sciences. Correspondence to: Chuan Zhou <zhouchuan@amss.ac.cn>.

*Proceedings of the 43$^{rd}$ International Conference on Machine Learning*, Seoul, South Korea. PMLR 306, 2026. Copyright 2026 by the author(s).

---

[1]Within the scope of this paper, fine-tuning for GNN-based GFMs denotes full-parameter fine-tuning.

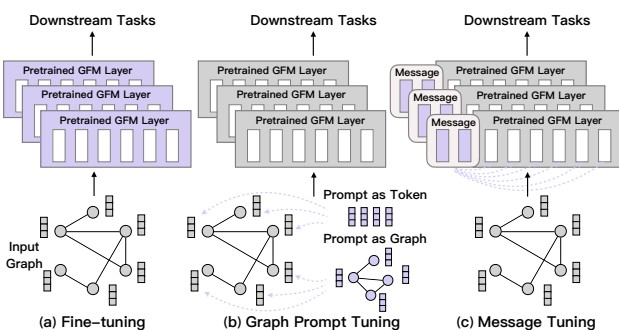

*Figure 1.* Fine-tuning (a) updates all GFM parameters (purple GFM Layer boxes), while Graph Prompt Tuning (b) typically updates prompt tokens or the prompt graph (purple prompt vectors) to transform the input graph, keeping GFM parameters frozen. We propose Message Tuning (c), which also freezes GFM parameters but optimizes the messages in each GFM Layer (purple message blocks) to regulate message fusion. The purple dashed lines indicate the learnable inserting patterns of the additional parameters.

eration perspective. However, how to rigorously measure its adaptation capacity on a specific GFM remains an open problem. Addressing this fundamental problem is crucial for precisely characterizing the capability bound of graph prompt tuning and for theoretically guiding the design of more powerful adaptation methods. To this end, we propose **P**rismatic **S**pace Theory (PS-Theory), providing a rigorous mathematical framework to quantify the capacity of adaptation approaches. Specifically, by analyzing the backbone architectures of mainstream GNN-based GFMs, we model each layer of GFMs as a piecewise linear mapping and leverage ideas from geometric measure theory to quantify the *refractive* power of each layer map. We then leverage PS-Theory to establish an upper bound for the adaptation capacity of graph prompt tuning, revealing the inherent limitations of methods operating solely at the input data level.

Building upon PS-Theory, we introduce **M**essage **T**uning for **G**FMs (MTG), a novel adaptation approach that injects learnable message prototypes into each layer and dynamically fuses them with the model's native messages, which is compatible with GFMs using either MPNN or GT backbones, as illustrated in Figure 1(c). Through our PS-Theory, we prove that the adaptation capacity of MTG can exceed the theoretical upper bound of graph prompt tuning, demonstrating its enhanced adaptation capability. To comprehensively evaluate MTG's practical performance, we benchmark it against the definitive Graph Prompt Learning benchmark ProG[2] (Zi et al., 2024). Extensive experiments demonstrate MTG's consistent superiority over state-of-the-art graph prompt baselines across diverse few-shot downstream tasks, which validates our theoretical claims regarding its enhanced adaptation capability. The robustness and efficiency

---
[2]ProG encompasses a variety of graph prompt methods, including graph prompt tuning. See Appendix A for the taxonomy of graph prompt methods.

of MTG are also confirmed by experiments involving sensitivity analysis and computational efficiency.

The contributions of this paper are summarized as follows:

- **Theoretical Foundation.** We propose PS-Theory, providing a rigorous mathematical framework to quantify adaptation capacity and establish the upper bound for the adaptation capacity of graph prompt tuning.

- **Adaptation Method.** We introduce MTG[3], a novel lightweight approach that dynamically guides message fusion by injecting learnable message prototypes across all layers without updating pre-trained weights, significantly enhancing adaptation capability. Through PS-Theory, we prove that MTG's adaptation capacity can exceed the upper bound of graph prompt tuning.

- **Extensive Experiments.** Through comprehensive evaluations across diverse few-shot downstream tasks, we demonstrate MTG's consistent superiority over state-of-the-art graph prompt baselines, validating our theoretical claims on its enhanced adaptation capability.

## 2. Problem Settings

In this section, we provide a mathematical formalization of graph prompt tuning, aiming to offer an intuitive perspective for theoretical analysis. Following Wang et al. (2025a), we assume all tasks are at the graph level, because for node-level and edge-level tasks, many studies (Sun et al., 2023a; Liu et al., 2023b) have proven that we can always find solutions to translate these tasks into graph-level tasks. Let $f_{\text{GFM}}$ denote a pre-trained GNN-based GFM with frozen parameters, and let $g_\theta$ denote a graph prompt function with parameters $\theta$ that transforms the input graph $\mathcal{G}$ into a prompted graph $g_\theta(\mathcal{G})$. Given a downstream dataset $\mathbb{G} = \{\mathcal{G}_i\}_{i=1}^n$, the goal of graph prompt tuning is to optimize $\theta$ to maximize the likelihood of the optimal representation $v_{\mathcal{G}_i}$ for a graph $\mathcal{G}_i$ from $\mathbb{G}$. This objective can be formulated as:

$$\max_\theta P_{f_{\text{GFM}}}(v_{\mathcal{G}_i}|g_\theta(\mathcal{G}_i)) \tag{1}$$

The theory in Wang et al. (2025a) rests on the assumption that a GNN model acts as a surjective mapping operator from the prompted graph space $g_\theta(\mathbb{G})$ to $\mathbb{R}^F$, where $F$ is the dimensionality of the representation. However, since real-world graph data is inherently bounded, the model's output is confined to a subset of $\mathbb{R}^F$. We posit that a principled way to measure the adaptation capacity of graph prompt tuning is to analyze the geometric properties of this constrained output space. To this end, we focus on characterizing the range of the function $f_{\text{GFM}}$ over the prompted graph space, i.e., $f_{\text{GFM}}(g_\theta(\mathbb{G})) = \{f_{\text{GFM}}(g_\theta(\mathcal{G}_i)) \mid \mathcal{G}_i \in \mathbb{G}\}$, which involves addressing the following three research questions:

---
[3]The code is available at https://github.com/CYCUCAS/MTG.

- **RQ1**: How to model the function $f_{\text{GFM}}$? (Section 3.1)
- **RQ2**: How to quantify the geometric transformation of the function $f_{\text{GFM}}$ on a given input space? (Section 3.2)
- **RQ3**: How to model the prompted graph space $g_\theta(\mathbb{G})$ and measure the output space $f_{\text{GFM}}(g_\theta(\mathbb{G}))$? (Section 3.3)

## 3. Prismatic Space Theory

In this section, we introduce Prismatic Space Theory (PS-Theory), providing a novel perspective and rigorous mathematical framework to quantify the capacity of adaptation approaches for GFMs and establish the upper bound for the adaptation capacity of graph prompt tuning. Given the mathematical complexity, we refer the reader to the **Reading Guideline** in Appendix B before proceeding.

### 3.1. A Unified Formulation for GNN-based GFMs

To facilitate theoretical analysis, we begin by introducing a unified formal framework that generalizes both MPNNs and GTs architectures. Consider a graph $\mathcal{G} = (\mathcal{V}, \mathcal{E})$ with $N = |\mathcal{V}|$ nodes. Let $\boldsymbol{X} \in \mathbb{R}^{N \times d_0}$ denote the node feature matrix and $\boldsymbol{A} \in \{0, 1\}^{N \times N}$ the adjacency matrix. Then, the $\ell$-th layer of a general GNN-based GFM is defined by the following formulation.

**Definition 3.1** (Unified GFM Layer). For any layer $\ell \in \{1, \ldots, L\}$, the layer-wise transformation that maps the previous node representation matrix $\boldsymbol{H}^{(\ell-1)} \in \mathbb{R}^{N \times d_{\ell-1}}$ to the updated representation $\boldsymbol{H}^{(\ell)} \in \mathbb{R}^{N \times d_\ell}$ is defined by the composition of three core operators:

$$
\boldsymbol{H}^{(\ell)} = \mathfrak{U}^{(\ell)}\Bigg(\mathfrak{M}^{(\ell)}\bigg(\mathfrak{A}^{(\ell)}\Big(\boldsymbol{A}, \boldsymbol{H}^{(\ell-1)}; \boldsymbol{\Theta}_a^{(\ell)}\Big),
$$
$$
\boldsymbol{H}^{(\ell-1)}; \boldsymbol{\Theta}_m^{(\ell)}\bigg), \boldsymbol{H}^{(\ell-1)}; \boldsymbol{\Theta}_u^{(\ell)}\Bigg), \tag{2}
$$

where $\boldsymbol{H}^{(0)} = \boldsymbol{X}$, $\{\boldsymbol{\Theta}_a^{(\ell)}, \boldsymbol{\Theta}_m^{(\ell)}, \boldsymbol{\Theta}_u^{(\ell)}\}$ are the learnable parameters. $\mathfrak{A}^{(\ell)}$, $\mathfrak{M}^{(\ell)}$, and $\mathfrak{U}^{(\ell)}$ denote the attention, message fusion, and update operators respectively: $\mathfrak{A}^{(\ell)}$ computes attention weights (including both learnable dynamic attention and static structural attention), $\mathfrak{M}^{(\ell)}$ performs weighted aggregation of node messages using attention scores, and $\mathfrak{U}^{(\ell)}$ combines previous node representations with the fused messages to obtain the updated representation.

We next explain how the Unified GFM Layer in this definition corresponds to actual model backbones, using GCN (Kipf & Welling, 2017) as an example.

**GCN (Graph Convolutional Network)** employs a fixed, non-learnable attention mechanism based on the normalized adjacency matrix and a simple update function.

Attention Operator $\mathfrak{A}^{(\ell)}$ computes a static, structural attention weight for each edge $(i, j)$ based on the normalized adjacency matrix:

$$
\mathfrak{A}^{(\ell)}\left(\boldsymbol{A}, \boldsymbol{H}^{(\ell-1)}; \boldsymbol{\Theta}_a^{(\ell)}\right) = \tilde{\boldsymbol{D}}^{-\frac{1}{2}} \tilde{\boldsymbol{A}} \tilde{\boldsymbol{D}}^{-\frac{1}{2}}, \tag{3}
$$

where $\tilde{\boldsymbol{A}} = \boldsymbol{A} + \boldsymbol{I}_N$ is the adjacency matrix with self-loops and $\tilde{\boldsymbol{D}}$ is the corresponding degree matrix. Notably, no parameters are used in this operation ($\boldsymbol{\Theta}_a^{(\ell)} = \varnothing$).

Message Fusion Operator $\mathfrak{M}^{(\ell)}$ computes a weighted aggregation of the neighbors' features using the normalized adjacency matrix:

$$
\mathfrak{M}^{(\ell)}\left(\tilde{\boldsymbol{D}}^{-\frac{1}{2}} \tilde{\boldsymbol{A}} \tilde{\boldsymbol{D}}^{-\frac{1}{2}}, \boldsymbol{H}^{(\ell-1)}; \boldsymbol{\Theta}_m^{(\ell)}\right)
$$
$$
= \tilde{\boldsymbol{D}}^{-\frac{1}{2}} \tilde{\boldsymbol{A}} \tilde{\boldsymbol{D}}^{-\frac{1}{2}} \boldsymbol{H}^{(\ell-1)} \boldsymbol{W}^{(\ell)}, \tag{4}
$$

where $\boldsymbol{W}^{(\ell)} \in \mathbb{R}^{d_{\ell-1} \times d_\ell}$ is a learnable weight matrix ($\boldsymbol{\Theta}_m^{(\ell)} = \boldsymbol{W}^{(\ell)}$).

Update Operator $\mathfrak{U}^{(\ell)}$ applies a non-linear activation function $\sigma$ (e.g., ReLU) to the aggregated messages:

$$
\mathfrak{U}^{(\ell)}\left(\tilde{\boldsymbol{D}}^{-\frac{1}{2}} \tilde{\boldsymbol{A}} \tilde{\boldsymbol{D}}^{-\frac{1}{2}} \boldsymbol{H}^{(\ell-1)} \boldsymbol{W}^{(\ell)}, \boldsymbol{H}^{(\ell-1)}; \boldsymbol{\Theta}_u^{(\ell)}\right)
$$
$$
= \sigma\left(\tilde{\boldsymbol{D}}^{-\frac{1}{2}} \tilde{\boldsymbol{A}} \tilde{\boldsymbol{D}}^{-\frac{1}{2}} \boldsymbol{H}^{(\ell-1)} \boldsymbol{W}^{(\ell)}\right), \tag{5}
$$

where the previous representation $\boldsymbol{H}^{(\ell-1)}$ is not explicitly used in the update, making the update a direct transformation of the messages ($\boldsymbol{\Theta}_u^{(\ell)} = \varnothing$).

The resulting layer formulation is:

$$
\boldsymbol{H}^{(\ell)} = \sigma\left(\tilde{\boldsymbol{D}}^{-\frac{1}{2}} \tilde{\boldsymbol{A}} \tilde{\boldsymbol{D}}^{-\frac{1}{2}} \boldsymbol{H}^{(\ell-1)} \boldsymbol{W}^{(\ell)}\right). \tag{6}
$$

The formulation in Definition 3.1 serves as a unified formalization of the core structure rather than encompassing all architecture details. The detailed correspondence between this formulation and more classical backbone architectures is presented in Appendix B.2.

### 3.2. A Geometric Measure Theoretic Formulation

**Prism Metaphor**. Graph prompt tuning typically operates by injecting a low-dimensional prompt into the high-dimensional input space of a frozen GFM. To quantify its efficacy, we need to understand how the GFM's architecture transforms this input space. We posit that each layer of a GFM, particularly those employing piecewise linear activations like ReLU (Nair & Hinton, 2010) or LeakyReLU (Maas et al., 2013), acts not merely as a contraction but as a *prism*. The non-isometric, piecewise linear action of a *prism* refracts the input space, collapsing some dimensions into oblivion, and progressively folding the input manifold. We model the GFMs as a sequence of measurable maps that transform the input space into a sequence of increasingly

complex, lower-dimensional prismatic space. We quantify the *refractive* power of each layer by leveraging ideas from geometric measure theory, focusing on the singular values of the layer's Jacobian and their effect on the intrinsic dimension and measure of the data manifold.

Due to space constraints, we provide the detailed mathematical definitions and theoretical analysis in Appendix B. First, we adopt the unified GFM layer from Definition 3.1 and introduce the piecewise linear map, which is a key abstraction for understanding the model's mechanisms.

**Definition 3.2** (Piecewise Linear Function for Matrix Maps). A function $F : \mathbb{R}^{N \times d_{\text{in}}} \to \mathbb{R}^{N \times d_{\text{out}}}$ is said to be piecewise linear if there exists a finite collection of polyhedral regions $\{R_i\}_{i=1}^K$ in $\mathbb{R}^{N \times d_{\text{in}}}$ such that $\mathbb{R}^{N \times d_{\text{in}}} = \bigcup_{i=1}^K R_i$ and for each region $R_i$, the function $F$ is affine, i.e., there exists $\boldsymbol{A}_i \in \mathbb{R}^{Nd_{\text{out}} \times Nd_{\text{in}}}$ and $\boldsymbol{b}_i \in \mathbb{R}^{Nd_{\text{out}}}$ such that:

$$\text{vec}(F(\boldsymbol{H})) = \boldsymbol{A}_i \cdot \text{vec}(\boldsymbol{H}) + \boldsymbol{b}_i \quad \text{for all } \boldsymbol{H} \in R_i, \quad (7)$$

where vec is the vectorization operation: $\mathbb{R}^{\alpha \times \beta} \to \mathbb{R}^{\alpha\beta}$ (i.e. stacking the columns of a matrix into a vector). Equivalently, in matrix form, $F(\boldsymbol{H}) = \text{unvec}(\boldsymbol{A}_i \cdot \text{vec}(\boldsymbol{H}) + \boldsymbol{b}_i)$, where unvec: $\mathbb{R}^{\alpha\beta} \to \mathbb{R}^{\alpha \times \beta}$. (See Appendix B.3 for details.)

**Proposition 3.3.** *The attention, message fusion, and update operators $\mathfrak{A}^{(\ell)}, \mathfrak{M}^{(\ell)}, \mathfrak{U}^{(\ell)}$ are generally continuous, piecewise linear functions and differentiable almost everywhere.*

**Proposition 3.4.** *The unified GFM layer map $F^{(\ell)} : \mathbb{H}^{(\ell-1)}(\subset \mathbb{R}^{N \times d_{\ell-1}}) \to \mathbb{H}^{(\ell)}(\subset \mathbb{R}^{N \times d_{\ell}})$ is a piecewise linear function. For any point $\boldsymbol{H}$ where $F^{(\ell)}$ is differentiable, its Jacobian $\boldsymbol{J}^{(\ell)}(\boldsymbol{H}) \in \mathbb{R}^{Nd_{\ell} \times Nd_{\ell-1}}$ exists.*

The proofs of Propositions 3.3 and 3.4 are provided in Appendices B.4 and B.5, respectively. Having abstracted the model architecture, we adopt a geometric perspective to model the input data and output space, with particular emphasis on capturing the bounded nature of the input data.

**Definition 3.5** (Input Manifold and Representation Space). The input space is modeled as a compact, smooth data manifold $\mathcal{M}_0 \subset \mathcal{X} \subset \mathbb{R}^{N \times d_0}$, with intrinsic dimension $d_{\text{int}}(\mathcal{M}_0) = D_0$. $\mathcal{X}$ denotes the entire set of possible input data forms for the model. $\mathcal{M}_0$ represents a low-dimensional subset of $\mathcal{X}$ endowed with specific semantic and geometric structures (see Appendix B.6 for details). The representation at layer $\ell$ is the image of the input manifold under the composite map $\Phi^{(\ell)} = F^{(\ell)} \circ \cdots \circ F^{(1)}$:

$$\mathcal{M}^{(\ell)} = \Phi^{(\ell)}(\mathcal{M}_0) \subset \mathbb{R}^{N \times d_{\ell}}. \quad (8)$$

**Definition 3.6** (Prismatic Space). A set $\mathcal{M}$ is said to be prismatic space if there exists a compact, smooth manifold $\mathcal{N}$ and a piecewise linear map $f$ such that $\mathcal{M} = f(\mathcal{N})$.

This definition formalizes the preceding **Prism Metaphor**. Real-world graph data can be viewed as points sampled

from the input manifold. And the output representations of graph data lie in the prismatic space, defined as the image of the input manifold under the GFM's mapping. Hence, such a geometric modeling approach is of practical significance.

**Proposition 3.7.** $\Phi^{(\ell)} = F^{(\ell)} \circ \cdots \circ F^{(1)}$ *is piecewise linear. Assume that $\Phi^{(\ell)}$ is injective on each polyhedral region, then $\mathcal{M}^{(\ell)} = \Phi^{(\ell)}(\mathcal{M}_0)$ is a prismatic space and may have singularities.*

The proof of Proposition 3.7 is provided in Appendix B.7. As in many geometric theories, an intuitive strategy for analyzing complex geometric space is to begin with a local perspective, particularly since the formation process of prismatic space is already well understood. Consequently, we focus on the singular value decomposition (SVD) of the layer Jacobians, at the core of the prismatic effect.

**Definition 3.8** (Spectral Prism of a Layer Map). For a point $\boldsymbol{H} \in \mathbb{H}^{(\ell-1)}$ where $F^{(\ell)}$ is differentiable, let $\boldsymbol{J}^{(\ell)}(\boldsymbol{H}) = \boldsymbol{U}^{(\ell)}\boldsymbol{\Sigma}^{(\ell)}(\boldsymbol{V}^{(\ell)})^{\top}$ be its SVD (see Appendix B.8 for details). The diagonal matrix $\boldsymbol{\Sigma}^{(\ell)} = \text{diag}(\sigma_1^{(\ell)}, \sigma_2^{(\ell)}, ..., \sigma_{r_\ell}^{(\ell)}, 0, ..., 0)$ contains the singular values, where $r_\ell$ is the rank.

**Theorem 3.9** (Local Measure Contraction Factor). *Let $\mathbb{S} \subset \mathbb{H}^{(\ell-1)}$ be a sufficiently small measurable set contained in an $s$-dimensional subspace $\mathbb{V}$ on which $F^{(\ell)}$ is linear and injective, with constant Jacobian $\boldsymbol{J}^{(\ell)}$ of rank $r_\ell$ ($s \leq r_\ell$). Assume $\mathbb{V}$ is the subspace spanned by the first $s$ right singular vectors of $\boldsymbol{J}^{(\ell)}$, corresponding to the $s$ largest singular values $\sigma_1^{(\ell)} \geq \cdots \geq \sigma_s^{(\ell)} > 0$. Then, for the $s$-dimensional Hausdorff measure $\mathcal{H}^s$ (Krantz & Parks, 2008):*

$$\mathcal{H}^s(F^{(\ell)}(\mathbb{S})) = \Big( \prod_{i=1}^{s} \sigma_i^{(\ell)} \Big) \mathcal{H}^s(\mathbb{S}). \quad (9)$$

*If $s = r_\ell$, the volume contraction factor is $\prod_{i=1}^{r_\ell} \sigma_i^{(\ell)}$.*

**Corollary 3.10** (Local ReLU Prism Effect). *Consider the ReLU activation function used within $F^{(\ell)}$. At points where ReLU is differentiable, its Jacobian $\boldsymbol{J}_{ReLU}$ is a diagonal matrix with diagonal entries either 0 or 1, and hence idempotent ($\boldsymbol{J}_{ReLU}^2 = \boldsymbol{J}_{ReLU}$). This implies that ReLU acts as a local projection, nullifying some dimensions (setting outputs to zero) and preserving others. The ReLU component contributes to the prismatic effect by introducing sparsity and reducing the effective rank of the Jacobian in local regions.*

The proof of Theorem 3.9 is provided in Appendix B.9. Corollary 3.10 provides a detailed explanation of the ReLU activation function. Having characterized the local properties via the singular values of the layer Jacobians, we integrate this local view with the global perspective. This requires an abstract mathematical technique: constructing a global partition from local pieces is a common approach,

even foundational to calculus. By Proposition 3.7, the piecewise linearity of the GFM network implies that the input manifold $\mathcal{M}_0$ is partitioned into multiple linear regions.

**Definition 3.11** (Linear Region Partition). For each layer $\ell \in 1, \dots, L$, let $\Omega^{(\ell)}$ be the set of polytopic regions in $\mathbb{H}^{(\ell-1)}$ on which the function $F^{(\ell)}$ is linear. The GFM network $\Phi = F^{(L)} \circ \cdots \circ F^{(1)}$ defines a recursive partition of the input manifold $\mathcal{M}_0$ into cells $\{C_k\}$, where each cell $C_k$ is a connected subset of $\mathcal{M}_0$ such that there exists a sequence of regions $R_1 \in \Omega^{(1)}, R_2 \in \Omega^{(2)}, \dots, R_L \in \Omega^{(L)}$ satisfying the following sequential compatibility condition:

$$C_k \subseteq R_1, F^{(1)}(C_k) \subseteq R_2, F^{(2)}(F^{(1)}(C_k)) \subseteq R_3, \\ \dots, F^{(L-1)} \circ \cdots \circ F^{(1)}(C_k) \subseteq R_L, \quad (10)$$

and on each cell $C_k$, the full network map $\Phi$ is linear. The total number of $\{C_k\}$ is related to the specific architecture and parameters of the GFM network.

**Theorem 3.12** (Prismatic Folding and Intrinsic Dimension). *The global map $\Phi : \mathcal{M}_0 \to \mathcal{M}^{(L)}$ is piecewise linear. The intrinsic dimension of the final representation space is bounded by the maximum over linear regions of the minimum rank achieved across layers:*

$$d_{int}(\mathcal{M}^{(L)}) \leq \max_k \min_\ell rank(\boldsymbol{J}^{(\ell)}|_{\Phi^{(\ell-1)}(C_k)}). \quad (11)$$

*Furthermore, the map $\Phi$ is piecewise constant on its rank. The final output $\mathcal{M}^{(L)}$ is a prismatic space in $\mathbb{R}^{N \times d_L}$, likely with an intrinsic dimension much lower than that of $\mathcal{M}_0$.*

The proof of Theorem 3.12 in Appendix B.11 provides a method for analyzing the upper bound of the intrinsic dimension of the output prismatic space. This bound is analytical, derived from the partition of the input manifold induced by the GFM network as defined in Definition 3.11, making it difficult to compute numerically. Building on the local measure computation derived in Theorem 3.9, we formulate the definition of a global measure on the prismatic space in the following theorem.

**Theorem 3.13** (Measure of the Final Prismatic Space). *Assume the piecewise linear map $\Phi = F^{(L)} \circ \cdots \circ F^{(1)}$ is injective on the partition $C_k$ of the input manifold $\mathcal{M}_0$, where each $C_k$ is a cell in the linear region partition. Then, the $d_{int}$-dimensional Hausdorff measure of the final prismatic space $\mathcal{M}^{(L)} = \Phi(\mathcal{M}_0)$ is given by:*

$$\mathcal{H}^{d_{int}}(\mathcal{M}^{(L)}) = \sum_k \mathcal{H}^{d_{int}}(\Phi(C_k)) \\ = \sum_k \Big( \prod_{\ell=1}^{L} \prod_{i=1}^{d_{int}} \sigma_{i,k}^{(\ell)} \Big) \mathcal{H}^{d_{int}}(C_k), \quad (12)$$

*where for each layer $\ell$ and cell $C_k$, $\sigma_{i,k}^{(\ell)}$ for $i = 1, \dots, d_{int}$ are the $d_{int}$ largest singular values of the Jacobian $\boldsymbol{J}^{(\ell)}$ of*

$F^{(\ell)}$ *restricted to the tangent space of $\Phi^{(\ell-1)}(C_k)$ (which is $d_{int}$-dimensional). If $\Phi$ is not injective, the formula provides an upper bound.*

The proof of Theorem 3.13 is provided in Appendix B.12. This theorem precisely quantifies the prismatic effect: the total *volume* of the final representation is the sum of the volumes of all fragments of the input manifold, each shrunk by the product of the singular values of the Jacobians along its path through the network.

At this point, we have established a mathematical framework (PS-Theory) for analyzing the output prismatic space of GFM. However, corresponding theoretical results on adaptation capacity still require integration with specific adaptation methods, such as graph prompt tuning.

### 3.3. Adaptation Capacity of Graph Prompt Tuning

As demonstrated by Lemma 1 in Wang et al. (2025a), graph prompt tuning methods, such as GPF (Fang et al., 2023) and All-in-One (Sun et al., 2023a), are equivalent to a transformation of the node feature matrix $\boldsymbol{X}$. This transformation can be simplified to the form $\boldsymbol{X}_\omega = \tilde{\boldsymbol{X}} + \boldsymbol{c}\boldsymbol{p}^\top$, where $\boldsymbol{c} \geq \boldsymbol{0}$ is the coefficient vector, $\boldsymbol{p}$ is the prompt vector, and $\tilde{\boldsymbol{X}}$ can be either $\boldsymbol{X}$ or the natural extension of $\boldsymbol{X}$: $\begin{bmatrix} \boldsymbol{X} \\ \boldsymbol{0} \end{bmatrix}$. Leveraging the geometric interpretation of this additive formulation, we model graph prompt tuning as a perturbation on the input manifold and measure the adaptation capacity of graph prompt tuning by deriving the prompt efficacy bound.

**Definition 3.14** (Prompt Perturbation Manifold). Assume the original input manifold $\mathcal{M}_0$ is perturbed by a prompt $\boldsymbol{P}$, forming a compact smooth manifold $\mathcal{M}_0(\boldsymbol{P})$, e.g., $\mathcal{M}_0(\boldsymbol{P}) = \{\boldsymbol{X} + \boldsymbol{P} \mid \boldsymbol{X} \in \mathcal{M}_0\}$. The prompt space $\mathcal{P}$ defines a family of manifolds: $\{\mathcal{M}_0(\boldsymbol{P}) \mid \boldsymbol{P} \in \mathcal{P}\}$. (See Appendix B.13 for details.)

**Theorem 3.15** (The Prompt Efficacy Bound). *The adaptation capacity of a prompt $\boldsymbol{P}$ to influence model output is bounded by the measure and diameter of $\mathcal{M}^{(L)}(\boldsymbol{P})$:*

$$\mathcal{H}^{d_{int}}(\mathcal{M}^{(L)}(\boldsymbol{P})) \leq \Big( \sup_k \prod_{\ell=1}^{L} \prod_{i=1}^{d_{int}} \sigma_{i,k}^{(\ell)} \Big) \cdot \mathcal{H}^{d_{int}}(\mathcal{M}_0(\boldsymbol{P})), \quad (13)$$

$$diam(\mathcal{M}^{(L)}(\boldsymbol{P})) \leq \Big( \prod_{\ell=1}^{L} \sup_k |\boldsymbol{J}_k^{(\ell)}|_{op} \Big) \cdot diam(\mathcal{M}_0(\boldsymbol{P})), \quad (14)$$

*where $diam(\mathcal{M}) = \sup_{x,y \in \mathcal{M}} \|x - y\|$ denote the diameter of a set $\mathcal{M}$, $|\cdot|_{op}$ is the spectral norm (the largest singular value), $\sigma_{i,k}^{(\ell)}$ are the singular values of the Jacobian of the $\ell$-th layer in the $k$-th linear region, and $d_{int}$ is the intrinsic dimension of $\mathcal{M}^{(L)}(\boldsymbol{P})$.*

The proof of Theorem 3.15 is provided in Appendix B.15. This theorem reveals that graph prompt tuning is funda-

mentally constrained by the frozen GFM's backbone architecture. The prompt's influence is compressed by the product of layer-wise Jacobian singular values, leading to irreversible information loss. Since the prismatic, piecewise linear structure of the pre-trained model is immutable, the prompt can only shift the input within this fixed, contracting geometric framework. Due to space constraints, we provide a detailed analysis of Theorem 3.15 in Appendix B.16.

The establishment of Prismatic Space Theory revolves around graph prompt tuning, yet it offers a more fundamental geometric perspective on how graph foundation models process input manifolds. The theory is developed layer by layer, making it not limited to adaptation methods that operate solely at the input data level, but also applicable to the analysis of other types of adaptation approaches. We leave this as a direction for future research.

## 4. Message Tuning for GFMs

In this section, we first utilize the PS-Theory in Section 3 to derive design insights for improving graph prompt tuning. Then we introduce Message Tuning for GFMs (MTG), a novel lightweight adaptation approach that dynamically guides message fusion across all layers (Figure 1). Through the PS-Theory, we prove that MTG's adaptation capacity can exceed the upper bound of graph prompt tuning.

### 4.1. Design Insight

The analysis in Section 3.3 and Appendix B.16 implies that the influence of a prompt $\boldsymbol{P}$ on the output space is constrained by the compositional prismatic effect of the frozen GFM layers. An intuitive improvement is to apply prompts at each layer of the model, an idea that aligns with prefix-tuning (Li & Liang, 2021) widely used in language models. Under the framework of PS-Theory, applying prompt perturbation (Definition 3.14) at each layer can expand the range of the representation space (Definition 3.5). This effectively mitigates the prismatic effect caused by layer mappings that compress the space, thereby enhancing the expressive capacity introduced by the prompt. However, prefix-tuning is specifically designed for transformer architectures and generative tasks on sequential data, making it not directly applicable to graph-structured data. In this work, we introduce a general message tuning framework tailored for graph foundation models with diverse backbone architectures.

### 4.2. Core Mechanism

The core mechanism of MTG is to inject a small set of learnable message prototypes into each layer, which then undergo a dynamic fusion with the native messages computed by the model, while the original parameters $\boldsymbol{\Theta}^{(\ell)} = \{\boldsymbol{\Theta}_a^{(\ell)}, \boldsymbol{\Theta}_m^{(\ell)}, \boldsymbol{\Theta}_u^{(\ell)}\}$ in Eq.(2) are kept frozen.

**Learnable Message Prototypes.** Formally, for each layer $\ell$, we introduce a small set of $m$ learnable prototype vectors, denoted as $M^{(\ell)} = [m_1^{(\ell)}, m_2^{(\ell)}, \ldots, m_m^{(\ell)}]^\top \in \mathbb{R}^{m \times d_{\ell-1}}$. Then the GFM layer after injecting message prototypes can be expressed as:

$$
\begin{aligned}
\boldsymbol{H}^{(\ell)} = \mathfrak{U}^{(\ell)} \Bigg( \mathfrak{M}^{(\ell)} \Big( \mathfrak{A}^{(\ell)} \left( \boldsymbol{A}, \boldsymbol{H}_M^{(\ell-1)}; \boldsymbol{\Theta}_a^{(\ell)} \right), \\
\boldsymbol{H}_M^{(\ell-1)}; \boldsymbol{\Theta}_m^{(\ell)} \Big), \boldsymbol{H}_M^{(\ell-1)}; \boldsymbol{\Theta}_u^{(\ell)} \Bigg),
\end{aligned}
\tag{15}
$$

$$
\boldsymbol{H}_M^{(\ell-1)} = \mathfrak{F}^{(\ell)}(\boldsymbol{H}^{(\ell-1)}, M^{(\ell)}; \boldsymbol{\Theta}_f^{(\ell)}),
\tag{16}
$$

where $\mathfrak{F}^{(\ell)}$ denotes dynamic message fusion operator, $M^{(\ell)}$ and $\boldsymbol{\Theta}_f^{(\ell)}$ are the learnable parameters. This is equivalent to replacing $\boldsymbol{H}^{(\ell-1)}$ in Eq.(2) with $\boldsymbol{H}_M^{(\ell-1)}$ defined in Eq.(16), resulting in Eq.(15).

**Dynamic Message Fusion.** While both are message fusion operators, $\mathfrak{F}^{(\ell)}$ differs from $\mathfrak{M}^{(\ell)}$ in that it dynamically fuses learnable message prototypes with the input message representations at each layer, instead of fusing messages between nodes. We simply employ a linear projection followed by a row-wise Softmax operation to compute the attention for fusing $\boldsymbol{H}^{(\ell-1)}$ with $M^{(\ell)}$. Thus, $\mathfrak{F}^{(\ell)}$ can be expressed as:

$$
\begin{aligned}
\mathfrak{F}^{(\ell)}(\boldsymbol{H}^{(\ell-1)}, M^{(\ell)}; \boldsymbol{\Theta}_f^{(\ell)}) \\
= \boldsymbol{H}^{(\ell-1)} + \text{Softmax}(\boldsymbol{H}^{(\ell-1)} W_p^{(\ell)}) \cdot M^{(\ell)}
\end{aligned}
\tag{17}
$$

where $\boldsymbol{\Theta}_f^{(\ell)} = W_p^{(\ell)} \in \mathbb{R}^{d_{\ell-1} \times m}$ is the projection matrix. We choose a linear projection instead of MLPs or other alternatives because of its higher computational efficiency.

The key distinction between MTG and prefix-tuning (Li & Liang, 2021) is that MTG dynamically modifies the core message-passing within each layer via learnable parameters, while prefix-tuning only statically prepends learnable context as external input at each layer, with the relationship between prefixes being represented by the model's fixed attention modules. Therefore, MTG is not a simple transfer of prefix-tuning from LLMs to GNN-based GFMs.

### 4.3. Theoretical Analysis

We analyze the adaptation capacity of MTG through the lens of PS-Theory. Consider a pre-trained GFM $\Phi$ with $L$ layers as defined in Definition 3.1, and let $\mathcal{M}_0 \subset \mathbb{R}^{N \times d_0}$ be the compact smooth input manifold with intrinsic dimension $D_0$. Let $\mathcal{P}$ be the set of possible prompts for graph prompt tuning, and for any prompt $\boldsymbol{P} \in \mathcal{P}$, let $\mathcal{M}_0(\boldsymbol{P})$ be the perturbed input manifold. The final representation space under graph prompt tuning is $\mathcal{M}_{\text{PT}}^{(L)}(\boldsymbol{P}) = \Phi(\mathcal{M}_0(\boldsymbol{P}))$.

**Theorem 4.1** (Message Tuning Has Greater Adaptation Capacity). *For message tuning, we inject learnable message*

prototypes $\boldsymbol{M}^{(\ell)} \in \mathbb{R}^{m \times d_{\ell-1}}$ and fusion parameters $\boldsymbol{\Theta}_f^{(\ell)}$ at each layer $\ell$, resulting in a modified network $\Phi_{MTG}$. Let $\mathcal{M}_{MTG}^{(L)}$ be the final representation space under message tuning with optimally chosen parameters. Then for all $\boldsymbol{P} \in \mathcal{P}$, the following inequalities hold:

$$d_{int}(\mathcal{M}_{MTG}^{(L)}) \geq d_{int}(\mathcal{M}_{PT}^{(L)}(\boldsymbol{P})), \quad (18)$$

$$\mathcal{H}^{d_{int}}(\mathcal{M}_{MTG}^{(L)}) \geq \mathcal{H}^{d_{int}}(\mathcal{M}_{PT}^{(L)}(\boldsymbol{P})), \quad (19)$$

$$diam(\mathcal{M}_{MTG}^{(L)}) \geq diam(\mathcal{M}_{PT}^{(L)}(\boldsymbol{P})). \quad (20)$$

*Moreover, there exists a message tuning configuration such that the inequalities are strict.*

In the semantic context of the geometric properties of the prismatic space output by the GFM, this theorem reveals that MTG's adaptation capacity can exceed the upper bound of graph prompt tuning. The proof of Theorem 4.1 and further theoretical analysis are provided in Appendix C.

# 5. Experiments

In this section, we conduct extensive experiments to evaluate our proposed MTG on the ProG (Zi et al., 2024) benchmark, addressing the following six research questions:

**Q1**: How does MTG's adaptation capacity compare to graph prompt baselines? (Section 5.2)
**Q2**: How do different pre-training strategies affect MTG's adaptation capability? (Section 5.3)
**Q3**: Can MTG effectively mitigate negative transfer under different pre-training strategies? (Section 5.4)
**Q4**: How does MTG's performance vary on different backbone models and model depths? ( Appendix F.2)
**Q5**: What is MTG's computational efficiency (time/memory) compared to graph prompt methods? (Appendix F.3)
**Q6**: How sensitive is MTG to the number $m$ of message prototypes? (Appendix F.4)

## 5.1. Experiment Setting

**Datasets.** To investigate its adaptability, we evaluate MTG across 15 graph datasets, spanning 7 node classification datasets (including homophilic, heterophilic, and large-scale graphs) and 8 graph classification datasets (from biological, molecular, and social domains). We provide the dataset statistics and full details in Appendix D.2.

**Backbones.** Since the recent studies (Luo et al., 2024; 2025) have once again validated the powerful capabilities of GCN (Kipf & Welling, 2017), we choose it as the backbone to compare MTG with graph prompt baselines. We also investigate other commonly used backbones for GFMs, such as GraphSAGE (Hamilton et al., 2017), GAT (Veličković et al., 2018), GIN (Xu et al., 2019), and Graph Transformer (Ying et al., 2021). We provide the details in Appendix E.1.

**Pre-training Strategies.** We adopt six pre-training strategies including DGI (Veličković et al., 2019) and GraphMAE (Hou et al., 2022) (node level), EdgePreGPPT (Sun et al., 2022) and EdgePreGprompt (Liu et al., 2023b) (edge-level), and GraphCL (You et al., 2020) with SimGRACE (Xia et al., 2022) (graph-level). See Appendix E.2 for details.

**Graph Prompt Baselines.** We define negative transfer as adaptation underperforming a supervised learning baseline. We compare MTG with fine-tuning and major graph prompt methods: GPPT (Sun et al., 2022), Gprompt (Liu et al., 2023b), All-in-one (Sun et al., 2023a), GPF and GPF-plus (Fang et al., 2023). We provide the details in Appendix E.3.

**Implementation.** We adopt the same data splits and preprocessing procedures as in Zi et al. (2024). To ensure robustness, we repeat sampling five times and report average and standard deviation over these five results. Hyperparameters are optimized via random search. A comprehensive description of the experimental setup is provided in Appendix D.

## 5.2. Upper Bound Performance of Message Tuning

Few-shot node/graph classification is one of the most challenging downstream adaptation tasks for graph foundation models, as it requires learning the characteristics of an entire class using only a few samples. In Table 1, we present the best results achieved by various adaptation methods across 15 datasets, which represent the top adaptation performance from pre-trained models with different strategies. This offers an intuitive reflection of the upper bound performance of each type of adaptation method. The results in the tables demonstrate that our adaptation method MTG achieves a higher performance upper bound across all 15 datasets compared to state-of-the-art graph prompt tuning baselines, which aligns with our theoretical insights. Despite being trained on only a small number of parameters, MTG still exhibits a substantial advantage over supervised learning and fine-tuning approaches, which underscores its high parameter efficiency. Among node-level tasks, GPF-plus is the method that performs closest to MTG, while on graph-level tasks, All-in-one ranks as the second most effective method after MTG. We provide the detailed results in Appendix F.1.

## 5.3. Robustness Performance of MTG across Pre-training Strategies

We further analyze whether MTG exhibits strong robustness across different pre-training strategies through detailed experimental results. In Section 5.2, we have verified that GPF-plus and All-in-one are the best-performing graph prompt tuning methods for few-shot node classification and few-shot graph classification tasks, respectively. Therefore, we select these two methods along with Fine-tuning for a more detailed comparison with MTG on 1-shot node/graph classification tasks. Due to space constraints, we present the

*Table 1.* Performance comparison of adaptation methods on 1/3/5-shot node classification and graph classification (accuracy). The best and second-best results are highlighted in purple and blue, respectively.

| Method | Cora | | | Citeseer | | | Pubmed | | | Wisconsin | | | Texas | | | Actor | | | ogbn-arxiv | | |
|---|---|---|---|---|---|---|---|---|---|---|---|---|---|---|---|---|---|---|---|---|---|
| | 1-shot | 3-shot | 5-shot | 1-shot | 3-shot | 5-shot | 1-shot | 3-shot | 5-shot | 1-shot | 3-shot | 5-shot | 1-shot | 3-shot | 5-shot | 1-shot | 3-shot | 5-shot | 1-shot | 3-shot | 5-shot |
| Supervised | 26.56 | 37.79 | 50.25 | 21.78 | 35.18 | 41.22 | 39.37 | 57.33 | 67.88 | 41.60 | 41.03 | 39.43 | 37.97 | 40.78 | 43.91 | 20.57 | 18.62 | 21.92 | 10.99 | 19.03 | 22.38 |
| Fine-tuning | 52.61 | 51.97 | 62.66 | 35.05 | 45.08 | 39.54 | 46.74 | 65.40 | 70.91 | 40.69 | 42.40 | 42.97 | 46.88 | 43.13 | 47.19 | 20.74 | 22.11 | 22.92 | 16.21 | 27.34 | 28.84 |
| GPPT | 43.15 | 43.84 | 51.98 | 37.26 | 42.34 | 45.77 | 48.31 | 67.43 | 66.97 | 30.40 | 34.29 | 37.00 | 31.81 | 38.90 | 48.82 | 22.58 | 21.65 | 21.58 | 14.65 | 22.46 | 28.90 |
| Gprompt | 56.66 | 63.78 | 69.03 | 53.21 | 60.00 | 66.13 | 39.74 | 66.68 | 67.87 | 77.07 | 92.52 | 78.22 | 33.25 | 39.00 | 39.32 | 25.26 | 29.67 | 34.67 | 75.72 | 73.92 | 85.40 |
| All-in-one | 52.39 | 48.09 | 30.36 | 40.41 | 48.09 | 27.93 | 45.17 | 65.79 | 46.16 | 66.29 | 89.62 | 87.16 | 65.49 | 88.69 | 73.28 | 24.61 | 24.23 | 21.49 | 13.16 | 31.15 | 13.01 |
| GPF | 38.57 | 34.84 | 35.43 | 31.16 | 25.92 | 25.12 | 49.99 | 71.20 | 68.96 | 78.35 | 93.85 | 98.26 | 73.54 | 95.47 | 98.42 | 28.70 | 37.44 | 44.07 | 65.11 | 59.67 | 71.83 |
| GPF-plus | 55.77 | 56.38 | 66.22 | 59.67 | 72.48 | 75.73 | 46.64 | 70.85 | 69.59 | 82.11 | 98.15 | 99.01 | 76.10 | 97.66 | 99.12 | 29.32 | 43.59 | 44.58 | 71.98 | 64.63 | 66.88 |
| MTG (Ours) | 58.54 | 66.11 | 71.81 | 62.31 | 73.81 | 76.34 | 50.70 | 71.38 | 70.84 | 83.32 | 98.58 | 99.12 | 79.13 | 98.17 | 98.76 | 29.44 | 37.62 | 45.09 | 75.97 | 76.01 | 85.94 |

| Method | IMDB-B | | | COLLAB | | | PROTEINS | | | MUTAG | | | ENZYMES | | | COX2 | | | BZR | | | D&D | | |
|---|---|---|---|---|---|---|---|---|---|---|---|---|---|---|---|---|---|---|---|---|---|---|---|---|
| | 1-shot | 3-shot | 5-shot | 1-shot | 3-shot | 5-shot | 1-shot | 3-shot | 5-shot | 1-shot | 3-shot | 5-shot | 1-shot | 3-shot | 5-shot | 1-shot | 3-shot | 5-shot | 1-shot | 3-shot | 5-shot | 1-shot | 3-shot | 5-shot |
| Supervised | 57.30 | 53.33 | 62.60 | 47.23 | 50.77 | 55.23 | 56.36 | 61.33 | 62.90 | 65.20 | 59.47 | 73.47 | 20.58 | 15.96 | 25.67 | 27.08 | 65.15 | 64.99 | 25.80 | 52.35 | 51.48 | 55.33 | 59.77 | 63.59 |
| Fine-tuning | 57.75 | 66.10 | 65.40 | 48.10 | 56.10 | 60.72 | 63.44 | 62.72 | 63.33 | 65.47 | 59.87 | 75.33 | 22.21 | 22.71 | 7.46 | 76.19 | 69.97 | 73.19 | 34.69 | 52.22 | 72.96 | 57.15 | 59.70 | 64.71 |
| GPPT | 50.15 | 59.48 | 66.37 | 47.18 | 50.88 | 54.05 | 60.92 | 64.74 | 58.27 | 60.40 | 64.13 | 70.53 | 21.29 | 19.12 | 22.17 | 78.23 | 71.90 | 67.88 | 59.32 | 70.93 | 69.63 | 57.69 | 59.00 | 60.02 |
| Gprompt | 54.75 | 64.35 | 66.70 | 48.25 | 54.95 | 60.76 | 59.17 | 64.94 | 62.94 | 73.60 | 66.53 | 73.07 | 22.29 | 22.08 | 21.46 | 54.64 | 51.53 | 53.35 | 55.43 | 54.63 | 59.38 | 57.81 | 55.99 | 58.28 |
| All-in-one | 60.07 | 65.67 | 63.62 | 51.66 | 57.12 | 57.86 | 66.49 | 69.84 | 71.37 | 75.20 | 80.00 | 80.93 | 23.96 | 23.96 | 26.71 | 76.14 | 66.06 | 62.95 | 64.38 | 61.98 | 62.78 | 59.72 | 58.96 | 63.44 |
| GPF | 59.65 | 65.97 | 67.80 | 47.42 | 53.87 | 59.65 | 63.91 | 63.35 | 63.37 | 68.40 | 74.27 | 74.00 | 22.00 | 23.87 | 27.00 | 65.79 | 65.31 | 66.27 | 71.67 | 74.38 | 61.05 | 59.36 | 59.07 | 61.06 |
| GPF-plus | 57.93 | 64.38 | 68.13 | 47.24 | 56.50 | 60.68 | 62.92 | 63.55 | 63.51 | 65.20 | 75.20 | 73.87 | 22.92 | 24.46 | 26.87 | 33.78 | 65.25 | 72.87 | 71.17 | 71.67 | 71.54 | 57.62 | 59.51 | 64.80 |
| MTG (Ours) | 62.25 | 66.95 | 69.15 | 52.25 | 57.49 | 63.11 | 66.98 | 70.49 | 70.10 | 75.80 | 78.13 | 81.60 | 26.08 | 29.71 | 35.08 | 78.27 | 73.86 | 71.84 | 74.81 | 74.65 | 76.37 | 60.68 | 60.85 | 66.07 |

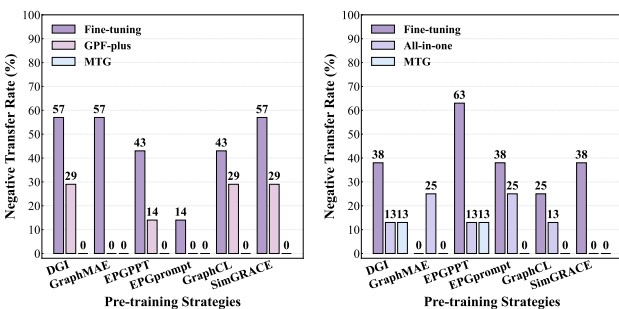

*Figure 2.* NTR comparison across different methods on (left) 1-shot node classification and (right) 1-shot graph classification.

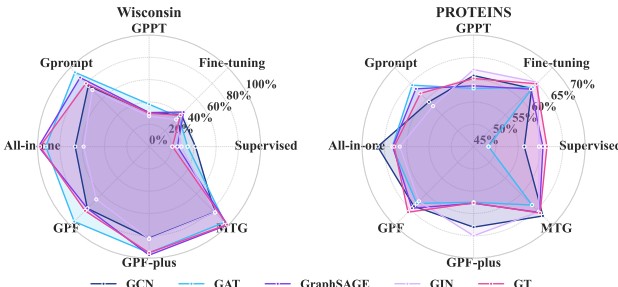

*Figure 3.* Performance comparison across backbone models for 1-shot node classification accuracy (%) on Wisconsin and 1-shot graph classification accuracy (%) on PROTEINS.

detailed result tables in Appendix F. As shown in Tables 7 and 8, Fine-tuning experiences performance collapse on the ogbn-arxiv dataset under the DGI and GraphCL pre-training strategies, with the accuracy dropping as low as 7.21% and 4.65%. Similarly, GPF-plus exhibits performance degradation on the Cora dataset under the DGI pre-training strategy, achieving an accuracy as low as 17.29%. All-in-one also shows relatively low performance on the IMDB-B dataset under the GraphMAE and EdgePreGprompt pre-training strategies. In contrast, MTG demonstrates more stable performance across all datasets and all pre-training strategies, highlighting broader compatibility and better robustness.

## 5.4. Mitigation of Negative Transfer

Compared to visual images and natural language, fine-tuning pre-trained models on graph data for downstream tasks is more prone to negative transfer (Wang et al., 2021).

Therefore, the ability to effectively mitigate negative transfer serves as an important criterion for evaluating an adaptation method. As shown in Figure 2, MTG achieves a lower NTR (Negative Transfer Rate) than GPF-plus and All-in-one, and is markedly superior to fine-tuning. It is particularly noteworthy that MTG completely eliminates negative transfer in the 1-shot node classification task. The definition of NTR and the detailed results can be found in Tables 7 and 8. We provide a preliminary theoretical analysis of why MTG mitigates negative transfer in Appendix C.2.

## 5.5. Additional Experiments

Additional experiments are presented in Appendix F, including the performance of MTG across **different backbone models and model depths** (Appendix F.2), a comparative analysis of **computational efficiency** between MTG and graph prompt methods (Appendix F.3) and a **sensitivity**

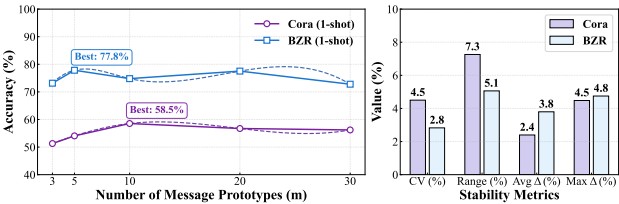

*Figure 4.* Sensitivity analysis of MTG performance to the number of message prototypes $m$: (left) stability trend (accuracy %) and (right) stability metrics (CV, Range, Avg, Max %).

**analysis** on hyperparameter $m$ (Appendix F.4). Figures 3 and 4 provide an overview of selected experimental results, summarizing a performance comparison across different backbone models and a comprehensive sensitivity analysis.

## 6. Conclusion

In this paper, we propose the Prismatic Space Theory (PS-Theory) to quantify the adaptation capacity of graph prompt tuning. Inspired by PS-Theory, we introduce Message Tuning for GFMs (MTG), a lightweight adaptation method that injects learnable message prototypes into each layer of the GNN backbone without updating the frozen pre-trained weights. Theoretical and empirical results demonstrate that MTG outshines graph prompt tuning for GFMs.

## Limitations

**Prismatic Space Theory.** Our theoretical framework relies on several assumptions that delineate its scope. First, PS-Theory models the input data as a compact and smooth manifold $\mathcal{M}_0$ (Definition 3.5), and characterizes each GFM layer as a piecewise linear map. This characterization is exact for backbones equipped with piecewise linear activations such as ReLU or LeakyReLU. For backbones with smooth non-linearities (e.g., GELU, $\tanh$), and for the Softmax fusion used in Theorem 4.1, the framework relies on linear approximation and should be interpreted as approximate. Second, the prismatic space construction in Proposition 3.7 and the measure formula in Theorem 3.13 assume injectivity on each linear region. Third, the intrinsic dimension bound in Theorem 3.12 is analytical and not directly tractable to compute numerically. Finally, PS-Theory quantifies the geometric adaptation capacity of an adaptation method, namely the dimension, measure, and diameter of the reachable output space. Translating this capacity into precise generalization or optimization guarantees on specific downstream tasks is left to future work.

**Message Tuning.** MTG is designed for GNN-based GFMs that admit the unified layer formulation in Definition 3.1, including representative MPNN and Graph Transformer backbones. As a result, MTG requires white-box, layer-wise access to the pre-trained backbone, and is not

directly applicable to GFM paradigms whose backbones do not expose internal message-passing layers, such as black-box LLM-based or API-only graph foundation models. Furthermore, the prototype number $m$ and the lightweight linear fusion operator $\mathfrak{F}^{(\ell)}$ are deliberately kept simple for efficiency. More expressive fusion designs (e.g., MLP-based or attention-based variants) may yield further gains, but at the cost of the parameter and runtime efficiency that motivate MTG. Lastly, although Theorem 4.1 establishes that MTG strictly exceeds the adaptation capacity of graph prompt tuning, an enlarged capacity does not automatically guarantee monotonic improvements in every extreme low-resource regime. We leave a tighter analysis of the capacity and generalization trade-off under few-shot supervision as a promising direction for future work.

**Experiments.** Empirically, we validate MTG on the ProG benchmark (Zi et al., 2024) under the standard few-shot protocol, reporting 1/3/5-shot performance for both node classification and graph classification over 15 datasets, with multiple GNN backbones and representative self-supervised pre-training strategies (Section 5). Generalization of our findings to other task families (e.g., link-level prediction, regression, or richly supervised fine-tuning pipelines) and to encoders outside the ProG-style GFM setting is not established by this study. Our evaluation moreover assumes benign training and inference conditions and does not probe adversarial and backdoor robustness across node-level and graph-level learning pipelines (Lin et al., 2023; Jin et al., 2025; Wang et al., 2025b). Likewise, we do not address uncertainty quantification (Lin et al., 2024b), graph-level out-of-distribution detection (Lin et al., 2025), or multi-task causality modeling (Lin et al., 2026). These remain important directions for extending PS-Theory and MTG.

## Acknowledgments

The authors would like to thank the anonymous reviewers and area chairs for their valuable comments and suggestions. This work was partially supported by the Strategic Priority Research Program of the Chinese Academy of Sciences (No. XDB0680101), the National Natural Science Foundation of China (No. 62472416 and 62402491), and the CAS Project for Young Scientists in Basic Research (No. YSBR-008). The model training was performed on the robotic AI-Scientist platform of Chinese Academy of Science.

## Impact Statement

This paper presents work whose goal is to advance the field of Machine Learning. There are many potential societal consequences of our work, none of which we feel must be specifically highlighted here.

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

# Appendix

## A. Related Work

The adaptation of GNN-based GFMs involves tailoring models or adjusting input data to align with specific downstream tasks or domains through techniques such as fine-tuning and graph prompt learning.

**Graph Prompt Learning.** As a lightweight tuning method, graph prompt learning (Sun et al., 2023b) typically freezes pre-trained model parameters while introducing additional learnable components to manipulate data and fill the task gap by reformulating downstream tasks to the pretext. Graph prompt tuning methods discussed in this paper modify graph topology or node features in the input space to enhance task performance and boost adaptability. For instance, GPF (Fang et al., 2023) introduces an optimizable uniform feature vector for all nodes to adapt pre-trained GNNs across strategies, while All-in-one (Sun et al., 2023a) reformulates node-level and edge-level tasks to graph-level tasks and treats an additional subgraph as a prompt that merges with the node subgraph. Beyond graph prompt tuning, there exists a broader category of graph prompt methods that insert prompts at various locations within the model. For instance, GPPT (Sun et al., 2022) transforms node classification into link prediction via class-specific token pairs, while GraphPrompt (Liu et al., 2023b) unifies tasks through subgraph similarity and learns task-specific prompt vectors to adapt the Readout operation, bridging link prediction and downstream tasks. It should be noted that the scope of our theoretical analysis is limited to graph prompt tuning, whereas the experimental evaluation of MTG extends to a broader set of graph prompt methods for comparative assessment.

## B. Extra Materials for Prismatic Space Theory

**Reading Guideline**: Appendix B is organized in strict accordance with the order in which definitions, theorems, and corollaries appear in the main text, serving as a detailed supplement. This includes supplementary explanations of definitions, lemmas required for proving theorems, interpretations of theorems, and more. **Some mathematical concepts not explicitly defined or elaborated in this paper can be found in Halmos (1950); Greub (1975); Lang (1993); Krantz & Parks (2008); Lee (2011).** We recommend that readers first review the related work, such as other related theoretical works and relevant mathematical textbooks, to establish a theoretical foundation before proceeding through the main text in sequence with the aid of Appendix B.

**Notation**: The notation used in this paper has been closely aligned with the standardized notation provided in https://github.com/goodfeli/dlbook_notation/. A summary of the primary notation used is provided in the table below.

### B.1. Related Work on Prismatic Space Theory

We introduce, for the first time, the Prismatic Space Theory to provide a unified analysis of adaptation methods for graph foundation models. In constructing this theoretical framework, we adopt the perspective of piecewise linear maps, an approach that is not entirely new, as several outstanding theoretical studies have employed similar ideas to analyze ReLU neural networks (Arora et al., 2018; Zhang & Wu, 2020; Liu et al., 2023a; Fu, 2025; Beshkov, 2025).

Previously, only Wang et al. (2025a) conducted theoretical research on graph prompt tuning for GFMs, explaining why graph prompt tuning works from the perspective of data operations, primarily using mathematical tools from linear algebra, convex optimization, and probability. In contrast, our Prismatic Space Theory offers a more profound and fundamental geometric perspective to quantify the upper bound of graph prompt tuning's capability.

### B.2. Details of Definition 3.1

The unified GFM layer formulation provided in Definition 3.1 offers a general framework that encapsulates a wide range of popular GNN architectures. The three core operators $\mathfrak{A}^{(\ell)}$ (attention), $\mathfrak{M}^{(\ell)}$ (message fusion), and $\mathfrak{U}^{(\ell)}$ (update) can be instantiated in different ways to recover specific models. Below, we delineate how several classic models are special cases of this unified formulation.

**GraphSAGE (Hamilton et al., 2017)** employs a uniform (or degree-based) attention weight over the sampled neighborhood, a configurable message aggregation function (e.g., mean, pool, LSTM), and an update function that concatenates the node's previous representation with the aggregated message.

Attention Operator $\mathfrak{A}^{(\ell)}$ often uses a static, uniform attention weight $\frac{1}{|\mathcal{N}(i)|}$ for each sampled neighbor of node $i$, or a learned

*Table 2.* Primary Notation.

| Notation | Description |
| --- | --- |
| $\mathcal{G}$ | A graph |
| $\mathcal{V}$ | The set of nodes |
| $\mathcal{E}$ | The set of edges |
| $N$ | Number of nodes |
| $\ell$ | Number of layers |
| $\mathbb{R}$ | The set of real numbers |
| $\{0, 1\}$ | The set containing 0 and 1 |
| $\boldsymbol{X}, \boldsymbol{A}, \boldsymbol{H}^{(\ell)}$ | The matrices |
| $\mathfrak{A}, \mathfrak{M}, \mathfrak{U}$ | The operators |
| $\boldsymbol{\Theta}$ | The parameters |
| $\mathbb{V}, \mathbb{S}, \mathbb{H}^{(\ell)}$ | The sets |
| $F^{(\ell)} : \mathbb{H}^{(\ell-1)} \to \mathbb{H}^{(\ell)}$ | The map $F^{(\ell)}$ with domain $\mathbb{H}^{(\ell-1)}$ and range $\mathbb{H}^{(\ell)}$ |
| $\mathcal{X}, \mathcal{M}$ | The manifold or space |
| $\Phi^{(\ell)} = F^{(\ell)} \circ \cdots \circ F^{(1)}$ | The composition of maps |
| $d_{\text{int}}$ | The intrinsic dimension |
| $\boldsymbol{J}^{(\ell)}(\boldsymbol{H})$ | The Jacobian matrix |
| $\sigma_i^{(\ell)}$ | The singular values |
| $\mathcal{H}^s$ | The $s$-dimensional Hausdorff measure |
| $\{C_k\}$ | The set of cells |
| $R_1, ..., R_L$ | The regions |
| $\Omega^{(\ell)}$ | The set of polytopic regions |

weight based on node degree in some variants. It can be represented as a matrix $\boldsymbol{S}^{(\ell)}$:

$$\mathfrak{A}^{(\ell)}\left(\boldsymbol{A}, \boldsymbol{H}^{(\ell-1)}; \boldsymbol{\Theta}_a^{(\ell)}\right) = \boldsymbol{S}^{(\ell)}, \quad \boldsymbol{S}_{ij}^{(\ell)} = \begin{cases} \frac{1}{|\mathcal{N}(i)|} & \text{if } j \in \mathcal{N}(i) \\ 0 & \text{otherwise} \end{cases} \tag{21}$$

where $\mathcal{N}(i)$ denotes the sampled neighbors of node $i$ and no parameters are used ($\boldsymbol{\Theta}_a^{(\ell)} = \varnothing$).

Message Fusion Operator $\mathfrak{M}^{(\ell)}$ aggregates messages from the sampled neighborhood using the specified aggregator AGGREGATE$^{(\ell)}$ (e.g., mean, pool, LSTM). For the mean aggregator:

$$\mathfrak{M}^{(\ell)}\left(\boldsymbol{S}^{(\ell)}, \boldsymbol{H}^{(\ell-1)}; \boldsymbol{\Theta}_m^{(\ell)}\right) = \boldsymbol{S}^{(\ell)} \boldsymbol{H}^{(\ell-1)}, \tag{22}$$

where parameters $\boldsymbol{\Theta}_m^{(\ell)}$ depend on the choice of aggregator.

Update Operator $\mathfrak{U}^{(\ell)}$ concatenates the node's previous representation $\boldsymbol{H}^{(\ell-1)}$ with the aggregated neighborhood message, applies a linear transformation $\boldsymbol{W}^{(\ell)}$, and a non-linear activation function $\sigma$:

$$\mathfrak{U}^{(\ell)}\left(\boldsymbol{S}^{(\ell)} \boldsymbol{H}^{(\ell-1)}, \boldsymbol{H}^{(\ell-1)}; \boldsymbol{\Theta}_u^{(\ell)}\right) = \sigma\left(\boldsymbol{W}^{(\ell)} \cdot \text{CONCAT}(\boldsymbol{H}^{(\ell-1)}, \boldsymbol{S}^{(\ell)} \boldsymbol{H}^{(\ell-1)})\right), \tag{23}$$

where $\boldsymbol{\Theta}_u^{(\ell)} = \boldsymbol{W}^{(\ell)}$.

The resulting layer formulation for the mean aggregator is:

$$\boldsymbol{H}^{(\ell)} = \sigma\left(\boldsymbol{W}^{(\ell)} \cdot \text{CONCAT}(\boldsymbol{H}^{(\ell-1)}, \boldsymbol{S}^{(\ell)} \boldsymbol{H}^{(\ell-1)})\right). \tag{24}$$

**GAT (Graph Attention Network)** (Veličković et al., 2018) introduces a learnable self-attention mechanism to compute dynamic attention weights between nodes.

Attention Operator $\mathfrak{A}^{(\ell)}$ computes pairwise attention coefficients $\alpha_{ij}$ for nodes $i$ and $j$ using a learnable function (a shared attentional mechanism $a$):

$$e_{ij}^{(\ell)} = a(\boldsymbol{W}^{(\ell)} \boldsymbol{h}_i^{(\ell-1)}, \boldsymbol{W}^{(\ell)} \boldsymbol{h}_j^{(\ell-1)}) = \text{LeakyReLU}\left(\tilde{\mathbf{a}}^{(\ell)T}[\boldsymbol{W}^{(\ell)} \boldsymbol{h}_i^{(\ell-1)} \| \boldsymbol{W}^{(\ell)} \boldsymbol{h}_j^{(\ell-1)}]\right), \tag{25}$$

$$\alpha_{ij}^{(\ell)} = \text{Softmax}(e_{ij}) = \frac{\exp(e_{ij})}{\sum_{k \in \mathcal{N}(i)} \exp(e_{ik})}, \tag{26}$$

$$\mathfrak{A}^{(\ell)} \left( \boldsymbol{A}, \boldsymbol{H}^{(\ell-1)}; \boldsymbol{\Theta}_a^{(\ell)} \right) = \boldsymbol{A}_a^{(\ell)}, \quad \boldsymbol{A}_a^{(\ell)}{}_{ij} = \begin{cases} \alpha_{ij}^{(\ell)} & \text{if } j \in \mathcal{N}(i) \\ 0 & \text{otherwise} \end{cases} \tag{27}$$

where $\boldsymbol{h}_i^{(\ell-1)}$ represents the vector of node $i$ at the $\ell-1$-th layer, $^T$ represents transposition, $\|$ is the concatenation operation, and the attention mechanism $a$ is a single-layer feedforward neural network parametrized by a weight vector $\tilde{\mathbf{a}}^{(\ell)}$ ($\boldsymbol{\Theta}_a^{(\ell)} = \{\tilde{\mathbf{a}}^{(\ell)}, \boldsymbol{W}^{(\ell)}\}$).

Message Fusion Operator $\mathfrak{M}^{(\ell)}$ performs a weighted sum of the transformed neighbor features based on the computed attention weights:

$$\mathfrak{M}^{(\ell)} \left( \boldsymbol{A}_a^{(\ell)}, \boldsymbol{H}^{(\ell-1)}; \boldsymbol{\Theta}_m^{(\ell)} \right) = \boldsymbol{A}_a^{(\ell)} \cdot (\boldsymbol{H}^{(\ell-1)} \boldsymbol{W}^{(\ell)}), \tag{28}$$

where $\boldsymbol{W}^{(\ell)}$ is a shared linear transformation applied to every node and is also used by the Attention Operator ($\boldsymbol{\Theta}_m^{(\ell)} = \boldsymbol{W}^{(\ell)}$). The operations of the Attention Operator and the Message Fusion Operator are partially overlapping.

Update Operator $\mathfrak{U}^{(\ell)}$ combines the aggregated representations from multiple attention heads, typically through concatenation (for intermediate layers) or averaging (for the output layer), followed by application of a non-linear activation function $\sigma$ to produce the new node representations:

$$\mathfrak{U}^{(\ell)} \left( \{\boldsymbol{A}_a^{(\ell)k} \cdot (\boldsymbol{H}^{(\ell-1)} \boldsymbol{W}^{(\ell)k})\}_{k=1}^K, \boldsymbol{H}^{(\ell-1)}; \boldsymbol{\Theta}_u^{(\ell)} \right) = \bigg\|_{k=1}^K \sigma \left( \boldsymbol{A}_a^{(\ell)k} \cdot (\boldsymbol{H}^{(\ell-1)} \boldsymbol{W}^{(\ell)k}) \right), \tag{29}$$

where $\|$ represents concatenation, $K$ represents the number of attention heads and no parameters are used in this operator ($\boldsymbol{\Theta}_u^{(\ell)} = \varnothing$).

The resulting layer formulation is:

$$\boldsymbol{H}^{(\ell)} = \bigg\|_{k=1}^K \sigma \left( \boldsymbol{A}_a^{(\ell)k} \cdot (\boldsymbol{H}^{(\ell-1)} \boldsymbol{W}^{(\ell)k}) \right). \tag{30}$$

**GIN (Graph Isomorphism Network)** (Xu et al., 2019) uses a fixed, uniform attention weight for neighbors and a powerful update function based on an MLP to achieve expressiveness equivalent to the Weisfeiler-Lehman graph isomorphism test.

Attention Operator $\mathfrak{A}^{(\ell)}$ employs a static attention weight of 1 for all neighbors and a weight of $(1 + \epsilon^{(\ell)})$ for the central node itself:

$$\mathfrak{A}^{(\ell)} \left( \boldsymbol{A}, \boldsymbol{H}^{(\ell-1)}; \boldsymbol{\Theta}_a^{(\ell)} \right) = \tilde{\boldsymbol{A}}_\epsilon = \boldsymbol{A} + (1 + \epsilon^{(\ell)}) \boldsymbol{I} = \tilde{\boldsymbol{A}}_\epsilon, \tag{31}$$

where $\boldsymbol{\Theta}_a^{(\ell)} = \epsilon^{(\ell)}$ is a potentially learnable parameter.

Message Fusion Operator $\mathfrak{M}^{(\ell)}$ sums the neighbor messages and the scaled central node's message:

$$\mathfrak{M}^{(\ell)} \left( \tilde{\boldsymbol{A}}_\epsilon, \boldsymbol{H}^{(\ell-1)}; \boldsymbol{\Theta}_m^{(\ell)} \right) = \tilde{\boldsymbol{A}}_\epsilon \boldsymbol{H}^{(\ell-1)}, \tag{32}$$

where no parameters are used ($\boldsymbol{\Theta}_m^{(\ell)} = \varnothing$).

Update Operator $\mathfrak{U}^{(\ell)}$ applies a multi-layer perceptron ($\text{MLP}^{(\ell)}$) to the fused message:

$$\mathfrak{U}^{(\ell)} \left( \tilde{\boldsymbol{A}}_\epsilon \boldsymbol{H}^{(\ell-1)}, \boldsymbol{H}^{(\ell-1)}; \boldsymbol{\Theta}_u^{(\ell)} \right) = \text{MLP}^{(\ell)} \left( \tilde{\boldsymbol{A}}_\epsilon \boldsymbol{H}^{(\ell-1)} \right), \tag{33}$$

where $\boldsymbol{\Theta}_u^{(\ell)}$ are the parameters of the MLP.

The resulting layer formulation is:

$$\boldsymbol{H}^{(\ell)} = \text{MLP}^{(\ell)} \left( \left( \boldsymbol{A} + (1 + \epsilon^{(\ell)}) \boldsymbol{I} \right) \boldsymbol{H}^{(\ell-1)} \right). \tag{34}$$

**GT (Graph Transformer)** (Ying et al., 2021) enhances the standard transformer architecture to incorporate structural information of graphs, often by augmenting the self-attention mechanism with structural biases.

Attention Operator $\mathfrak{A}^{(\ell)}$ computes the query, key matrices $\boldsymbol{Q}^{(\ell)}, \boldsymbol{K}^{(\ell)} \in \mathbb{R}^{d_{\ell-1} \times d_\ell}$ via linear projections, with the core attention weight $\hat{\boldsymbol{A}}^{(\ell)}$ formulated as a sum of standard semantic attention and a structural attention component $\boldsymbol{B}^{(\ell)}$ (e.g., from positional encodings, edge features, connectivity patterns or node degrees).

$$\boldsymbol{Q}^{(\ell)} = \boldsymbol{H}^{(\ell-1)} \boldsymbol{W}_Q^{(\ell)}, \quad \boldsymbol{K}^{(\ell)} = \boldsymbol{H}^{(\ell-1)} \boldsymbol{W}_K^{(\ell)}, \tag{35}$$

$$\hat{\boldsymbol{A}}^{(\ell)} = \text{Softmax} \left( \frac{\boldsymbol{Q}^{(\ell)} (\boldsymbol{K}^{(\ell)})^T}{\sqrt{d_\ell}} + \boldsymbol{B}^{(\ell)} \right), \tag{36}$$

$$\mathfrak{A}^{(\ell)} \left( \boldsymbol{A}, \boldsymbol{H}^{(\ell-1)}; \boldsymbol{\Theta}_a^{(\ell)} \right) = \hat{\boldsymbol{A}}^{(\ell)}, \tag{37}$$

where $\boldsymbol{\Theta}_a^{(\ell)}$ including the projection weights for $\boldsymbol{Q}^{(\ell)}, \boldsymbol{K}^{(\ell)}$ and parameters for computing $\boldsymbol{B}^{(\ell)}$.

Message Fusion Operator $\mathfrak{M}^{(\ell)}$ computes the value matrice $\boldsymbol{V}^{(\ell)} \in \mathbb{R}^{d_{\ell-1} \times d_\ell}$ via linear projection $\boldsymbol{W}_V^{(\ell)}$ and performs the weighted aggregation of the value vectors using the computed attention matrix $\hat{\boldsymbol{A}}^{(\ell)}$:

$$\mathfrak{M}^{(\ell)} \left( \hat{\boldsymbol{A}}^{(\ell)}, \boldsymbol{H}^{(\ell-1)}; \boldsymbol{\Theta}_m^{(\ell)} \right) = \hat{\boldsymbol{A}}^{(\ell)} \boldsymbol{V}^{(\ell)} = \hat{\boldsymbol{A}}^{(\ell)} \cdot (\boldsymbol{H}^{(\ell-1)} \boldsymbol{W}_V^{(\ell)}), \tag{38}$$

where $\boldsymbol{\Theta}_m^{(\ell)} = \boldsymbol{W}_V^{(\ell)}$.

Update Operator $\mathfrak{U}^{(\ell)}$ applies a residual connection, layer normalization (LN), a position-wise feed-forward network (FFN), another residual connection, and layer normalization.

$$\tilde{\boldsymbol{H}}^{(\ell)} = \text{LN} \left( \boldsymbol{H}^{(\ell-1)} + \hat{\boldsymbol{A}}^{(\ell)} \cdot (\boldsymbol{H}^{(\ell-1)} \boldsymbol{W}_V^{(\ell)}) \right), \tag{39}$$

$$\text{FFN}^{(l)}(\tilde{\boldsymbol{H}}^{(\ell)}) = \sigma \left( \tilde{\boldsymbol{H}}^{(\ell)} \boldsymbol{W}_1^{(l)} + \boldsymbol{b}_1^{(l)} \right) \boldsymbol{W}_2^{(l)} + \boldsymbol{b}_2^{(l)} \tag{40}$$

$$\mathfrak{U}^{(\ell)} \left( \hat{\boldsymbol{A}}^{(\ell)} \boldsymbol{V}^{(\ell)}, \boldsymbol{H}^{(\ell-1)}; \boldsymbol{\Theta}_u^{(\ell)} \right) = \text{LN} \left( \tilde{\boldsymbol{H}}^{(\ell)} + \text{FFN}^{(\ell)}(\tilde{\boldsymbol{H}}^{(\ell)}) \right), \tag{41}$$

where $\boldsymbol{\Theta}_u^{(\ell)}$ are the parameters of the $\text{FFN}^{(\ell)}$. Multi-head self-attention (MHA) can also correspond to the Update Operator.

The resulting layer formulation is:

$$\boldsymbol{H}^{(\ell)} = \text{LN} \left( \text{LN} \left( \boldsymbol{H}^{(\ell-1)} + \hat{\boldsymbol{A}}^{(\ell)} \boldsymbol{V}^{(\ell)} \right) + \text{FFN}^{(\ell)} \left( \text{LN} \left( \boldsymbol{H}^{(\ell-1)} + \hat{\boldsymbol{A}}^{(\ell)} \boldsymbol{V}^{(\ell)} \right) \right) \right). \tag{42}$$

This analysis demonstrates that the proposed unified GFM layer provides a powerful and expressive framework that generalizes a broad spectrum of prevalent GNN architectures. The specific choices of the operators $\mathfrak{A}^{(\ell)}$, $\mathfrak{M}^{(\ell)}$, and $\mathfrak{U}^{(\ell)}$ determine the particular inductive biases and capabilities of the resulting model.

### B.3. Details of Definition 3.2

**Definition B.1** (Polyhedral Region). In the context of Euclidean spaces, a polyhedral region (or polyhedron) is a subset of $\mathbb{R}^n$ defined by a finite set of linear inequalities. Formally, a set $R \subseteq \mathbb{R}^n$ is a polyhedral region if there exist matrices $\boldsymbol{A} \in \mathbb{R}^{m \times n}$ and vectors $\boldsymbol{b} \in \mathbb{R}^m$ such that:

$$R = \{ \boldsymbol{x} \in \mathbb{R}^n \mid \boldsymbol{A}\boldsymbol{x} \leq \boldsymbol{b} \}, \tag{43}$$

where the inequality is applied component-wise.

*Remark* B.2. A polyhedral region may be described as the intersection of finitely many closed half-spaces and/or hyperplanes, making it a convex polytope (possibly unbounded). In many analytical contexts, polyhedral regions are assumed to be non-empty and may be required to have a non-empty interior to avoid degenerate cases.

**Definition B.3** (Polyhedral Region in Matrix Space). A set $R \subseteq \mathbb{R}^{N \times d}$ is a polyhedral region if there exists a matrix $\boldsymbol{A} \in \mathbb{R}^{m \times Nd}$ and a vector $\boldsymbol{b} \in \mathbb{R}^m$ such that:

$$R = \left\{ \boldsymbol{H} \in \mathbb{R}^{N \times d} \mid \boldsymbol{A} \cdot \text{vec}(\boldsymbol{H}) \leq \boldsymbol{b} \right\}, \tag{44}$$

where $\text{vec}(\boldsymbol{H}) \in \mathbb{R}^{Nd}$ denotes the vectorization of the matrix $\boldsymbol{H}$ (i.e., the column vector obtained by stacking the columns of $\boldsymbol{H}$). The inequality $\leq$ is applied component-wise.

*Remark* B.4. Since the spaces $\mathbb{R}^{\alpha \times \beta}$ and $\mathbb{R}^{\alpha\beta}$ are isomorphic as vector spaces via the vectorization operation $\text{vec} : \mathbb{R}^{\alpha \times \beta} \to \mathbb{R}^{\alpha\beta}$ (which stacks the columns of a matrix into a vector) and its inverse $\text{unvec} : \mathbb{R}^{\alpha\beta} \to \mathbb{R}^{\alpha \times \beta}$, many theorems and proofs in this paper do not strictly distinguish between the matrix form and the vectorized form. This isomorphism allows us to apply concepts from Euclidean geometry and measure theory directly to matrix-valued functions by considering their vectorized counterparts, without loss of generality. Consequently, in the following analysis, we may interchangeably use matrix or vector representations as convenient, ensuring that all results hold equivalently in both forms.

**Definition B.5** (Piecewise Linear Function). A function $f : \mathbb{R}^n \to \mathbb{R}^m$ is said to be piecewise linear if there exists a finite set of polyhedral regions $\{R_i\}_{i=1}^K$ such that $\mathbb{R}^n = \bigcup_{i=1}^K R_i$ and $f$ is affine on each $R_i$, i.e., $f(\boldsymbol{x}) = \boldsymbol{A}_i \boldsymbol{x} + \boldsymbol{b}_i$ for all $\boldsymbol{x} \in R_i$, where $\boldsymbol{A}_i \in \mathbb{R}^{m \times n}$ and $\boldsymbol{b}_i \in \mathbb{R}^m$.

**Definition B.6** (Jacobian of a Matrix Map). For a function $F : \mathbb{R}^{N \times d_{\text{in}}} \to \mathbb{R}^{N \times d_{\text{out}}}$ that is differentiable at a point $\boldsymbol{H}$, the Jacobian of $F$ at $\boldsymbol{H}$ is defined as the Jacobian matrix of the vectorized function. Specifically, let $f : \mathbb{R}^{Nd_{\text{in}}} \to \mathbb{R}^{Nd_{\text{out}}}$ be given by $f(\boldsymbol{h}) = \text{vec}(F(\text{unvec}(\boldsymbol{h})))$. Then, the Jacobian matrix $\boldsymbol{J}_F(\boldsymbol{H}) \in \mathbb{R}^{Nd_{\text{out}} \times Nd_{\text{in}}}$ is:

$$\boldsymbol{J}_F(\boldsymbol{H}) = \left. \frac{\partial f}{\partial \boldsymbol{h}} \right|_{\boldsymbol{h}=\text{vec}(\boldsymbol{H})}. \tag{45}$$

This matrix contains all first-order partial derivatives of the vectorized output with respect to the vectorized input.

**Lemma B.7** (Composition of Piecewise Linear Functions). *If $f : \mathbb{R}^n \to \mathbb{R}^m$ and $g : \mathbb{R}^m \to \mathbb{R}^p$ are piecewise linear functions, then the composition $g \circ f : \mathbb{R}^n \to \mathbb{R}^p$ is also piecewise linear.*

*Proof.* Since $f$ is piecewise linear, there exists a partition of $\mathbb{R}^n$ into polyhedral regions $R_i$ such that $f$ is affine on each $R_i$. Similarly, $g$ is piecewise linear with polyhedral regions $S_j$ in $\mathbb{R}^m$ where $g$ is affine. For each $i$ and $j$, consider the set $R_i \cap f^{-1}(S_j)$. Since $f$ is affine on $R_i$, $f(R_i)$ is a polyhedral set, and $f^{-1}(S_j) \cap R_i$ is polyhedral (as the intersection of polyhedral sets). On $R_i \cap f^{-1}(S_j)$, $g \circ f$ is affine because $f$ is affine and $g$ is affine on $S_j$. The collection of all such sets $R_i \cap f^{-1}(S_j)$ covers $\mathbb{R}^n$, and there are finitely many such sets. Thus, $g \circ f$ is piecewise linear. $\square$

### B.4. Proof of Proposition 3.3

*Proof.* Based on the derivations in Appendix B.1, we observe that most operators primarily involve matrix multiplication or can be approximated by matrix multiplications. Some operators further apply a piecewise linear activation function (e.g., ReLU or LeakyReLU) or an MLP based on ReLU after the matrix multiplication. As a result, these operators are generally piecewise linear functions. Their piecewise linearity stems directly from the piecewise linear activation functions used. Since the activation functions are continuous, the continuity of these operators is obvious. Piecewise linear functions are differentiable a.e. because they are differentiable in the interior of each polyhedral region (where they are affine) and non-differentiable only on the boundaries, which have Lebesgue measure zero. $\square$

*Remark* B.8. Linear functions are considered a special case of piecewise linear functions. If operator $\mathfrak{A}^{(\ell)}$ computes scaled dot-product attention, it somewhat exceeds the scope of our theoretical framework. Alternatively, approximating the computation of dynamic attention using piecewise linear mappings may, from a mathematical limit perspective, exhibit certain compatibility with the theoretical framework presented in this paper. This constitutes a promising direction for future research aimed at extending the current theory.

### B.5. Proof of Proposition 3.4

*Proof.* The layer map $F^{(\ell)}$ is defined by the composition of the operators $\mathfrak{A}^{(\ell)}, \mathfrak{M}^{(\ell)}, \mathfrak{U}^{(\ell)}$, as given in Definition 3.1. This can be viewed as a function $F^{(\ell)}$ that maps $\boldsymbol{H}^{(\ell-1)}$ to $\boldsymbol{H}^{(\ell)}$. Since each operator is piecewise linear, and by Lemma B.7, the composition of piecewise linear functions is itself piecewise linear. Therefore, $F^{(\ell)}$ is a piecewise linear function. More formally, let $f_1 = \mathfrak{A}^{(\ell)}$, $f_2 = \mathfrak{M}^{(\ell)}$, and $f_3 = \mathfrak{U}^{(\ell)}$. Then $F^{(\ell)} = f_3 \circ (f_2 \circ (f_1, \text{id}), \text{id})$, where id denotes the identity

function (which is linear and thus piecewise linear). The composition involves piecewise linear functions and Cartesian products (which preserve piecewise linearity), so $F^{(\ell)}$ is piecewise linear.

By Definition 3.2, there exists a finite set of polyhedral regions $\{R_i\}_{i=1}^{K}$ such that $\mathbb{R}^{N \times d_{\ell-1}} = \bigcup_{i=1}^{K} R_i$ and $F^{(\ell)}$ is affine on each $R_i$, i.e., $F^{(\ell)}(\boldsymbol{H}) = \text{unvec}(\boldsymbol{A}_i \cdot \text{vec}(\boldsymbol{H}) + \boldsymbol{b}_i)$ for all $\boldsymbol{H} \in R_i$, where $\boldsymbol{A}_i \in \mathbb{R}^{Nd_\ell \times Nd_{\ell-1}}$ and $\boldsymbol{b}_i \in \mathbb{R}^{Nd_\ell}$. An affine function is differentiable everywhere in the interior of its region. The polyhedral regions $R_i$ are closed and have boundaries that are sets of measure zero (since they are defined by finite sets of linear inequalities). Therefore, $F^{(\ell)}$ is differentiable almost everywhere (a.e.)—specifically, in the interior of each region $R_i$. At any point $\boldsymbol{H}$ where $F^{(\ell)}$ is differentiable (i.e., in the interior of some $R_i$), the derivative is given by the constant matrix $\boldsymbol{A}_i$. The Jacobian matrix $\boldsymbol{J}^{(\ell)}(\boldsymbol{H})$ is precisely this matrix $\boldsymbol{A}_i$, which exists and has dimensions $\mathbb{R}^{Nd_\ell \times Nd_{\ell-1}}$ (since the input space has dimension $Nd_{\ell-1}$ and the output space has dimension $Nd_\ell$). Hence, for any point $\boldsymbol{H}$ where $F^{(\ell)}$ is differentiable, the Jacobian $\boldsymbol{J}^{(\ell)}(\boldsymbol{H})$ exists. $\square$

### B.6. Details of Definition 3.5

**Definition B.9** (Compact Smooth Manifold). A set $\mathcal{M} \subset \mathbb{R}^n$ is said to be a *compact smooth manifold* of dimension $D_0$ if it satisfies the following two conditions:

  1. (Smooth Structure) For every point $p \in \mathcal{M}$, there exists an open neighborhood $U \subset \mathbb{R}^n$ containing $p$ and a smooth ($C^\infty$) mapping $F : U \to \mathbb{R}^{n-D_0}$ such that: $U \cap \mathcal{M} = F^{-1}(0) = \{x \in U \mid F(x) = 0\}$ and the Jacobian matrix $DF(x) \in \mathbb{R}^{(n-D_0) \times n}$ has full rank $(n - D_0)$ for all $x \in U \cap \mathcal{M}$.

  2. (Compactness) $\mathcal{M}$ is compact in the subspace topology induced from $\mathbb{R}^n$, which by the Heine-Borel theorem is equivalent to being closed and bounded in $\mathbb{R}^n$.

**Definition B.10** (Intrinsic Dimension). The *intrinsic dimension* $D_0 = d_{\text{int}}(\mathcal{M}_0)$ of a manifold $\mathcal{M}_0$ is the minimum number of parameters needed to locally parameterize the manifold. Formally, it is the dimension of the tangent space $T_p\mathcal{M}_0$ at any point $p \in \mathcal{M}_0$, which is constant for smooth connected manifolds.

*Remark* B.11. For a more basic definition of manifold, please refer to introductory mathematics textbook (Lee, 2011). In our subsequent discussion of prismatic space, we generalize the concept of intrinsic dimension. Since prismatic space lacks the well-behaved mathematical properties of smooth manifold, we define the intrinsic dimension as the maximum of the dimensions at all locally smooth points of the space.

### B.7. Proof of Proposition 3.7

*Proof.* We proceed by leveraging the definitions provided and establishing the piecewise linearity of the composite map $\Phi^{(\ell)}$, then analyzing its image on the input manifold $\mathcal{M}_0$.

By Proposition 3.4, each layer map $F^{(\ell)} : \mathbb{H}^{(\ell-1)} \to \mathbb{H}^{(\ell)}$ is a piecewise linear function. This follows from the assumptions that the operators $\mathfrak{A}^{(\ell)}$, $\mathfrak{M}^{(\ell)}$, and $\mathfrak{U}^{(\ell)}$ are piecewise linear and almost everywhere differentiable, and that $\mathfrak{U}^{(\ell)}$ uses piecewise linear activations. Since the composition of piecewise linear functions is piecewise linear (Lemma B.7), the composite map $\Phi^{(\ell)} = F^{(\ell)} \circ \cdots \circ F^{(1)}$ is also piecewise linear. Formally, there exists a finite set of polyhedral regions $\{R_i\}_{i=1}^{K}$ covering the domain of $\Phi^{(\ell)}$ such that for each $i$, the restriction of $\Phi^{(\ell)}$ to $R_i$ is affine:

$$\Phi^{(\ell)}(\boldsymbol{H}) = \text{unvec}(\boldsymbol{A}_i \cdot \text{vec}(\boldsymbol{H}) + \boldsymbol{b}_i) \quad \text{for all } \boldsymbol{H} \in R_i, \tag{46}$$

where $\boldsymbol{A}_i \in \mathbb{R}^{Nd_\ell \times Nd_0}$ and $\boldsymbol{b}_i \in \mathbb{R}^{Nd_\ell}$ are constants specific to region $R_i$.

The input manifold $\mathcal{M}_0 \subset \mathbb{R}^{N \times d_0}$ is compact and smooth by Definition 3.5. Consider the intersection of $\mathcal{M}_0$ with the polyhedral regions $R_i$:

$$\mathcal{M}_0^{(i)} = \mathcal{M}_0 \cap R_i. \tag{47}$$

Since $\mathcal{M}_0$ is a smooth manifold and each $R_i$ is polyhedral, the sets $\mathcal{M}_0^{(i)}$ are submanifolds with boundaries (possibly with corners). The collection $\{\mathcal{M}_0^{(i)}\}_{i=1}^{K}$ forms a finite cover of $\mathcal{M}_0$.

On each $\mathcal{M}_0^{(i)}$, the map $\Phi^{(\ell)}$ is affine. Therefore, the image $\Phi^{(\ell)}(\mathcal{M}_0^{(i)})$ is an affine transformation of $\mathcal{M}_0^{(i)}$:

$$\Phi^{(\ell)}(\mathcal{M}_0^{(i)}) = \{\text{unvec}(\boldsymbol{A}_i \cdot \text{vec}(\boldsymbol{H}) + \boldsymbol{b}_i) \mid \boldsymbol{H} \in \mathcal{M}_0^{(i)}\}. \tag{48}$$

Assuming that $\Phi^{(\ell)}$ is injective on each $R_i$, it is also injective on each $\mathcal{M}_0^{(i)}$. Since affine maps preserve linear structures and injectivity ensures that the map is an embedding on each piece, $\Phi^{(\ell)}(\mathcal{M}_0^{(i)})$ is itself a submanifold with boundary (possibly with corners) in $\mathbb{R}^{N \times d_\ell}$.

The full representation space is the union of these images:

$$\mathcal{M}^{(\ell)} = \bigcup_{i=1}^{K} \Phi^{(\ell)}(\mathcal{M}_0^{(i)}). \tag{49}$$

Such a union is termed a prismatic space.

Singularities occur at the boundaries between the regions. Specifically:

- The boundaries between different $\mathcal{M}_0^{(i)}$ correspond to points where $\Phi^{(\ell)}$ transitions from one affine piece to another.

- At these boundaries, the Jacobian of $\Phi^{(\ell)}$ may be discontinuous or undefined, leading to non-smooth points in $\mathcal{M}^{(\ell)}$.

- Since $\mathcal{M}_0$ is compact and smooth, it generically intersects multiple regions $R_i$, making such singularities typical. For example, if $\mathcal{M}_0$ is transversal to the boundaries of $R_i$, the intersections will be lower-dimensional manifolds where the image under $\Phi^{(\ell)}$ may not be smooth.

Thus, $\mathcal{M}^{(\ell)}$ is a prismatic space and may have singularities along the boundaries of the pieces $\Phi^{(\ell)}(\mathcal{M}_0^{(i)})$. $\qquad\square$

*Remark* B.12. The prismatic space we define constitutes a geometric structure more complex than a conventional topological manifold. While its interior may largely exhibit the properties of a smooth manifold, its boundary can contain intricate corners or even singularities. As a result, it is highly unlikely that the prismatic space satisfies the standard definitions of a topological manifold. It should be emphasized that constructing a rigorous topological definition of this geometric structure is highly challenging. Therefore, within the framework of this paper, we adopt a simplified definition grounded in piecewise linear map.

### B.8. Details of Definition 3.8

*Remark* B.13. The prismatic effect of different singular values on space:

- A singular value $\sigma_i^{(\ell)} \approx 1$ represents an unrefracted dimension, typically corresponding to node features preserved through linear identity paths or attention mechanisms that remain active.

- A singular value $0 < \sigma_i^{(\ell)} < 1$ represents a contracted dimension, potentially arising from the scaling of weight matrices ($\|W^{(\ell)}\| < 1$) and the gradient attenuation of activation functions like ReLU/LeakyReLU in their unsaturated regimes.

- A singular value $\sigma_i^{(\ell)} = 0$ represents a nullified dimension, resulting directly from the sparsity induced by ReLU activations which reduces the rank of the layer's Jacobian.

- A singular value $\sigma_i^{(\ell)} > 1$ represents an expanded dimension, potentially arising from feature amplification in weight matrices ($\|W^{(\ell)}\| > 1$) or certain graph convolution operations.

### B.9. Proof of Theorem 3.9

*Proof.* Since $F^{(\ell)}$ is linear on $\mathbb{S}$, there exists a matrix $A^{(\ell)} \in \mathbb{R}^{N d_\ell \times N d_{\ell-1}}$ and a vector $b^{(\ell)}$ such that for all $X \in \mathbb{S}$:

$$F^{(\ell)}(X) = \text{unvec}(A^{(\ell)} \cdot \text{vec}(X) + b^{(\ell)}). \tag{50}$$

The Jacobian $J^{(\ell)}$ is constant and equal to $A^{(\ell)}$. By assumption, $A^{(\ell)}$ has rank $r_\ell$, and its singular value decomposition is:

$$A^{(\ell)} = U^{(\ell)} \Sigma^{(\ell)} V^{(\ell)\top}, \tag{51}$$

where $U^{(\ell)}$ and $V^{(\ell)}$ are orthogonal matrices, and $\Sigma^{(\ell)} = \text{diag}(\sigma_1^{(\ell)}, \ldots, \sigma_{r_\ell}^{(\ell)}, 0, \ldots, 0)$ with $\sigma_1^{(\ell)} \geq \sigma_2^{(\ell)} \geq \cdots \geq \sigma_{r_\ell}^{(\ell)} > 0$.

Let $\boldsymbol{V}_s^{(\ell)}$ be the first $s$ columns of $\boldsymbol{V}^{(\ell)}$, spanning the subspace $\mathbb{V}^{(\ell)}$. The restriction of $\boldsymbol{A}^{(\ell)}$ to $\mathbb{V}^{(\ell)}$ is the linear map $L^{(\ell)} : \mathbb{V}^{(\ell)} \to \mathbb{R}^M$ defined by $L^{(\ell)}(\boldsymbol{x}) = \boldsymbol{A}^{(\ell)}\boldsymbol{x}$.

Since $\boldsymbol{A}^{(\ell)}$ is injective on $\mathbb{V}^{(\ell)}$ (as $\mathbb{V}^{(\ell)}$ is spanned by right singular vectors corresponding to positive singular values), $L^{(\ell)}$ is injective. The image $L^{(\ell)}(\mathbb{V}^{(\ell)})$ is an $s$-dimensional subspace of $\mathbb{R}^M$, spanned by the first $s$ columns of $\boldsymbol{U}^{(\ell)}$.

Let $\{\boldsymbol{v}_1^{(\ell)}, \ldots, \boldsymbol{v}_s^{(\ell)}\}$ be an orthonormal basis for $\mathbb{V}^{(\ell)}$ (e.g., the columns of $\boldsymbol{V}_s^{(\ell)}$). Then $\{L^{(\ell)}(\boldsymbol{v}_1^{(\ell)}), \ldots, L^{(\ell)}(\boldsymbol{v}_s^{(\ell)})\}$ is a basis for $L^{(\ell)}(\mathbb{V}^{(\ell)})$, and:

$$L^{(\ell)}(\boldsymbol{v}_i^{(\ell)}) = \sigma_i^{(\ell)}\boldsymbol{u}_i^{(\ell)}, \tag{52}$$

where $\boldsymbol{u}_i^{(\ell)}$ is the $i$-th column of $\boldsymbol{U}^{(\ell)}$. Thus, $\{\boldsymbol{u}_1^{(\ell)}, \ldots, \boldsymbol{u}_s^{(\ell)}\}$ is an orthonormal basis for $L^{(\ell)}(\mathbb{V}^{(\ell)})$.

The $s$-dimensional Hausdorff measure $\mathcal{H}^s$ is equivalent to the $s$-dimensional Lebesgue measure on $s$-dimensional subspaces. Consider the linear map $L^{(\ell)} : \mathbb{V}^{(\ell)} \to L^{(\ell)}(\mathbb{V}^{(\ell)})$. Since $\mathbb{V}^{(\ell)}$ and $L^{(\ell)}(\mathbb{V}^{(\ell)})$ are $s$-dimensional Euclidean spaces, we can compute the change in measure using the determinant of $L^{(\ell)}$ (in orthonormal coordinates).

Let $\boldsymbol{x} \in \mathbb{V}^{(\ell)}$ have coordinates $\boldsymbol{x} = \sum_{i=1}^s x_i\boldsymbol{v}_i^{(\ell)}$. Then:

$$L^{(\ell)}(\boldsymbol{x}) = \sum_{i=1}^s x_i L^{(\ell)}(\boldsymbol{v}_i) = \sum_{i=1}^s x_i \sigma_i^{(\ell)}\boldsymbol{u}_i^{(\ell)}. \tag{53}$$

Thus, the matrix representation of $L^{(\ell)}$ with respect to the bases $\boldsymbol{v}_i^{(\ell)}$ and $\boldsymbol{u}_i^{(\ell)}$ is the diagonal matrix $\mathrm{diag}(\sigma_1^{(\ell)}, \ldots, \sigma_s^{(\ell)})$.

The absolute determinant of this matrix is $\prod_{i=1}^s \sigma_i^{(\ell)}$. Therefore, for any measurable set $\mathbb{S} \subset \mathbb{V}^{(\ell)}$:

$$\mathcal{H}^s(L^{(\ell)}(\mathbb{S})) = \left(\prod_{i=1}^s \sigma_i^{(\ell)}\right)\mathcal{H}^s(\mathbb{S}). \tag{54}$$

Since $F^{(\ell)}(\boldsymbol{X}) = L^{(\ell)}(\boldsymbol{X}) + \boldsymbol{b}^{(\ell)}$ and translation preserves Hausdorff measure, we have:

$$\mathcal{H}^s(F^{(\ell)}(\mathbb{S})) = \mathcal{H}^s(L^{(\ell)}(\mathbb{S}) + \boldsymbol{b}^{(\ell)}) = \mathcal{H}^s(L^{(\ell)}(\mathbb{S})) = \left(\prod_{i=1}^s \sigma_i^{(\ell)}\right)\mathcal{H}^s(\mathbb{S}). \tag{55}$$

When $s = r_\ell$, $\mathbb{V}^{(\ell)}$ is the entire row space of $\boldsymbol{A}^{(\ell)}$, and the product is over all positive singular values. This gives the volume contraction factor for the full rank part of the map. $\qquad\square$

*Remark* B.14. We will not elaborate on mathematical concepts such as Hausdorff measure and Lebesgue measure in this article. For details, please refer to mathematics textbook (Krantz & Parks, 2008).

### B.10. Simple Linear Algebra

**Lemma B.15** (The Rank Inequality for Composition of Linear Maps). *Let $A : \mathbb{V} \to \mathbb{W}$ and $B : \mathbb{W} \to \mathbb{U}$ be linear maps between vector spaces. The composition $B \circ A : \mathbb{V} \to \mathbb{U}$ is also a linear map. The rank of a linear map is defined as the dimension of its image:*

$$rank(A) = \dim(im(A)), \quad rank(B) = \dim(im(B)), \quad rank(B \circ A) = \dim(im(B \circ A)). \tag{56}$$

*Then:*

$$rank(B \circ A) \leq \min(rank(A), rank(B)). \tag{57}$$

*Proof.* Prove the first inequality: $\mathrm{rank}(B \circ A) \leq \mathrm{rank}(A)$.

Observe that for any $\boldsymbol{v} \in \mathbb{V}$,

$$(B \circ A)(\boldsymbol{v}) = B(A(\boldsymbol{v})), \tag{58}$$

so the image of $B \circ A$ is:

$$im(B \circ A) = \{B(A(\boldsymbol{v})) : \boldsymbol{v} \in \mathbb{V}\} = B(\{A(\boldsymbol{v}) : \boldsymbol{v} \in \mathbb{V}\}) = B(im(A)). \tag{59}$$

Thus, $\text{im}(B \circ A) = B(\text{im}(A))$. Since $\text{im}(A) \subseteq \mathbb{W}$, we can restrict $B$ to $\text{im}(A)$, obtaining a linear map:

$$B|_{\text{im}(A)} : \text{im}(A) \to \mathbb{U}. \tag{60}$$

The image of this restricted map is exactly $B(\text{im}(A)) = \text{im}(B \circ A)$. By the Rank-Nullity Theorem (or simply by the fact that the image of a linear map cannot exceed the dimension of its domain), we have:

$$\dim(B(\text{im}(A))) \leq \dim(\text{im}(A)). \tag{61}$$

Therefore,

$$\text{rank}(B \circ A) = \dim(\text{im}(B \circ A)) \leq \dim(\text{im}(A)) = \text{rank}(A). \tag{62}$$

Prove the second inequality: $\text{rank}(B \circ A) \leq \text{rank}(B)$.

We now show that $\text{im}(B \circ A) \subseteq \text{im}(B)$. Let $\boldsymbol{u} \in \text{im}(B \circ A)$. Then there exists $\boldsymbol{v} \in \mathbb{V}$ such that:

$$\boldsymbol{u} = (B \circ A)(\boldsymbol{v}) = B(A(\boldsymbol{v})). \tag{63}$$

Since $A(\boldsymbol{v}) \in \mathbb{W}$, it follows that $\boldsymbol{u} = B(\boldsymbol{w})$ for some $\boldsymbol{w} \in \mathbb{W}$, so $\boldsymbol{u} \in \text{im}(B)$. Hence,

$$\text{im}(B \circ A) \subseteq \text{im}(B), \tag{64}$$

and therefore:

$$\dim(\text{im}(B \circ A)) \leq \dim(\text{im}(B)) \quad \Rightarrow \quad \text{rank}(B \circ A) \leq \text{rank}(B). \tag{65}$$

Combining both inequalities (62) and (65), we conclude:

$$\text{rank}(B \circ A) \leq \min(\text{rank}(A), \text{rank}(B)). \tag{66}$$

$\square$

## B.11. Proof of Theorem 3.12

*Proof.* From Proposition 3.4, each layer map $F^{(\ell)}$ is piecewise linear and differentiable almost everywhere. By Proposition 3.7, the composite map $\Phi = F^{(L)} \circ \cdots \circ F^{(1)}$ is also piecewise linear and $\mathcal{M}^{(L)}$ is a prismatic space. On each linear region $C_k$ (as defined in Definition 3.11), $\Phi$ is linear, so its rank is constant on $C_k$. Thus, $\Phi$ is piecewise constant on its rank.

Let $\{C_k\}$ be the linear region partition of $\mathcal{M}_0$ from Definition 3.11. For each $C_k$, the map $\Phi|_{C_k}$ is linear. Let $T_k = \Phi|_{C_k}$ denote this linear map. The image $\Phi(C_k)$ is contained in a linear subspace of dimension $\text{rank}(T_k)$.

The local dimension of $\mathcal{M}^{(L)}$ at any point in $\Phi(C_k)$ is at most $\text{rank}(T_k)$. Since $\mathcal{M}^{(L)} = \bigcup_k \Phi(C_k)$, the intrinsic dimension $d_{\text{int}}(\mathcal{M}^{(L)})$ is the supremum of the local dimensions over all points in $\mathcal{M}^{(L)}$. Thus,

$$d_{\text{int}}(\mathcal{M}^{(L)}) \leq \max_k \text{rank}(T_k). \tag{67}$$

Now, we bound $\text{rank}(T_k)$. Since $T_k = F^{(L)} \circ \cdots \circ F^{(1)}|_{C_k}$, and each $F^{(\ell)}$ is linear on the relevant region, we have:

$$\text{rank}(T_k) \leq \min_\ell \text{rank}\left(F^{(\ell)}|_{\Phi^{(\ell-1)}(C_k)}\right). \tag{68}$$

This follows from Lemma B.15: for linear maps $A$ and $B$, $\text{rank}(B \circ A) \leq \min(\text{rank}(A), \text{rank}(B))$. By induction, this holds for the composition of $L$ linear maps.

For each layer $\ell$, $\text{rank}\left(F^{(\ell)}|_{\Phi^{(\ell-1)}(C_k)}\right) = \text{rank}\left(\boldsymbol{J}^{(\ell)}|_{\Phi^{(\ell-1)}(C_k)}\right)$ because the Jacobian is constant on the region where $F^{(\ell)}$ is linear (from Definition 3.11).

Let $r_{\ell,k} = \text{rank}\left(\boldsymbol{J}^{(\ell)}|_{\Phi^{(\ell-1)}(C_k)}\right)$. Then,

$$\text{rank}(T_k) \leq \min_\ell r_{\ell,k}. \tag{69}$$

Therefore,

$$d_{\text{int}}(\mathcal{M}^{(L)}) \leq \max_k \text{rank}(T_k) \leq \max_k \min_\ell r_{\ell,k}. \tag{70}$$

Due to the contraction effect of the layers (especially with ReLUs, which project dimensions to zero), the ranks $r_{\ell,k}$ are often much smaller than the input dimension $D_0$. Thus, $\max_k \min_\ell r_{\ell,k}$ is typically less than $D_0$, implying that $\mathcal{M}^{(L)}$ has a lower intrinsic dimension than $D_0$. $\square$

### B.12. Proof of Theorem 3.13

*Proof.* By Proposition 3.4, each layer map $F^{(\ell)}$ is piecewise linear and differentiable almost everywhere. By Proposition 3.7, the composite map $\Phi = F^{(L)} \circ \cdots \circ F^{(1)}$ is piecewise linear. By Definition 3.11, the input manifold $\mathcal{M}_0$ is partitioned into cells $C_k$ such that on each $C_k$, $\Phi$ is linear.

We assume $\Phi$ is injective on $C_k$. This implies that for each $C_k$, $\Phi$ restricted to $C_k$ is a linear injection, so $d_{\text{int}}(\Phi(C_k)) = d_{\text{int}}(C_k) = d_{\text{int}}$, where $d_{\text{int}} = D_0$ is the intrinsic dimension of $\mathcal{M}_0$ (Definition 3.5). Since $\Phi$ is injective, each layer $F^{(\ell)}$ must be injective on $\Phi^{(\ell-1)}(C_k)$ for all $\ell$ and $k$. Otherwise, the composition would not be injective. Thus, for each $\ell$ and $k$, the Jacobian $\boldsymbol{J}^{(\ell)}$ of $F^{(\ell)}$ restricted to the tangent space of $\Phi^{(\ell-1)}(C_k)$ has rank at least $d_{\text{int}}$. Since the tangent space is $d_{\text{int}}$-dimensional, $\boldsymbol{J}^{(\ell)}$ has exactly $d_{\text{int}}$ positive singular values $\sigma_{1,k}^{(\ell)} \geq \sigma_{2,k}^{(\ell)} \geq \cdots \geq \sigma_{d_{\text{int}},k}^{(\ell)} > 0$ on the region corresponding to $C_k$.

Consider a fixed cell $C_k$. Since $\Phi$ is linear on $C_k$, we can write $\Phi(\boldsymbol{X}) = \text{unvec}(\boldsymbol{J}_k \cdot \text{vec}(\boldsymbol{X}) + \boldsymbol{b}_k)$ for $\boldsymbol{X} \in C_k$, where $\boldsymbol{J}_k$ is the Jacobian of $\Phi$ on $C_k$ (constant). However, to understand the layer-wise measure change, we use the composition structure.

For the first layer $F^{(1)}$, since it is linear on $C_k$, it maps $C_k$ to $F^{(1)}(C_k)$. By Theorem 3.9, the $d_{\text{int}}$-dimensional Hausdorff measure changes as:

$$\mathcal{H}^{d_{\text{int}}}(F^{(1)}(C_k)) = \Big(\prod_{i=1}^{d_{\text{int}}} \sigma_{i,k}^{(1)}\Big) \mathcal{H}^{d_{\text{int}}}(C_k), \tag{71}$$

where $\sigma_{i,k}^{(1)}$ are the singular values of $\boldsymbol{J}^{(1)}$ restricted to the tangent space of $C_k$ (which is $d_{\text{int}}$-dimensional).

For the second layer $F^{(2)}$, it is linear on $F^{(1)}(C_k)$ (which is $d_{\text{int}}$-dimensional). It maps $F^{(1)}(C_k)$ to $F^{(2)}(F^{(1)}(C_k))$. Again, by Theorem 3.9:

$$\mathcal{H}^{d_{\text{int}}}(F^{(2)}(F^{(1)}(C_k))) = \Big(\prod_{i=1}^{d_{\text{int}}} \sigma_{i,k}^{(2)}\Big) \mathcal{H}^{d_{\text{int}}}(F^{(1)}(C_k)) = \Big(\prod_{i=1}^{d_{\text{int}}} \sigma_{i,k}^{(2)}\Big)\Big(\prod_{i=1}^{d_{\text{int}}} \sigma_{i,k}^{(1)}\Big) \mathcal{H}^{d_{\text{int}}}(C_k), \tag{72}$$

where $\sigma_{i,k}^{(2)}$ are the singular values of $\boldsymbol{J}^{(2)}$ restricted to the tangent space of $F^{(1)}(C_k)$.

Proceeding inductively for all $L$ layers, we get:

$$\mathcal{H}^{d_{\text{int}}}(\Phi(C_k)) = \Big(\prod_{\ell=1}^{L}\prod_{i=1}^{d_{\text{int}}} \sigma_{i,k}^{(\ell)}\Big) \mathcal{H}^{d_{\text{int}}}(C_k). \tag{73}$$

This is because each layer's measure change factor is multiplicative, and the composition preserves the $d_{\text{int}}$-dimensional measure up to the product of the singular values.

Since the cells $C_k$ form a partition of $\mathcal{M}_0$ (Definition 3.11), and $\Phi$ is injective on $C_k$, the images $\Phi(C_k)$ are disjoint and cover $\mathcal{M}^{(L)}$ (up to sets of measure zero, due to piecewise linearity). Therefore, by the additivity of the Hausdorff measure:

$$\mathcal{H}^{d_{\text{int}}}(\mathcal{M}^{(L)}) = \sum_k \mathcal{H}^{d_{\text{int}}}(\Phi(C_k)) = \sum_k \Big(\prod_{\ell=1}^{L}\prod_{i=1}^{d_{\text{int}}} \sigma_{i,k}^{(\ell)}\Big) \mathcal{H}^{d_{\text{int}}}(C_k). \tag{74}$$

This establishes the desired formula.

If $\Phi$ is not injective, then the images $\Phi(C_k)$ may overlap. Since the Hausdorff measure is subadditive, we have:

$$\mathcal{H}^{d_{\text{int}}}(\mathcal{M}^{(L)}) \leq \sum_k \mathcal{H}^{d_{\text{int}}}(\Phi(C_k)) = \sum_k \Big(\prod_{\ell=1}^{L}\prod_{i=1}^{d_{\text{int}}} \sigma_{i,k}^{(\ell)}\Big) \mathcal{H}^{d_{\text{int}}}(C_k). \tag{75}$$

Thus, the formula provides an upper bound. $\square$

## B.13. Details of Definition 3.14

*Remark* B.16. This definition formalizes the notion of how a prompt $\boldsymbol{P}$ modifies the input data manifold in the context of graph prompt tuning. The original input manifold $\mathcal{M}_0$, which represents the natural data distribution (e.g., graph node features), is typically assumed to be a compact smooth manifold embedded in $\mathbb{R}^{N \times d_0}$. The prompt $\boldsymbol{P}$ is a low-dimensional perturbation applied to every point in $\mathcal{M}_0$, resulting in a new manifold $\mathcal{M}_0(\boldsymbol{P})$. The operation $\mathcal{M}_0(\boldsymbol{P}) = \{\boldsymbol{X} + \boldsymbol{P} \mid \boldsymbol{X} \in \mathcal{M}_0\}$ is a translation of the entire manifold by $\boldsymbol{P}$, which preserves the topological and geometric properties of $\mathcal{M}_0$, such as compactness and smoothness, since translation is a diffeomorphism. The prompt space $\mathcal{P}$ is the set of all possible prompts, often constrained to be low-dimensional (e.g., a subspace of $\mathbb{R}^{N \times d_0}$), and each prompt $\boldsymbol{P} \in \mathcal{P}$ defines a distinct perturbed manifold. This family of manifolds $\{\mathcal{M}_0(\boldsymbol{P}) \mid \boldsymbol{P} \in \mathcal{P}\}$ encapsulates the variability introduced by graph prompt tuning, and the goal is to understand how the GNN-based Graph Foundation Models (GFMs) transform these manifolds through its layers.

## B.14. Lipschitz Continuous and Jacobian

**Lemma B.17** (Continuity of the Layer Map $F^{(\ell)}$). *Assume the operators $\mathfrak{A}^{(\ell)}$, $\mathfrak{M}^{(\ell)}$, and $\mathfrak{U}^{(\ell)}$ defining the GFM layer in Definition 3.1 are continuous. Then, the layer map $F^{(\ell)}$ is continuous.*

*Proof.* Similar to the proof of Proposition 3.4, let $f_1 = \mathfrak{A}^{(\ell)}$, $f_2 = \mathfrak{M}^{(\ell)}$, and $f_3 = \mathfrak{U}^{(\ell)}$. Then $F^{(\ell)} = f_3 \circ (f_2 \circ (f_1, \mathrm{id}), \mathrm{id})$, where id denotes the identity function Since the composition of continuous functions is continuous, the overall layer map $F^{(\ell)}$ is continuous. $\square$

**Lemma B.18** (Lipschitz Continuity of the GFM Map $\Phi$). *Let $\mathcal{M}_0(\boldsymbol{P}) \subset \mathbb{R}^{N \times d_0}$ be the compact prompt-perturbed input manifold as defined in Definition 3.14. The composite map $\Phi = F^{(L)} \circ F^{(L-1)} \circ \cdots \circ F^{(1)}$, where each $F^{(\ell)}$ is a piecewise linear layer map (Proposition 3.4), is Lipschitz continuous on $\mathcal{M}_0(\boldsymbol{P})$. That is, there exists a constant $L_\Phi < \infty$ such that for all $\boldsymbol{X}, \boldsymbol{Y} \in \mathcal{M}_0(\boldsymbol{P})$,*

$$\|\Phi(\boldsymbol{X}) - \Phi(\boldsymbol{Y})\| \leq L_\Phi \|\boldsymbol{X} - \boldsymbol{Y}\|. \tag{76}$$

*Moreover, the Lipschitz constant $L_\Phi$ satisfies:*

$$L_\Phi \leq \prod_{\ell=1}^{L} L_\ell, \tag{77}$$

*where $L_\ell$ is the Lipschitz constant of the $\ell$-th layer $F^{(\ell)}$ on the appropriate domain.*

*Proof.* Prove the piecewise linear layers are Lipschitz continuous.

Each layer map $F^{(\ell)} : \mathbb{R}^{N \times d_{\ell-1}} \to \mathbb{R}^{N \times d_\ell}$ is piecewise linear and continuous by Proposition 3.4 and Lemma B.17. Since $\mathcal{M}_0(\boldsymbol{P})$ is compact and each $F^{(\ell)}$ is continuous, the image $F^{(\ell)}(\mathcal{M}_0(\boldsymbol{P}))$ is also compact. The piecewise linearity implies that there exists a finite partition of the domain into polyhedral regions $R_k^{(\ell)}$ such that $F^{(\ell)}$ is linear on each region $R_k^{(\ell)}$. On each such region, for any $\boldsymbol{H}, \boldsymbol{H}' \in R_k^{(\ell)}$, we have

$$\|F^{(\ell)}(\boldsymbol{H}) - F^{(\ell)}(\boldsymbol{H}')\| = \|\boldsymbol{A}_k^{(\ell)}(\boldsymbol{H} - \boldsymbol{H}')\| \leq \|\boldsymbol{A}_k^{(\ell)}\|_{\mathrm{op}} \|\boldsymbol{H} - \boldsymbol{H}'\|, \tag{78}$$

where $\boldsymbol{A}_k^{(\ell)}$ is the matrix representing the linear map on $R_k^{(\ell)}$ and $|\cdot|_{\mathrm{op}}$ denotes the operator norm (spectral norm). Define the local Lipschitz constant for $F^{(\ell)}$ on region $R_k^{(\ell)}$ as $L_k^{(\ell)} = |\boldsymbol{A}_k^{(\ell)}|_{\mathrm{op}}$. Since the number of regions intersecting the compact set $\mathcal{M}_0(\boldsymbol{P})$ is finite, the global Lipschitz constant for $F^{(\ell)}$ on $\mathcal{M}_0(\boldsymbol{P})$ is finite and given by

$$L_\ell = \max_k L_k^{(\ell)} < \infty. \tag{79}$$

Thus, for any $\boldsymbol{H}, \boldsymbol{H}' \in \mathcal{M}_0(\boldsymbol{P})$,

$$\|F^{(\ell)}(\boldsymbol{H}) - F^{(\ell)}(\boldsymbol{H}')\| \leq L_\ell \|\boldsymbol{H} - \boldsymbol{H}'\|. \tag{80}$$

Prove the composite map $\Phi$ is Lipschitz continuous.

The composite map $\Phi = F^{(L)} \circ F^{(L-1)} \circ \cdots \circ F^{(1)}$ is a composition of Lipschitz continuous maps. For any $\boldsymbol{X}, \boldsymbol{Y} \in \mathcal{M}_0(\boldsymbol{P})$, let $\boldsymbol{H}^{(\ell)} = F^{(\ell)} \circ \cdots \circ F^{(1)}(\boldsymbol{X})$ and $\boldsymbol{K}^{(\ell)} = F^{(\ell)} \circ \cdots \circ F^{(1)}(\boldsymbol{Y})$ denote the intermediate representations. Then,

$$
\begin{aligned}
\|\boldsymbol{H}^{(1)} - \boldsymbol{K}^{(1)}\| &= \|F^{(1)}(\boldsymbol{X}) - F^{(1)}(\boldsymbol{Y})\| \leq L_1 \|\boldsymbol{X} - \boldsymbol{Y}\|, \\
\|\boldsymbol{H}^{(2)} - \boldsymbol{K}^{(2)}\| &= \|F^{(2)}(\boldsymbol{H}^{(1)}) - F^{(2)}(\boldsymbol{K}^{(1)})\| \leq L_2 \|\boldsymbol{H}^{(1)} - \boldsymbol{K}^{(1)}\| \leq L_2 L_1 \|\boldsymbol{X} - \boldsymbol{Y}\|, \\
&\vdots \\
\|\boldsymbol{H}^{(L)} - \boldsymbol{K}^{(L)}\| &= \|\Phi(\boldsymbol{X}) - \Phi(\boldsymbol{Y})\| \leq L_L \|\boldsymbol{H}^{(L-1)} - \boldsymbol{K}^{(L-1)}\| \leq \Big(\prod_{\ell=1}^{L} L_\ell\Big) \|\boldsymbol{X} - \boldsymbol{Y}\|.
\end{aligned}
\tag{81}
$$

Therefore, $\Phi$ is Lipschitz continuous with constant $L_\Phi = \prod_{\ell=1}^{L} L_\ell$.

This completes the proof. $\qquad\square$

**Lemma B.19** (Existence of the Jacobian $\boldsymbol{J}_\Phi(\boldsymbol{X})$). *In the context of the unified GFM framework, we aim to prove that the Jacobian of the composite map $\Phi = F^{(L)} \circ F^{(L-1)} \circ \cdots \circ F^{(1)}$ exists almost everywhere (a.e.) on the input manifold $\mathcal{M}_0(\boldsymbol{P})$, and that at points where it exists, it is given by the product of the layer Jacobians.*

*Proof.* By Proposition 3.4, each layer map $F^{(\ell)} : \mathbb{R}^{N \times d_{\ell-1}} \to \mathbb{R}^{N \times d_\ell}$ is piecewise linear. This means that the domain of $F^{(\ell)}$ can be partitioned into a finite number of polyhedral regions $R_k^{(\ell)}$ such that $F^{(\ell)}$ is linear on each region. Since linear functions are differentiable everywhere, $F^{(\ell)}$ is differentiable on the interior of each region. The boundaries between regions have Lebesgue measure zero in $\mathbb{R}^{N \times d_{\ell-1}}$ (as they are subsets of lower-dimensional affine spaces). Therefore, $F^{(\ell)}$ is differentiable almost everywhere in its domain. Let $D_\ell$ denote the set of points where $F^{(\ell)}$ is differentiable; then $D_\ell$ has full measure (i.e., its complement has measure zero).

The composite map $\Phi$ is defined as $\Phi = F^{(L)} \circ F^{(L-1)} \circ \cdots \circ F^{(1)}$. Consider the sets where each $F^{(\ell)}$ is differentiable. Since each $F^{(\ell)}$ is differentiable a.e., the set of points where all $F^{(\ell)}$ are differentiable along the composition path is also of full measure. More formally, define:

- $E_1 = D_1$ (the set where $F^{(1)}$ is differentiable).

- For $\ell = 2$ to $L$, define $E_\ell = \{\boldsymbol{X} \in E_{\ell-1} : F^{(\ell)}\}$ is differentiable at $\Phi^{(\ell-1)}(\boldsymbol{X})$, where $\Phi^{(\ell-1)} = F^{(\ell-1)} \circ \cdots \circ F^{(1)}$.

Since $F^{(\ell)}$ is differentiable a.e., and $\Phi^{(\ell-1)}$ is continuous and piecewise linear (hence Lipschitz), it preserves sets of measure zero. Thus, by induction, each $E_\ell$ has full measure. Therefore, the set $E = E_L$ where all $F^{(\ell)}$ are differentiable at the appropriate points has full measure in $\mathcal{M}_0(\boldsymbol{P})$. For any $\boldsymbol{X} \in E$, the composite map $\Phi$ is differentiable at $\boldsymbol{X}$ by the chain rule.

At a point $\boldsymbol{X} \in E$, the chain rule applies. Let $\boldsymbol{H}^{(0)} = \boldsymbol{X}$, and for $\ell = 1$ to $L$, define $\boldsymbol{H}^{(\ell)} = F^{(\ell)}(\boldsymbol{H}^{(\ell-1)})$. Then, the Jacobian of $\Phi$ at $\boldsymbol{X}$ is given by:

$$
\boldsymbol{J}_\Phi(\boldsymbol{X}) = \boldsymbol{J}^{(L)}(\boldsymbol{H}^{(L-1)}) \cdot \boldsymbol{J}^{(L-1)}(\boldsymbol{H}^{(L-2)}) \cdots \boldsymbol{J}^{(1)}(\boldsymbol{X}),
\tag{82}
$$

where $\boldsymbol{J}^{(\ell)}(\boldsymbol{H}^{(\ell-1)})$ is the Jacobian of $F^{(\ell)}$ at $\boldsymbol{H}^{(\ell-1)}$. This product is well-defined because each Jacobian exists at the respective points.

Since $\mathcal{M}_0(\boldsymbol{P})$ is a compact smooth manifold embedded in $\mathbb{R}^{N \times d_0}$, it has a Lipschitz parameterization. The above argument holds for almost every point in $\mathcal{M}_0(\boldsymbol{P})$ with respect to the Lebesgue measure on the parameter space. Thus, $\boldsymbol{J}_\Phi(\boldsymbol{X})$ exists for almost every $\boldsymbol{X} \in \mathcal{M}_0(\boldsymbol{P})$. $\qquad\square$

## B.15. Proof of Theorem 3.15

*Proof.* Assume the input manifold $\mathcal{M}_0(\boldsymbol{P})$ is compact and smooth with intrinsic dimension $d_{\text{int}}$. By Definition 3.11, the piecewise linear map $\Phi = F^{(L)} \circ \cdots \circ F^{(1)}$ partitions $\mathcal{M}_0(\boldsymbol{P})$ into a countable collection of cells $\{C_k'\}$, where each $C_k'$ is a connected subset of $\mathcal{M}_0(\boldsymbol{P})$ such that $\Phi$ is linear on $C_k'$. This partition exists because $\Phi$ is piecewise linear (Theorem 3.12).

**Proof of the Measure Bound.**

For each cell $C'_k$, since $\Phi$ is linear on $C'_k$, the Jacobian $J_\Phi$ is constant on $C'_k$. By Theorem 3.13, the Hausdorff measure of the image $\Phi(C'_k)$ is given by:

$$\mathcal{H}^{d_{\text{int}}}(\Phi(C'_k)) = \Big(\prod_{\ell=1}^{L}\prod_{i=1}^{d_{\text{int}}} \sigma_{i,k}^{(\ell)}\Big)\mathcal{H}^{d_{\text{int}}}(C'_k), \tag{83}$$

where $\sigma_{i,k}^{(\ell)}$ are the first $d_{\text{int}}$ singular values of the Jacobian of the $\ell$-th layer evaluated in the linear region corresponding to $C'_k$. Note that the product $\prod_{i=1}^{d_{\text{int}}} \sigma_{i,k}^{(\ell)}$ is taken over the largest $d_{\text{int}}$ singular values, as the tangent space has dimension $d_{\text{int}}$.

The total measure of $\mathcal{M}^{(L)}(\boldsymbol{P})$ is the sum over all cells:

$$\mathcal{H}^{d_{\text{int}}}(\mathcal{M}^{(L)}(\boldsymbol{P})) \le \sum_k \mathcal{H}^{d_{\text{int}}}(\Phi(C'_k)) = \sum_k \Big(\prod_{\ell=1}^{L}\prod_{i=1}^{d_{\text{int}}} \sigma_{i,k}^{(\ell)}\Big)\mathcal{H}^{d_{\text{int}}}(C'_k). \tag{84}$$

Since $\prod_{\ell=1}^{L}\prod_{i=1}^{d_{\text{int}}} \sigma_{i,k}^{(\ell)} \le \sup_{k'} \prod_{\ell=1}^{L}\prod_{i=1}^{d_{\text{int}}} \sigma_{i,k'}^{(\ell)}$ for all $k$, we have:

$$\mathcal{H}^{d_{\text{int}}}(\mathcal{M}^{(L)}(\boldsymbol{P})) \le \Big(\sup_{k'} \prod_{\ell=1}^{L}\prod_{i=1}^{d_{\text{int}}} \sigma_{i,k'}^{(\ell)}\Big)\sum_k \mathcal{H}^{d_{\text{int}}}(C'_k) = \Big(\sup_{k'} \prod_{\ell=1}^{L}\prod_{i=1}^{d_{\text{int}}} \sigma_{i,k'}^{(\ell)}\Big)\mathcal{H}^{d_{\text{int}}}(\mathcal{M}_0(\boldsymbol{P})). \tag{85}$$

This proves the measure bound.

**Proof of the Diameter Bound.**

Let $\text{diam}(\mathcal{M})$ denote the diameter of a set $\mathcal{M}$, defined as:

$$\text{diam}(\mathcal{M}) = \sup_{x,y\in\mathcal{M}} \|x - y\|. \tag{86}$$

By Theorem 3.12 and Lemma B.18, the map $\Phi$ is piecewise linear and Lipschitz continuous on $\mathcal{M}_0(\boldsymbol{P})$. The global Lipschitz constant $L_\Phi$ satisfies:

$$\|\Phi(\boldsymbol{X}) - \Phi(\boldsymbol{Y})\| \le L_\Phi\|\boldsymbol{X} - \boldsymbol{Y}\| \quad \forall \boldsymbol{X}, \boldsymbol{Y} \in \mathcal{M}_0(\boldsymbol{P}). \tag{87}$$

The Lipschitz constant $L_\Phi$ can be bounded by the operator norms of the Jacobians of $\Phi$. For any point $\boldsymbol{X} \in \mathcal{M}_0(\boldsymbol{P})$, the Jacobian $J_\Phi(\boldsymbol{X})$ exists almost everywhere (by Definition B.19) and is given by the product of the layer Jacobians:

$$\boldsymbol{J}_\Phi(\boldsymbol{X}) = \boldsymbol{J}^{(L)}(F^{(L-1)}(\boldsymbol{X}))\cdots\boldsymbol{J}^{(1)}(\boldsymbol{X}). \tag{88}$$

The operator norm of $J_\Phi(\boldsymbol{X})$ satisfies:

$$|\boldsymbol{J}_\Phi(\boldsymbol{X})|_{\text{op}} \le |\boldsymbol{J}^{(L)}(F^{(L-1)}(\boldsymbol{X}))|_{\text{op}}\cdots|\boldsymbol{J}^{(1)}(\boldsymbol{X})|_{\text{op}}. \tag{89}$$

Each layer Jacobian $|\boldsymbol{J}^{(\ell)}(\boldsymbol{X}_\ell)|_{\text{op}}$ (where $\boldsymbol{X}_\ell = F^{(\ell-1)}(\boldsymbol{X})$) is constant on linear regions. Let $|\boldsymbol{J}_k^{(\ell)}|_{\text{op}}$ be the operator norm of the Jacobian of the $\ell$-th layer in the $k$-th linear region. Then:

$$|\boldsymbol{J}^{(\ell)}(\boldsymbol{X}_\ell)|_{\text{op}} \le \sup_k |\boldsymbol{J}_k^{(\ell)}|_{\text{op}} \quad \forall \boldsymbol{X}_\ell. \tag{90}$$

Therefore,

$$|\boldsymbol{J}_\Phi(\boldsymbol{X})|_{\text{op}} \le \prod_{\ell=1}^{L} \sup_k |\boldsymbol{J}_k^{(\ell)}|_{\text{op}} \quad \forall \boldsymbol{X}. \tag{91}$$

The global Lipschitz constant $L_\Phi$ is the supremum of $|\boldsymbol{J}_\Phi(\boldsymbol{X})|_{\text{op}}$ over $\boldsymbol{X} \in \mathcal{M}_0(\boldsymbol{P})$:

$$L_\Phi = \sup_{\boldsymbol{X}\in\mathcal{M}_0(\boldsymbol{P})} |\boldsymbol{J}_\Phi(\boldsymbol{X})|_{\text{op}} \le \prod_{\ell=1}^{L} \sup_k |\boldsymbol{J}_k^{(\ell)}|_{\text{op}}. \tag{92}$$

Now, for any $\boldsymbol{X}, \boldsymbol{Y} \in \mathcal{M}_0(\boldsymbol{P})$,

$$\|\Phi(\boldsymbol{X}) - \Phi(\boldsymbol{Y})\| \leq L_\Phi \|\boldsymbol{X} - \boldsymbol{Y}\| \leq \Big( \prod_{\ell=1}^{L} \sup_k |\boldsymbol{J}_k^{(\ell)}|_{\text{op}} \Big) \|\boldsymbol{X} - \boldsymbol{Y}\|. \tag{93}$$

Taking the supremum over $\boldsymbol{X}, \boldsymbol{Y} \in \mathcal{M}_0(\boldsymbol{P})$, we get:

$$\text{diam}(\mathcal{M}^{(L)}(\boldsymbol{P})) \leq \Big( \prod_{\ell=1}^{L} \sup_k |\boldsymbol{J}_k^{(\ell)}|_{\text{op}} \Big) \cdot \text{diam}(\mathcal{M}_0(\boldsymbol{P})). \tag{94}$$

This proves the diameter bound. $\qquad\square$

### B.16. Theoretical Limitations of Graph Prompt Tuning

The Prompt Efficacy Bound (Theorem 3.15) reveals fundamental theoretical limitations of graph prompt tuning in GFMs. Specifically, the measure and diameter bounds imply that the influence of a prompt $\boldsymbol{P}$ is constrained by the compositional prismatic effect of the frozen GFM layers.

#### Information Loss through Spectral Contraction.

The measure bound shows that the effective *volume* of the prompt-perturbed space $\mathcal{M}^{(L)}(\boldsymbol{P})$ is scaled by the product of singular values across layers and linear regions. Since deep GFMs often exhibit spectral decay (with many singular values $\sigma_i^{(\ell)} \ll 1$), the prompt-induced perturbations are compressed exponentially with depth. This irreversible contraction implies that fine-grained semantic nuances introduced by the prompt may be lost or distorted before reaching the output layer.

#### Intrinsic Dimensionality Collapse.

As shown in Theorem 3.12, the intrinsic dimension $d_{\text{int}}(\mathcal{M}^{(L)})$ of the final representation is bounded by the minimal rank achieved locally across layers. Graph prompt tuning operates on the input manifold $\mathcal{M}_0(\boldsymbol{P})$, but the frozen network's piecewise linear transformations inherently project the prompt into a lower-dimensional subspace. Thus, even if the prompt is high-dimensional, its effective influence is limited by the bottleneck rank of the Jacobians, reducing its capacity to encode complex instructions.

#### Sensitivity to Input Geometry.

The diameter bound depends on the operator norms of the layer Jacobians. If the network exhibits gradient explosion (large $\sup_k |\boldsymbol{J}_k^{(\ell)}|_{\text{op}}$) or vanishing (small singular values), the prompt's effect may be either amplified erratically or suppressed. This sensitivity makes graph prompt tuning highly dependent on the pre-trained model's architecture and parameterization, limiting its robustness.

#### Non-Adaptive Prismatic Structure.

Since the network is frozen, the prompt cannot alter the prismatic folding process (e.g., the partition into linear regions or the Jacobian spectra). The prompt is merely a shift in the input space, and its efficacy depends on how the fixed geometric transformation $\Phi$ distorts this shift. In contrast, full fine-tuning adapts $\Phi$ itself to preserve task-relevant information, which graph prompt tuning cannot achieve.

#### Trade-off Between Prompt Size and Expressivity.

While increasing the prompt dimension $\dim(\mathcal{P})$ might seem beneficial, the measure bound shows that the effective output scale is constrained by the product of Jacobian singular values. Thus, simply enlarging the prompt may not improve efficacy if the network's contraction forces are too strong. This suggests a fundamental trade-off between prompt complexity and the network's capacity to preserve prompt-induced variations.

In summary, graph prompt tuning is inherently limited by the frozen GFM's spectral properties and geometric structure. While it can induce some distributional shifts, its ability to convey nuanced instructions is bounded by the network's pre-existing prismatic contraction and rank collapse. These limitations motivate the need for architectural interventions (e.g., adding adapters) or alternative tuning strategies that can mitigate the loss of prompt information through deeper layers.

# C. Theoretical Analysis of Message Tuning

## C.1. Proof of Theorem 4.1

*Proof.* We prove the theorem using the geometric measure theoretic framework of Prismatic Space Theory. The key idea is that message tuning, by injecting learnable parameters at each layer, can compensate for the measure contraction and intrinsic dimension reduction caused by the prismatic effect of the frozen GFM layers, and can additionally expand the diameter of the output space.

**Intrinsic Dimension Comparison.**

In Prismatic Space Theory, the intrinsic dimension $d_{\text{int}}(\mathcal{M}_{\text{MTG}}^{(L)})$ refers to the topological dimension or Hausdorff dimension of the final representation space $\mathcal{M}_{\text{MTG}}^{(L)}$, which is the inherent dimensionality of the space itself, not the dimension of the embedding space. This intrinsic dimension is defined by the geometric properties of space, but we can use the rank of the Jacobian matrix of the mapping to provide an upper bound.

Specifically, for the mapping $\Phi_{\text{MTG}} : \mathcal{M}_0 \to \mathcal{M}_{\text{MTG}}^{(L)}$ (where $\Phi_{\text{MTG}}$ is the composite layer mapping after message tuning), we have:

$$d_{\text{int}}(\mathcal{M}_{\text{MTG}}^{(L)}) \leq \sup_{\boldsymbol{X} \in \mathcal{M}_0} \text{rank}(\boldsymbol{J}_{\Phi_{\text{MTG}}}(\boldsymbol{X})), \tag{95}$$

where $\boldsymbol{J}_{\Phi_{\text{MTG}}}(\boldsymbol{X})$ is the Jacobian matrix of the mapping $\Phi_{\text{MTG}}$ at point $\boldsymbol{X}$. This means that the intrinsic dimension of the space cannot exceed the maximum rank of the Jacobian matrix across all input points.

From Theorem 3.12, for graph prompt tuning, the intrinsic dimension of the final space is bounded by:

$$d_{\text{int}}(\mathcal{M}_{\text{PT}}^{(L)}(\boldsymbol{P})) \leq \max_k \min_\ell \text{rank}(\boldsymbol{J}^{(\ell)}|_{\Phi^{(\ell-1)}(C_k)}), \tag{96}$$

where $\boldsymbol{J}^{(\ell)}$ is the Jacobian of the $\ell$-th layer of the frozen GFM, and $C_k$ are the linear regions of the input manifold.

For message tuning, the layer map is modified to include the fusion operation $\mathfrak{F}^{(\ell)}$. Specifically, at each layer $\ell$, the input representation $\boldsymbol{H}^{(\ell-1)}$ is transformed to $\boldsymbol{H}_M^{(\ell-1)} = \mathfrak{F}^{(\ell)}(\boldsymbol{H}^{(\ell-1)}, \boldsymbol{M}^{(\ell)}; \boldsymbol{\Theta}_f^{(\ell)})$ before applying the standard layer map $F^{(\ell)}$. Thus, the effective layer map becomes $\Psi^{(\ell)} = F^{(\ell)} \circ \mathfrak{F}^{(\ell)}$.

The Jacobian of $\Psi^{(\ell)}$ at a point where it is differentiable is given by the chain rule:

$$\boldsymbol{J}_\Psi^{(\ell)} = \boldsymbol{J}_F^{(\ell)} \cdot \boldsymbol{J}_{\mathfrak{F}}^{(\ell)}, \tag{97}$$

where $\boldsymbol{J}_F^{(\ell)}$ is the Jacobian of $F^{(\ell)}$ and $\boldsymbol{J}_{\mathfrak{F}}^{(\ell)}$ is the Jacobian of $\mathfrak{F}^{(\ell)}$.

The core issue is that $\mathfrak{F}^{(\ell)}$ is not linear, but we can show that with learnable parameters, its Jacobian can be made full-rank, ensuring the desired rank inequality.

Recall that for message tuning, the fusion operation is defined as:

$$\mathfrak{F}^{(\ell)}(\boldsymbol{H}^{(\ell-1)}, \boldsymbol{M}^{(\ell)}; \boldsymbol{\Theta}_f^{(\ell)}) = \boldsymbol{H}^{(\ell-1)} + \text{Softmax}(\boldsymbol{H}^{(\ell-1)} \boldsymbol{W}_p^{(\ell)}) \cdot \boldsymbol{M}^{(\ell)}, \tag{98}$$

where $\boldsymbol{H}^{(\ell-1)} \in \mathbb{R}^{N \times d_{\ell-1}}$, $\boldsymbol{W}_p^{(\ell)} \in \mathbb{R}^{d_{\ell-1} \times m}$, and $\boldsymbol{M}^{(\ell)} \in \mathbb{R}^{m \times d_{\ell-1}}$.

The Jacobian of $\mathfrak{F}^{(\ell)}$ with respect to $\boldsymbol{H}^{(\ell-1)}$ is a block-diagonal matrix composed of $N$ blocks, each of size $d_{\ell-1} \times d_{\ell-1}$. For each node $i$, the block corresponds to the derivative of the $i$-th row of $\mathfrak{F}^{(\ell)}$ with respect to the $i$-th row of $\boldsymbol{H}^{(\ell-1)}$. Specifically, let $\boldsymbol{h}_i$ be the $i$-th row of $\boldsymbol{H}^{(\ell-1)}$, and let $\boldsymbol{a}_i = \boldsymbol{h}_i \boldsymbol{W}_p^{(\ell)}$. Then the Softmax output is $\alpha_i = \text{Softmax}(\boldsymbol{a}_i)$, and the $i$-th row of $\mathfrak{F}^{(\ell)}$ is $\boldsymbol{h}_i + \alpha_i \boldsymbol{M}^{(\ell)}$.

The Jacobian for node $i$ is:

$$\boldsymbol{B}_i = \boldsymbol{I} + \boldsymbol{M}^{(\ell)\top} \boldsymbol{J}_{\text{softmax}}(\boldsymbol{a}_i) \boldsymbol{W}_p^{(\ell)\top}, \tag{99}$$

where $\boldsymbol{I}$ is the identity matrix, and $\boldsymbol{J}_{\text{softmax}}(\boldsymbol{a}_i) \in \mathbb{R}^{m \times m}$ is the Jacobian of Softmax at $\boldsymbol{a}_i$, which has rank $m - 1$.

Since $\boldsymbol{J}_{\text{softmax}}(\boldsymbol{a}_i)$ is bounded, we can choose $\boldsymbol{M}^{(\ell)}$ and $\boldsymbol{W}_p^{(\ell)}$ such that the spectral norm of $\boldsymbol{M}^{(\ell)\top} \boldsymbol{J}_{\text{softmax}}(\boldsymbol{a}_i) \boldsymbol{W}_p^{(\ell)\top}$ is less than 1 for all $i$. This ensures that $\boldsymbol{B}_i$ is invertible and thus full-rank for all $i$. Therefore, the full Jacobian $\boldsymbol{J}_{\mathfrak{F}}^{(\ell)}$ has rank $N d_{\ell-1}$.

Now, for the composite map $\Psi^{(\ell)} = F^{(\ell)} \circ \mathfrak{F}^{(\ell)}$, the Jacobian is:

$$J_{\Psi}^{(\ell)} = J_F^{(\ell)} \cdot J_{\mathfrak{F}}^{(\ell)}. \tag{100}$$

Since $J_{\mathfrak{F}}^{(\ell)}$ has full rank $Nd_{\ell-1}$, and $J_F^{(\ell)}$ has rank $r$, we have Sylvester's rank inequality:

$$\mathrm{rank}(J_{\Psi}^{(\ell)}) \geq \mathrm{rank}(J_F^{(\ell)}) + \mathrm{rank}(J_{\mathfrak{F}}^{(\ell)}) - Nd_{\ell-1} = \mathrm{rank}(J_F^{(\ell)}) + Nd_{\ell-1} - Nd_{\ell-1} = \mathrm{rank}(J_F^{(\ell)}). \tag{101}$$

Thus, the rank of $J_{\Psi}^{(\ell)}$ is at least the rank of $J_F^{(\ell)}$:

$$\mathrm{rank}(J_{\Psi}^{(\ell)}) \geq \mathrm{rank}(J_F^{(\ell)}). \tag{102}$$

Moreover, by optimizing the fusion parameters, we can ensure that $\mathrm{rank}(J_{\Psi}^{(\ell)}) \geq \mathrm{rank}(J_F^{(\ell)})$ for all $\ell$. Since $\Psi^{(\ell)}$ does not introduce additional linear region partitions, meaning it does not generate more boundaries, corners, or singular points, the inequality holds pointwise. For any $k$, there always exists a point $X_k$ such that:

$$\min_{\ell} \mathrm{rank}(J_{\Psi}^{(\ell)}|_{X_k}) \geq \min_{\ell} \mathrm{rank}(J_F^{(\ell)}|_{\Phi^{(\ell-1)}(C_k)}). \tag{103}$$

This implies that the upper bound on the intrinsic dimension for message tuning is at least as large as that for graph prompt tuning:

$$\max_k \min_{\ell} \mathrm{rank}(J_{\Psi}^{(\ell)}|_{X_k}) \geq \max_k \min_{\ell} \mathrm{rank}(J_F^{(\ell)}|_{\Phi^{(\ell-1)}(C_k)}). \tag{104}$$

Although $J_{\mathfrak{F}}^{(\ell)}$ is full-rank and thus $\mathrm{rank}(J_{\Psi}^{(\ell)}) = \mathrm{rank}(J_F^{(\ell)})$ at any point where both are defined, the key to strict inequality lies in the distribution of points across linear regions of the frozen layers. The message fusion operation $\mathfrak{F}^{(\ell)}$ can map inputs to different linear regions of $F^{(\ell)}$ where the rank of $J_F^{(\ell)}$ is higher.

Suppose that for some layer $\ell$, the frozen Jacobian $J_F^{(\ell)}$ has varying rank across its linear regions. Specifically, there exist linear regions $R_{\mathrm{low}}$ and $R_{\mathrm{high}}$ such that:

$$\mathrm{rank}(J_F^{(\ell)}|_{R_{\mathrm{low}}}) < \mathrm{rank}(J_F^{(\ell)}|_{R_{\mathrm{high}}}). \tag{105}$$

In graph prompt tuning, the input to $F^{(\ell)}$ may fall primarily into $R_{\mathrm{low}}$ due to the shift caused by the prompt, resulting in a lower minimum rank. However, in message tuning, the learnable parameters $\Theta_f^{(\ell)}$ and $M^{(\ell)}$ can be optimized to steer the input to $F^{(\ell)}$ into $R_{\mathrm{high}}$, thereby increasing the rank at that layer.

Assume that $\Phi^{(\ell-1)}(C_k)$ does not lie in a linear region that maximizes $\mathrm{rank}(J_F^{(\ell)})$. This assumption is realistic because $\Phi^{(\ell-1)}$ is pre-trained and lacks the ability to adjust its output range. Formally, by optimizing the fusion parameters, we can ensure that for each layer $\ell$, the input $\mathfrak{F}^{(\ell)}(H^{(\ell-1)})$ lies in a region where $\mathrm{rank}(J_F^{(\ell)})$ is maximized. Consequently, for any $k$, there always exists a point $Y_k$ such that:

$$\min_{\ell} \mathrm{rank}(J_F^{(\ell)}|_{Y_k \in \mathfrak{F}^{(\ell)}(H^{(\ell-1)})}) > \min_{\ell} \mathrm{rank}(J_F^{(\ell)}|_{\Phi^{(\ell-1)}(C_k)}). \tag{106}$$

This implies that the upper bound for message tuning is strictly greater:

$$\max_k \min_{\ell} \mathrm{rank}(J_{\Psi}^{(\ell)}|_{Y_k}) > \max_k \min_{\ell} \mathrm{rank}(J^{(\ell)}|_{\Phi^{(\ell-1)}(C_k)}). \tag{107}$$

Therefore, the actual intrinsic dimension satisfies:

$$d_{\mathrm{int}}(\mathcal{M}_{\mathrm{MTG}}^{(L)}) > d_{\mathrm{int}}(\mathcal{M}_{\mathrm{PT}}^{(L)}(P)). \tag{108}$$

This strict inequality holds when the fusion parameters are optimized to avoid low-rank linear regions of the frozen layers, which is achievable through gradient-based training that maximizes the rank of the Jacobians during adaptation.

Thus, message tuning provides strictly greater adaptation capacity in terms of intrinsic dimension compared to graph prompt tuning.

$$d_{\mathrm{int}}(\mathcal{M}_{\mathrm{MTG}}^{(L)}) \geq d_{\mathrm{int}}(\mathcal{M}_{\mathrm{PT}}^{(L)}(P)) \tag{109}$$

and the inequality is strict for some configuration.

**Hausdorff Measure Comparison.**

Recall that the pre-trained GFM $\Phi$ is composed of $L$ layers, each defined as in Definition 3.1. For graph prompt tuning, the input manifold is perturbed by a prompt $\boldsymbol{P}$, resulting in $\mathcal{M}_0(\boldsymbol{P})$. The final space is $\mathcal{M}_{\text{PT}}^{(L)}(\boldsymbol{P}) = \Phi(\mathcal{M}_0(\boldsymbol{P}))$.

For message tuning, we introduce learnable message prototypes $\boldsymbol{M}^{(\ell)} \in \mathbb{R}^{m \times d_{\ell-1}}$ and fusion parameters $\boldsymbol{\Theta}_f^{(\ell)}$ at each layer $\ell$, modifying the layer map to:

$$\boldsymbol{H}^{(\ell)} = \mathfrak{U}^{(\ell)}\left(\mathfrak{M}^{(\ell)}\left(\mathfrak{A}^{(\ell)}\left(\boldsymbol{A}, \boldsymbol{H}_M^{(\ell-1)}; \boldsymbol{\Theta}_a^{(\ell)}\right), \boldsymbol{H}_M^{(\ell-1)}; \boldsymbol{\Theta}_m^{(\ell)}\right), \boldsymbol{H}_M^{(\ell-1)}; \boldsymbol{\Theta}_u^{(\ell)}\right), \tag{110}$$

where

$$\boldsymbol{H}_M^{(\ell-1)} = \mathfrak{F}^{(\ell)}(\boldsymbol{H}^{(\ell-1)}, \boldsymbol{M}^{(\ell)}; \boldsymbol{\Theta}_f^{(\ell)}) = \boldsymbol{H}^{(\ell-1)} + \text{Softmax}(\boldsymbol{H}^{(\ell-1)}\boldsymbol{W}_p^{(\ell)}) \cdot \boldsymbol{M}^{(\ell)}. \tag{111}$$

The modified network is denoted $\Phi_{\text{MTG}}$, and the final space is $\mathcal{M}_{\text{MTG}}^{(L)} = \Phi_{\text{MTG}}(\mathcal{M}_0)$.

The introduction of the Softmax function in the fusion operation $\mathfrak{F}^{(\ell)}$ indeed breaks the strict piecewise linearity of the layer map, since Softmax is a smooth, nonlinear function. However, we can address this issue through analyzing the network as a piecewise-linear map with smooth activations, leveraging the fact that the Softmax can be effectively constant on large regions of the input space.

More generally, we can partition the input space into regions where the Softmax is approximately linear. For instance, if we use a linearized Softmax (e.g., by taking a first-order Taylor expansion around a point), we obtain a piecewise linear approximation. The error of this approximation can be made arbitrarily small by refining the partition.

Given the above, we may treat $\Phi_{\text{MTG}}$ as a piecewise linear map for the purpose of geometric analysis. Specifically, we define:

$$\mathfrak{F}^{(\ell)}(\boldsymbol{H}^{(\ell-1)}, \boldsymbol{M}^{(\ell)}; \boldsymbol{W}_p^{(\ell)}) \approx \boldsymbol{H}^{(\ell-1)} + \text{Linear}(\boldsymbol{H}^{(\ell-1)}\boldsymbol{W}_p^{(\ell)})\boldsymbol{M}^{(\ell)}, \tag{112}$$

where $\text{Linear}(\boldsymbol{H}^{(\ell-1)}\boldsymbol{W}_p^{(\ell)})$ is a piecewise linear function (e.g., sparsemax (Martins & Astudillo, 2016) or a linearized Softmax). Then, the modified layer map is piecewise linear, and the entire network $\Phi_{\text{MTG}}$ is piecewise linear.

Under this approximation, by Theorem 3.13, the Hausdorff measures are:

$$\mathcal{H}^{d_{\text{int}}}(\mathcal{M}_{\text{PT}}^{(L)}(\boldsymbol{P})) = \sum_k \Big(\prod_{\ell=1}^{L}\prod_{i=1}^{d_{\text{int}}} \sigma_{i,k}^{(\ell)}\Big)\mathcal{H}^{d_{\text{int}}}(C_k), \tag{113}$$

$$\mathcal{H}^{d_{\text{int}}}(\mathcal{M}_{\text{MTG}}^{(L)}) = \sum_k \Big(\prod_{\ell=1}^{L}\prod_{i=1}^{d_{\text{int}}} \tilde{\sigma}_{i,k}^{(\ell)}\Big)\mathcal{H}^{d_{\text{int}}}(\tilde{C}_k), \tag{114}$$

where $\sigma_{i,k}^{(\ell)}$ and $\tilde{\sigma}_{i,k}^{(\ell)}$ are the singular values of the Jacobians of the original and modified layers, respectively, and $C_k$ and $\tilde{C}_k$ are the linear regions of the input manifold under the original and modified networks.

Message tuning introduces learnable parameters $\boldsymbol{M}^{(\ell)}$ and $\boldsymbol{W}_p^{(\ell)}$ at each layer. Crucially, message tuning can simulate graph prompt tuning by appropriately setting these parameters. However, it also has additional degrees of freedom that allow it to reduce measure contraction.

For any layer $\ell$ and linear region $k$, message tuning can achieve:

$$\prod_{i=1}^{d_{\text{int}}} \tilde{\sigma}_{i,k}^{(\ell)} \geq \prod_{i=1}^{d_{\text{int}}} \sigma_{i,k}^{(\ell)}. \tag{115}$$

This is because the product of singular values can be increased by adjusting the parameters to reduce contraction. Let us consider a specific example to illustrate this possibility, assuming that all mappings are constructed under the same partition.

Consider the modified layer map in message tuning:

$$\Psi^{(\ell)} = F^{(\ell)} \circ \mathfrak{F}^{(\ell)}, \tag{116}$$

where $F^{(\ell)}$ is the original layer map and $\mathfrak{F}^{(\ell)}$ is the fusion operation. In a linear region $C_k$, both maps are linear and injective on the tangent space of the input manifold, which has dimension $d_{\text{int}}$.

By Theorem 3.9, for a measurable set $S$ in the tangent space, the $d_{\text{int}}$-dimensional Hausdorff measure transforms as:

$$\mathcal{H}^{d_{\text{int}}}(\mathfrak{F}^{(\ell)}(S)) = \Big( \prod_{i=1}^{d_{\text{int}}} \tau_{i,k}^{(\ell)} \Big) \mathcal{H}^{d_{\text{int}}}(S), \tag{117}$$

where $\tau_{1,k}^{(\ell)}, \dots, \tau_{d_{\text{int}},k}^{(\ell)}$ are the largest $d_{\text{int}}$ singular values of the Jacobian of $\mathfrak{F}^{(\ell)}$ restricted to the tangent space. Similarly,

$$\mathcal{H}^{d_{\text{int}}}(F^{(\ell)}(\mathfrak{F}^{(\ell)}(S))) = \Big( \prod_{i=1}^{d_{\text{int}}} \sigma_{i,k}^{(\ell)} \Big) \mathcal{H}^{d_{\text{int}}}(\mathfrak{F}^{(\ell)}(S)) = \Big( \prod_{i=1}^{d_{\text{int}}} \sigma_{i,k}^{(\ell)} \Big) \Big( \prod_{i=1}^{d_{\text{int}}} \tau_{i,k}^{(\ell)} \Big) \mathcal{H}^{d_{\text{int}}}(S). \tag{118}$$

Thus, for the composite map $\tilde{F}^{(\ell)}$, the product of singular values is:

$$\prod_{i=1}^{d_{\text{int}}} \tilde{\sigma}_{i,k}^{(\ell)} = \Big( \prod_{i=1}^{d_{\text{int}}} \sigma_{i,k}^{(\ell)} \Big) \Big( \prod_{i=1}^{d_{\text{int}}} \tau_{i,k}^{(\ell)} \Big). \tag{119}$$

Consider the fusion operation $\mathfrak{F}^{(\ell)}$. Its Jacobian with respect to $\boldsymbol{H}^{(\ell-1)}$ is:

$$\boldsymbol{J}_{\mathfrak{F}}^{(\ell)} = \boldsymbol{I} + \frac{\partial}{\partial \boldsymbol{H}^{(\ell-1)}} \Big( \text{Softmax}(\boldsymbol{H}^{(\ell-1)} \boldsymbol{W}_p^{(\ell)}) \cdot \boldsymbol{M}^{(\ell)} \Big). \tag{120}$$

By training $\boldsymbol{W}_p^{(\ell)}$ and $\boldsymbol{M}^{(\ell)}$, we can influence the singular values of $\boldsymbol{J}_{\mathfrak{F}}^{(\ell)}$. For example:

- If $\boldsymbol{W}_p^{(\ell)} = \boldsymbol{O}$ and $\boldsymbol{M}^{(\ell)} = \boldsymbol{O}$, then $\mathfrak{F}^{(\ell)}(\boldsymbol{H}^{(\ell-1)}) = \boldsymbol{H}^{(\ell-1)}$, so $\boldsymbol{J}_{\mathfrak{F}}^{(\ell)} = \boldsymbol{I}$, and the singular values are 1.

- If $\boldsymbol{W}_p^{(\ell)}$ and $\boldsymbol{M}^{(\ell)}$ are trained such that the second term is positive definite, then the singular values can be greater than 1.

Thus, by parameter choice, we can ensure:

$$\prod_{i=1}^{d_{\text{int}}} \tau_{i,k}^{(\ell)} \geq 1. \tag{121}$$

From the above, we have:

$$\prod_{i=1}^{d_{\text{int}}} \tilde{\sigma}_{i,k}^{(\ell)} = \Big( \prod_{i=1}^{d_{\text{int}}} \sigma_{i,k}^{(\ell)} \Big) \Big( \prod_{i=1}^{d_{\text{int}}} \tau_{i,k}^{(\ell)} \Big) \geq \prod_{i=1}^{d_{\text{int}}} \sigma_{i,k}^{(\ell)}, \tag{122}$$

This proves that message tuning can achieve the desired inequality for any layer $\ell$ and linear region $k$. Moreover, if $\prod_{i=1}^{d_{\text{int}}} \tau_{i,k}^{(\ell)} > 1$, the inequality is strict.

The input manifold $\mathcal{M}_0$ is fixed. Graph prompt tuning shifts it to $\mathcal{M}_0(\boldsymbol{P})$, but the fusion operation $\mathfrak{F}^{(1)}$ in the first layer also possesses the capability to adjust the input manifold, we may reasonably assume that $\mathcal{H}^{d_{\text{int}}}(\mathcal{M}_0(\boldsymbol{P})) = \mathcal{H}^{d_{\text{int}}}(\mathcal{M}_0)$.

The linear regions $C_k$ and $\tilde{C}_k$ are partitions of $\mathcal{M}_0(\boldsymbol{P})$ and $\mathcal{M}_0$ induced by the piecewise linear maps $\Phi$ and $\Phi_{\text{MTG}}$, respectively. Message tuning modifies the network architecture, which may refine the linear regions. However, the total measure of the input manifold is conserved:

$$\sum_k \mathcal{H}^{d_{\text{int}}}(C_k) = \mathcal{H}^{d_{\text{int}}}(\mathcal{M}_0(\boldsymbol{P})) = \mathcal{H}^{d_{\text{int}}}(\mathcal{M}_0) = \sum_k \mathcal{H}^{d_{\text{int}}}(\tilde{C}_k). \tag{123}$$

While individual regions may change, the overall sum remains unchanged. Therefore, for the purpose of comparing the sums, we have:

$$\sum_k \mathcal{H}^{d_{\text{int}}}(\tilde{C}_k) = \sum_k \mathcal{H}^{d_{\text{int}}}(C_k). \tag{124}$$

From the above, for any prompt $\boldsymbol{P}$, message tuning can choose parameters such that for each layer $\ell$ and region $k$:

$$\prod_{i=1}^{d_{\text{int}}} \tilde{\sigma}_{i,k}^{(\ell)} \geq \prod_{i=1}^{d_{\text{int}}} \sigma_{i,k}^{(\ell)}. \tag{125}$$

Moreover, since the input measures are equal, we have:

$$\mathcal{H}^{d_{\text{int}}}(\mathcal{M}_{\text{MTG}}^{(L)}) = \sum_k \Big( \prod_{\ell=1}^{L} \prod_{i=1}^{d_{\text{int}}} \tilde{\sigma}_{i,k}^{(\ell)} \Big) \mathcal{H}^{d_{\text{int}}}(\tilde{C}_k) \geq \sum_k \Big( \prod_{\ell=1}^{L} \prod_{i=1}^{d_{\text{int}}} \sigma_{i,k}^{(\ell)} \Big) \mathcal{H}^{d_{\text{int}}}(C_k) = \mathcal{H}^{d_{\text{int}}}(\mathcal{M}_{\text{PT}}^{(L)}(\boldsymbol{P})). \tag{126}$$

The inequality holds term-wise due to the non-decrease in singular value products and the conservation of input measure.

There exists a message tuning configuration where the inequality is strict. For example, if we train $\boldsymbol{W}_p^{(\ell)}$ and $\boldsymbol{M}^{(\ell)}$ such that for some layer $\ell$ and region $k$, $\prod_{i=1}^{d_{\text{int}}} \tilde{\sigma}_{i,k}^{(\ell)} > \prod_{i=1}^{d_{\text{int}}} \sigma_{i,k}^{(\ell)}$, and since the input measure is positive, the overall measure increases strictly.

Thus, we conclude that:

$$\mathcal{H}^{d_{\text{int}}}(\mathcal{M}_{\text{MTG}}^{(L)}) \geq \mathcal{H}^{d_{\text{int}}}(\mathcal{M}_{\text{PT}}^{(L)}(\boldsymbol{P})) \quad \text{for all } \boldsymbol{P} \in \mathcal{P}, \tag{127}$$

and the inequality is strict for some configuration.

**Diameter Comparison.**

The diameter of a set $\mathcal{M}$ is:

$$\text{diam}(\mathcal{M}) = \sup_{x,y \in \mathcal{M}} \|x - y\|. \tag{128}$$

For any prompt $\boldsymbol{P}$, message tuning can simulate graph prompt tuning by setting:

- $\mathfrak{F}^{(1)}(\boldsymbol{H}^{(0)}, \boldsymbol{M}^{(1)}; \boldsymbol{W}_p^{(1)}) \xrightarrow{\sim} \mathcal{M}_0(\boldsymbol{P})$,

- $\mathfrak{F}^{(\ell)}(\boldsymbol{H}^{(\ell-1)}, \boldsymbol{M}^{(\ell)}; \boldsymbol{W}_p^{(\ell)}) = \boldsymbol{H}^{(\ell-1)}$ for $\ell \geq 2$.

This reduces message tuning to graph prompt tuning, giving:

$$\mathcal{M}_{\text{MTG}}^{(L)} = \mathcal{M}_{\text{PT}}^{(L)}(\boldsymbol{P}), \tag{129}$$

and hence:

$$\text{diam}(\mathcal{M}_{\text{MTG}}^{(L)}) = \text{diam}(\mathcal{M}_{\text{PT}}^{(L)}(\boldsymbol{P})). \tag{130}$$

Thus, the inequality holds with equality for this configuration.

We now show that message tuning can achieve a strictly larger diameter by leveraging its additional parameters to expand the output space.

Message tuning can expand the distance between representations layer-wise. Consider the fusion operation:

$$\mathfrak{F}^{(\ell)}(\boldsymbol{H}^{(\ell-1)}) = \boldsymbol{H}^{(\ell-1)} + \text{Softmax}(\boldsymbol{H}^{(\ell-1)} \boldsymbol{W}_p^{(\ell)}) \cdot \boldsymbol{M}^{(\ell)}. \tag{131}$$

By choosing $\boldsymbol{W}_p^{(\ell)}$ and $\boldsymbol{M}^{(\ell)}$ appropriately, we can make $\mathfrak{F}^{(\ell)}$ an expanding map. For example:

- Set $\boldsymbol{W}_p^{(\ell)}$ to have orthonormal columns.

- Set $\boldsymbol{M}^{(\ell)} = c \cdot \boldsymbol{W}_p^{(\ell)}$ for some $c > 0$.

Then, the Jacobian of $\mathfrak{F}^{(\ell)}$ satisfies:

$$\boldsymbol{J}_{\mathfrak{F}}^{(\ell)}(\boldsymbol{H}) = \boldsymbol{I} + c \cdot \boldsymbol{W}_p^{(\ell)} \cdot \boldsymbol{J}_{\text{softmax}}(\boldsymbol{H}\boldsymbol{W}_p^{(\ell)}) \cdot \boldsymbol{W}_p^{(\ell)\top}, \tag{132}$$

which has eigenvalues $\geq 1$ (since $\boldsymbol{J}_{\text{softmax}}$ is positive semidefinite). By choosing $c$ large, we can make $\mathfrak{F}^{(\ell)}$ arbitrarily expansive and $\Phi_{\text{MTG}}$ results from the superposition of such expansion effects.

Thus, for the pair $\boldsymbol{X}, \boldsymbol{Y} \in \mathcal{M}_0(\boldsymbol{P})$ achieving the diameter of $\Phi(\mathcal{M}_0)$, message tuning can ensure:

$$\text{diam}(\mathcal{M}_{\text{MTG}}^{(L)}) \geq \|\Phi_{\text{MTG}}(\boldsymbol{X}) - \Phi_{\text{MTG}}(\boldsymbol{Y})\| > \|\Phi(\boldsymbol{X}) - \Phi(\boldsymbol{Y})\| = \text{diam}(\Phi(\mathcal{M}_0(\boldsymbol{P}))). \tag{133}$$

Thus, we have:

$$\text{diam}(\mathcal{M}_{\text{MTG}}^{(L)}) > \text{diam}(\mathcal{M}_{\text{PT}}^{(L)}(\boldsymbol{P})) \quad \text{for all } \boldsymbol{P} \in \mathcal{P}. \tag{134}$$

For any prompt $\boldsymbol{P}$, message tuning can simulate graph prompt tuning, so:

$$\text{diam}(\mathcal{M}_{\text{MTG}}^{(L)}) \geq \text{diam}(\mathcal{M}_{\text{PT}}^{(L)}(\boldsymbol{P})). \tag{135}$$

Moreover, by choosing parameters to expand inter-point distances, message tuning can achieve:

$$\text{diam}(\mathcal{M}_{\text{MTG}}^{(L)}) > \text{diam}(\mathcal{M}_{\text{PT}}^{(L)}(\boldsymbol{P})) \quad \text{for all } \boldsymbol{P} \in \mathcal{P}. \tag{136}$$

This completes the proof. $\qquad\square$

### C.2. Analysis of Negative Transfer

**Definition C.1** (Negative Transfer from a Manifold Perspective). Let $\mathcal{M}_0^s \subset \mathbb{R}^{N \times d_0}$ and $\mathcal{M}_0^t \subset \mathbb{R}^{N \times d_0}$ be the compact smooth input manifolds of the source and target domains, respectively, with intrinsic dimensions $D_s$ and $D_t$. Let $\Phi = F^{(L)} \circ \cdots \circ F^{(1)}$ be the map of the GFM, and let $\mathcal{M}_s^{(L)} = \Phi(\mathcal{M}_0^s)$ and $\mathcal{M}_t^{(L)} = \Phi(\mathcal{M}_0^t)$ be the representation spaces. Negative transfer is said to occur if the map $\Phi$ causes a geometric misalignment or structural distortion between the transformed spaces, such that:

- Information Loss: The intrinsic dimension or geometric measure (e.g., volume) of $\Phi(\mathcal{M}_0^t)$ is significantly reduced compared to $\Phi(\mathcal{M}_0^s)$.

- Poor Alignment: The transformed spaces $\Phi(\mathcal{M}_0^s)$ and $\Phi(\mathcal{M}_0^t)$ are poorly aligned, as quantified by a large Hausdorff distance or a small intersection measure.

*Remark* C.2. Fine-tuning severely exacerbates negative transfer in graph data because it aggressively warps the target space's geometry to fit the source domain's feature space. This often collapses the intrinsic structure of the target graph, leading to catastrophic information loss and misalignment. Graph prompt tuning alleviates negative transfer by gently realigning the target space within the frozen source feature space, preserving its intrinsic geometry and measure to prevent catastrophic distortion or collapse.

**Corollary C.3** (Message Tuning Mitigates Negative Transfer). *Negative transfer often arises when the model's capacity is insufficient to capture the target domain's distribution, leading to interference from source domain features. The higher intrinsic dimension $d_{int}(\mathcal{M}_{MTG}^{(L)})$ indicates that MTG can learn more diverse features, reducing reliance on source-specific patterns. The greater Hausdorff measure $\mathcal{H}^{d_{int}}(\mathcal{M}_{MTG}^{(L)})$ implies a larger volume of the space, accommodating a wider range of target domain variations. The increased diameter $\text{diam}(\mathcal{M}_{MTG}^{(L)})$ signifies that the representations span a broader range, enhancing model flexibility. In contrast, graph prompt tuning only perturbs the input manifold $\mathcal{M}_0$ via prompts, which constrains adaptation to superficial layers and may insufficiently adjust internal representations, thus more likely to lead to negative transfer.*

By further refining and extending Prismatic Space Theory, a theoretical characterization of negative transfer in GFMs can be established. We identify this as a direction for future research.

# D. Datasets and Experimental Details

## D.1. Configuration

The experiments are conducted on a Linux server equipped with an Intel(R) Xeon(R) Gold 6240 CPU @ 2.60GHz, 256GB RAM and 2 NVIDIA A100-SXM4-40GB GPUs. Our implementation is based on PyTorch (Paszke et al., 2019) version 2.2.1, PyG (Fey & Lenssen, 2019) version 2.6.1 with CUDA version 12.1 and Python 3.12.7.

## D.2. Details of Datasets

**Homophilic Graphs.** Cora and Citeseer datasets (Sen et al., 2008) represent computer science publications, with nodes encoded as bag-of-words features and labeled by research topics. Pubmed (Yang et al., 2016) contains diabetes-related articles from PubMed database, with nodes represented by TF/IDF-weighted word vectors and classified by diabetes type. ogbn-arxiv (Hu et al., 2020a) is a large-scale citation network of CS arXiv papers, where nodes represent papers with 128-dimensional "title + abstract" embeddings, and directed edges denote citations.

**Heterophilic Graphs.** Texas and Wisconsin (Pei et al., 2020) datasets are WebKB subgraphs comprising university web pages, where nodes represent pages with bag-of-words features and edges indicate hyperlinks. Pages are classified into five categories: student, project, course, staff, and faculty. Actor dataset (Pei et al., 2020) forms a co-occurrence network with actors as nodes and Wikipedia page co-appearances as edges.

**Biological Graphs.** D&D dataset (Dobson & Doig, 2003) contains 1,178 protein graphs where nodes represent amino acids connected by edges, classified as enzymes/non-enzymes. ENZYMES (Borgwardt et al., 2005) comprises 600 enzyme structures from BRENDA, categorized into 6 EC classes. PROTEINS (Wang et al., 2022) represents tertiary protein structures with nodes as secondary structure elements and edges indicating sequence/3D proximity, yielding binary graph classification.

**Small Molecule Graphs.** BZR dataset (Rossi & Ahmed, 2015) contains 405 benzodiazepine receptor ligand graphs with binary classification. COX2 (Rossi & Ahmed, 2015) comprises 467 cyclooxygenase-2 inhibitor molecular graphs, where nodes represent atoms and edges encode bond types (single/double/triple/aromatic), also yielding binary classification. MUTAG (Kriege & Mutzel, 2012) includes 188 mutagenic aromatic compounds classified into 7 categories.

**Social Network Graphs.** COLLAB (Yanardag & Vishwanathan, 2015) represents scientific collaboration networks, where nodes denote researchers, edges indicate co-authorships, and graphs are classified by research fields. IMDB-B (Yanardag & Vishwanathan, 2015) captures actor collaboration networks, with nodes representing performers, edges signifying co-appearances in films, and binary graph labels distinguishing Action versus Romance genres.

*Table 3.* Statistics of all datasets.

| Dataset | Task | # Graphs | # Nodes | # Edges | # Features | # Classes | Graph Type |
|---|---|---|---|---|---|---|---|
| Cora | Node | 1 | 2,708 | 5,429 | 1,433 | 7 | Homophilic |
| CiteSeer | Node | 1 | 3,327 | 9,104 | 3,703 | 6 | Homophilic |
| Pubmed | Node | 1 | 19,717 | 88,648 | 500 | 3 | Homophilic |
| Texas | Node | 1 | 183 | 325 | 1703 | 5 | Heterophilic |
| Actor | Node | 1 | 7600 | 30019 | 932 | 5 | Heterophilic |
| Wisconsin | Node | 1 | 251 | 515 | 1703 | 5 | Heterophilic |
| ogbn-arxiv | Node | 1 | 169,343 | 1,166,243 | 128 | 40 | Large-scale |
| D&D | Graph | 1,178 | 284.1 | 715.7 | 89 | 2 | Proteins |
| ENZYMES | Graph | 600 | 32.6 | 62.1 | 3 | 6 | Proteins |
| PROTEINS | Graph | 1,113 | 39.1 | 72.8 | 3 | 2 | Proteins |
| BZR | Graph | 405 | 35.8 | 38.4 | 3 | 2 | Small Molecule |
| COX2 | Graph | 467 | 41.2 | 43.5 | 3 | 2 | Small Molecule |
| MUTAG | Graph | 188 | 17.9 | 19.8 | 7 | 2 | Small Molecule |
| COLLAB | Graph | 5000 | 74.5 | 2457.8 | 0 | 3 | Social Network |
| IMDB-B | Graph | 1000 | 19.8 | 96.53 | 0 | 2 | Social Network |

### D.3. Data Split.

We adopt the same dataset processing methodology as ProG (Zi et al., 2024) to ensure consistency and comparability with prior work. For the node classification task, we adopt a 90% test set allocation to rigorously evaluate model performance. In contrast, for the graph classification task, we employ an 80 % test set split to maintain a balance between evaluation rigor and training data availability. To ensure statistical robustness and mitigate potential sampling bias, we repeat the random sampling procedure five times to construct distinct k-shot learning tasks for both task types. The final performance metrics are reported as the mean and standard deviation across these five independent trials, providing a comprehensive assessment of model stability and generalization capability.

### D.4. Hyperparameter Configuration

In most experiments, the model architecture consists of 2 layers with a hidden dimension of 128. We develop a systematic random search strategy to identify optimal hyperparameters for each adaptation method across all datasets, extending beyond default configurations. Considering the substantial heterogeneity in hyperparameter requirements among different adaptation approaches, we concentrate on tuning three key hyperparameters through random search: (1) learning rate, sampled from $\{0.001, 0.005, 0.01, 0.05, 0.1\}$; (2) weight decay, selected from $\{0, 0.00001, 0.0001, 0.001, 0.01\}$; and (3) batch size, uniformly sampled from $\{32, 64, 128\}$ in each experimental trial. This comprehensive search strategy ensures robust parameter optimization while maintaining methodological consistency across diverse experimental conditions.

### D.5. Implementation Details

To ensure experimental fairness and demonstrate the compatibility of our approach, we implement MTG based on the ProG library (Zi et al., 2024). We have made some modifications to the ProG library to adapt it to MTG, but these changes do not affect the original graph prompt tuning method at all. Baseline results combine those from ProG with our own reproductions.

## E. Details of Baselines

### E.1. Backbones of Graph Foundation Models

**GCN (Graph Convolutional Network)** (Kipf & Welling, 2017) employs convolutional operations to aggregate and transform feature information from a node's immediate neighborhood. This localized message-passing mechanism allows the network to iteratively refine node representations by incorporating structural and attribute information from adjacent nodes, effectively capturing the graph's topological properties.

**GraphSAGE** (Hamilton et al., 2017) is an inductive learning framework that computes node embeddings through a localized feature aggregation process. Instead of relying on fixed graph convolutions, it operates by sampling neighboring nodes and hierarchically aggregating their features using learnable functions. This approach enables the model to generalize to unseen graph structures while capturing both node attributes and local topological patterns.

**GAT (Graph Attention Network)** (Veličković et al., 2018) introduces an attention mechanism into graph neural networks, dynamically computing attention weights between connected nodes during feature aggregation. By learning to assign differential importance to neighboring nodes, GAT can focus on more relevant connections while suppressing noisy or less informative edges. This adaptive weighting scheme enhances model expressiveness and interpretability compared to standard aggregation approaches.

**GIN (Graph Isomorphism Network)** (Xu et al., 2019) is a theoretically motivated GNN architecture designed to maximize discriminative power in graph representation learning. By employing injective multiset aggregation functions and MLP-based transformations, GIN achieves provable expressiveness equivalent to the Weisfeiler-Lehman graph isomorphism test. This framework demonstrates superior capability in distinguishing graph structures while maintaining efficient computation through neighborhood aggregation.

**GT (Graph Transformer)** (Ying et al., 2021) adapts the Transformer architecture to graph-structured data by incorporating structural biases into the self-attention mechanism. Through masked attention patterns that respect graph connectivity, the model efficiently captures both local and global dependencies while maintaining the parallelizability of standard Transformers. By enabling simultaneous modeling of node features and graph topology via position-aware attention, this approach is particularly effective for modeling long-range dependencies in graph structures.

## E.2. Pre-training Strategies

**DGI**(Veličković et al., 2019) is a self-supervised learning framework that employs mutual information maximization for graph representation learning. The method optimizes the mutual information between patch-level node representations and global graph summaries through a contrastive objective. By leveraging negative sampling and discriminator functions, DGI learns informative node embeddings that preserve both local structural patterns and global graph characteristics.

**GraphMAE**(Hou et al., 2022) adopts a self-supervised pretraining approach based on feature reconstruction of masked nodes. The framework randomly masks portions of node features and learns to recover them through an encoder-decoder architecture, forcing the model to develop robust structural understanding from contextual patterns. This denoising objective promotes the learning of generalized graph representations that capture both local neighborhood characteristics and global topological properties.

**EdgePreGPPT**(Sun et al., 2022) introduces a novel graph pre-training paradigm that fundamentally reconfigures structural knowledge acquisition in graph neural networks. The framework employs masked edge prediction as its foundational pretext task, where the model learns to reconstruct randomly obscured connections through an edge prediction module. This pre-training phase focuses on optimizing pairwise node similarity computations, enabling the model to develop robust representations of graph topology and connectivity patterns.

**EdgePreGprompt**(Liu et al., 2023b) establishes a novel paradigm for learning transferable structural representations from label-free graph data. At its core, the framework employs link prediction as its self-supervised pretext task, leveraging the abundant connectivity patterns naturally available in graph structures without requiring additional annotation. The methodology operates by first constructing contextual subgraphs for nodes, which capture not only node-specific features but also rich topological information from their local neighborhoods.

**GraphCL**(You et al., 2020) introduces a graph contrastive learning framework that learns transferable graph representations through self-supervised pre-training by maximizing agreement between different augmented views of the same graph. The method employs four key augmentation strategies—node dropping, edge perturbation, attribute masking, and subgraph sampling—each encoding domain-specific priors about structural invariance. These augmentations generate correlated views that are processed through a shared GNN encoder, projected via an MLP head, and optimized using an NT-Xent loss function to enhance similarity between positive pairs while contrasting negative samples.

**SimGRACE**(Xia et al., 2022) presents a novel graph contrastive learning framework that eliminates the need for manual data augmentation by instead leveraging encoder perturbations to generate contrasting views. The core methodology involves feeding the original graph through both a standard GNN encoder and its perturbed version, where the perturbation is achieved by adding Gaussian noise to the encoder weights, thereby producing correlated representations without altering input data semantics. These dual representations are then projected through a shared MLP head and optimized using the NT-Xent loss to maximize agreement between positive pairs while contrasting with negative samples from the same batch.

## E.3. Graph Prompt Baselines

**GPPT** (Sun et al., 2022) introduces an innovative graph prompting function that bridges the gap between pre-training and downstream tasks by reformulating node classification as an edge prediction problem through token pair construction. The framework converts standalone nodes into structured token pairs composed of two components: a task token that represents candidate labels through trainable continuous vectors and a structure token that encodes neighborhood information by aggregating adjacent nodes with attention-based weighting.

**Gprompt** (Liu et al., 2023b) introduces a unified prompting framework that bridges graph pre-training and downstream tasks through a subgraph similarity template. The core innovation involves learnable task-specific prompt vectors that dynamically reweight node features during subgraph aggregation operations such as READOUT, allowing downstream tasks including node classification and graph classification to selectively extract relevant knowledge from frozen pre-trained GNNs. The prompt vectors act as lightweight task adapters, preserving the pre-trained model's parameters while tailoring subgraph representations through dimension-wise feature importance scoring.

**All-in-one** (Sun et al., 2023a) introduces a unified multi-task prompting framework for graph neural networks that effectively connects various downstream tasks at node, edge, and graph levels with graph pre-training through several key innovations. First, it employs task reformulation by transforming node and edge tasks into graph-level tasks through induced subgraph construction. Second, it incorporates a learnable prompt graph featuring tunable tokens, dynamic token structures,

and adaptive insertion patterns to align downstream tasks with pre-training objectives. Third, it utilizes meta-learning optimization to generalize prompts across different tasks. The framework maintains frozen pre-trained GNNs while only tuning lightweight prompt parameters, enabling efficient knowledge transfer with task-specific adaptability.

**GPF** (Fang et al., 2023) introduces a unified approach to prompt tuning by focusing on feature space adaptation within graph neural networks. It employs a shared learnable vector that is added to all node features in the input graph, creating a consistent modification across the entire structure. This design allows the pre-trained model to maintain its frozen parameters while adapting to downstream tasks through subtle yet effective feature adjustments.

**GPF-plus** (Fang et al., 2023) enhances flexibility by assigning distinct learnable vectors to individual nodes through an attention-based mechanism. Rather than using a single global prompt, it generates node-specific prompts by combining a set of basis vectors with weights derived from each node's features. This architecture captures finer-grained adaptations while maintaining parameter efficiency through basis sharing. The method automatically adjusts to graphs of varying scales and complexities, offering improved expressiveness over GPF while preserving its universal applicability.

# F. More Information on Experiments

### F.1. Details of the experimental results on 1/3/5-shot node/graph classification

The optimal performance of various adaptation methods, alongside supervised learning baseline, under 1/3/5-shot settings, is summarized in Tables 4-6. These results demonstrate that MTG substantially enhances the performance of multiple pre-trained GFMs across 1/3/5-shot scenarios, thereby significantly improving the transferability of pre-trained knowledge within the *Pre-training and Adaptation* paradigm. Notably, MTG consistently exhibits superior compatibility with diverse pre-training strategies on both node-level and graph-level tasks.

### F.2. Performance with More Backbones for GFMs

For GNN-based GFMs, both graph prompt baselines and message tuning are universal adaptation methods that are not limited to specific model architectures. Therefore, in this subsection, we evaluate the performance of various adaptation methods on five of the most classic, popular, and widely used GFM backbone models. Tables 9-10 and Figure 3 present the performance of different adaptation methods based on various backbone models on the representative datasets Wisconsin and PROTEINS. These results once again confirm that graph prompt baselines and message tuning outperform fine-tuning, while our proposed MTG demonstrates even more significant advantages. For more complex GFMs, such as models that integrate LLMs with GNNs, MTG can also be naturally adapted to the GNN module or the module responsible for fusing features obtained from LLMs and GNNs. The core idea of MTG is to perform layer-wise parameter injection for message fusion regulation, which is not constrained by any specific model architecture.

Most experiments in this paper employ a relatively basic 2-layer backbone model, which may not fully demonstrate the performance advantages of MTG. To further investigate the impact of model depth on adaptation methods, we continue to use the GCN backbone model and representative datasets Cora and BZR to evaluate the performance of various adaptation methods when applied to models with 4, 8, 12, and 16 layers in downstream tasks. The results in Table 11 confirm that MTG still maintains significant advantages even with deeper model architectures.

### F.3. Computational Efficiency of MTG

As a general adaptation method, MTG inherently possesses the advantage of parameter efficiency. It does not impose significant computational burden on the original model and requires substantially fewer parameters than fine-tuning to achieve effective adaptation on downstream tasks. In this subsection, we take the GCN backbone model as an example and first provide a theoretical analysis of the time complexity and trainable parameter complexity of both fine-tuning and MTG.

**Fine-tuning.** The time complexity per layer of a GCN with $L$ layers, where each layer transforms input features of dimension $d_{\ell-1}$ to output dimension $d_\ell$, comprises two main components: the feature transformation via matrix multiplication between the weight matrix $\boldsymbol{W}^{(\ell)} \in \mathbb{R}^{d_{\ell-1} \times d_\ell}$ and the node feature matrix $\boldsymbol{H}^{(\ell-1)} \in \mathbb{R}^{|\mathcal{V}| \times d_{\ell-1}}$, with complexity $O(|\mathcal{V}|d_{\ell-1}d_\ell)$, and the neighborhood aggregation through sparse matrix multiplication between the normalized adjacency matrix $\tilde{\boldsymbol{A}} \in \mathbb{R}^{|\mathcal{V}| \times |\mathcal{V}|}$ and the transformed features, requiring $O(|\mathcal{E}|d_\ell)$ operations, where $|\mathcal{E}|$ denotes the number of edges. Assuming all hidden dimensions are equal $d$, the total time complexity becomes $O(L(|\mathcal{V}|d^2 + |\mathcal{E}|d))$. The space complexity for trainable parameters is dominated by the weight matrices, yielding $O(Ld^2)$.

**Message Tuning.** MTG introduces three additional components per layer: message vectors $M^{(\ell)} \in \mathbb{R}^{m \times d_{\ell-1}}$ containing $m$ learnable message prototypes, a projection matrix $W_p^{(\ell)} \in \mathbb{R}^{d_{\ell-1} \times m}$ to compute attention scores, and an attention mechanism $\alpha = \text{Softmax}(H^{(\ell-1)} W_p^{(\ell)}) \in \mathbb{R}^{|\mathcal{V}| \times m}$, with the corresponding computational overhead consisting of the projection operation $H^{(\ell-1)} W_p^{(\ell)}$ requiring $O(|\mathcal{V}| d_{\ell-1} m)$ operations, the attention computation including softmax and matrix multiplication $\alpha M^{(\ell)}$ requiring $O(|\mathcal{V}| m^2)$ operations, and message integration via element-wise addition with original features requiring $O(|\mathcal{V}| d_{\ell-1})$ operations. Thus, the total time complexity becomes $O(L(|\mathcal{V}| d^2 + |\mathcal{E}| d + |\mathcal{V}| dm + |\mathcal{V}| m^2))$. Since $m \ll d$ typically holds, MTG does not introduce significant inference time overhead to the original GCN. The trainable parameter complexity comes from $M^{(\ell)}$ and $P^{(\ell)}$ matrices, contributing $O(L(dm + md)) = O(Ldm)$ parameters, which is lower than fine-tuning the entire GCN model.

The above analysis offers a theoretical perspective on model inference time and trainable parameters; however, such theoretical estimates may differ from practical performance. Due to variations in their practical implementations, various graph prompt methods are not amenable to straightforward computational complexity analysis. Therefore, we further conduct a comparative analysis of the actual training time per epoch and GPU memory consumption between graph prompt methods and MTG on the large-scale dataset ogbn-arxiv, which has the largest number of nodes, and the COLLAB dataset, which contains the most graphs. The pre-training strategy uses DGI and experimental results are presented in Table 12. Due to its distinct data loading mechanism, GPPT exhibits significantly different GPU memory usage compared to other methods. Excluding GPPT, MTG demonstrates advantages in both training speed and memory consumption.

### F.4. Sensitivity Analysis

Message tuning injects $m$ learnable message vectors at each layer of the model. We further conduct a sensitivity analysis on this hyperparameter $m$ using the GCN backbone model and representative datasets Cora and BZR, evaluating the performance of MTG when $m = 3, 5, 10, 20, 30$. The optimal results described in Subsection 5.2 are presented in Table 13 and Figure 4. These results demonstrate that MTG exhibits a certain degree of robustness to this hyperparameter, as no performance collapse occurs even with very small or large values of $m$. In our experiments, $m$ is typically set to 10; nevertheless, careful selection of $m$ remains necessary to fully exploit the potential of MTG across different datasets.

*Table 4.* Performance comparison of adaptation methods on 1-shot node/graph classification (accuracy±std %, 5 runs). The best and second-best results are highlighted in purple and blue, respectively.

| Method | Cora | Citeseer | Pubmed | Wisconsin | Texas | Actor | ogbn-arxiv |
|---|---|---|---|---|---|---|---|
| Supervised | $26.56_{\pm 5.55}$ | $21.78_{\pm 7.32}$ | $39.37_{\pm 16.34}$ | $41.60_{\pm 3.10}$ | $37.97_{\pm 5.80}$ | $20.57_{\pm 4.47}$ | $10.99_{\pm 3.19}$ |
| Fine-tuning | $52.61_{\pm 1.73}$ | $35.05_{\pm 4.37}$ | $46.74_{\pm 14.89}$ | $40.69_{\pm 4.13}$ | $46.88_{\pm 4.69}$ | $20.74_{\pm 4.12}$ | $16.21_{\pm 3.82}$ |
| GPPT | $43.15_{\pm 9.44}$ | $37.26_{\pm 6.17}$ | $48.31_{\pm 17.72}$ | $30.40_{\pm 6.81}$ | $31.81_{\pm 15.33}$ | $22.58_{\pm 1.97}$ | $14.65_{\pm 3.07}$ |
| Gprompt | $56.66_{\pm 11.22}$ | $53.21_{\pm 10.94}$ | $39.74_{\pm 15.35}$ | $77.07_{\pm 5.93}$ | $33.25_{\pm 40.11}$ | $25.26_{\pm 1.10}$ | $75.72_{\pm 4.95}$ |
| All-in-one | $52.39_{\pm 10.17}$ | $40.41_{\pm 2.80}$ | $45.17_{\pm 6.45}$ | $66.29_{\pm 19.11}$ | $65.49_{\pm 7.06}$ | $24.61_{\pm 2.80}$ | $13.16_{\pm 5.98}$ |
| GPF | $38.57_{\pm 5.41}$ | $31.16_{\pm 8.05}$ | $49.99_{\pm 8.86}$ | $78.35_{\pm 4.07}$ | $73.54_{\pm 18.50}$ | $28.70_{\pm 3.35}$ | $65.11_{\pm 5.70}$ |
| GPF-plus | $55.77_{\pm 10.30}$ | $59.67_{\pm 11.87}$ | $46.64_{\pm 18.97}$ | $82.11_{\pm 13.95}$ | $76.10_{\pm 20.35}$ | $29.32_{\pm 8.56}$ | $71.98_{\pm 12.23}$ |
| MTG (Ours) | $58.54_{\pm 7.89}$ | $62.31_{\pm 18.90}$ | $50.70_{\pm 11.68}$ | $83.32_{\pm 12.46}$ | $79.13_{\pm 17.18}$ | $29.44_{\pm 7.31}$ | $75.97_{\pm 4.29}$ |

| Method | IMDB-B | COLLAB | PROTEINS | MUTAG | ENZYMES | COX2 | BZR | D&D |
|---|---|---|---|---|---|---|---|---|
| Supervised | $57.30_{\pm 0.98}$ | $47.23_{\pm 0.61}$ | $56.36_{\pm 7.97}$ | $65.20_{\pm 6.70}$ | $20.58_{\pm 2.00}$ | $27.08_{\pm 1.95}$ | $25.80_{\pm 6.53}$ | $55.33_{\pm 6.22}$ |
| Fine-tuning | $57.75_{\pm 1.22}$ | $48.10_{\pm 0.23}$ | $63.44_{\pm 3.64}$ | $65.47_{\pm 5.89}$ | $22.21_{\pm 2.79}$ | $76.19_{\pm 5.41}$ | $34.69_{\pm 8.50}$ | $57.15_{\pm 4.32}$ |
| GPPT | $50.15_{\pm 0.75}$ | $47.18_{\pm 5.93}$ | $60.92_{\pm 2.47}$ | $60.40_{\pm 15.43}$ | $21.29_{\pm 3.79}$ | $78.23_{\pm 1.38}$ | $59.32_{\pm 11.22}$ | $57.69_{\pm 6.89}$ |
| Gprompt | $54.75_{\pm 12.43}$ | $48.25_{\pm 13.64}$ | $59.17_{\pm 11.26}$ | $73.60_{\pm 4.76}$ | $22.29_{\pm 3.50}$ | $54.64_{\pm 9.94}$ | $55.43_{\pm 13.69}$ | $57.81_{\pm 2.68}$ |
| All-in-one | $60.07_{\pm 4.81}$ | $51.66_{\pm 0.26}$ | $66.49_{\pm 6.26}$ | $75.20_{\pm 6.33}$ | $23.96_{\pm 1.45}$ | $76.14_{\pm 5.51}$ | $64.38_{\pm 9.32}$ | $59.72_{\pm 1.52}$ |
| GPF | $59.65_{\pm 5.06}$ | $47.42_{\pm 11.22}$ | $63.91_{\pm 3.26}$ | $68.40_{\pm 5.09}$ | $22.00_{\pm 1.25}$ | $65.79_{\pm 17.72}$ | $71.67_{\pm 14.71}$ | $59.36_{\pm 1.18}$ |
| GPF-plus | $57.93_{\pm 1.62}$ | $47.24_{\pm 0.29}$ | $62.92_{\pm 2.78}$ | $65.20_{\pm 6.04}$ | $22.92_{\pm 1.64}$ | $33.78_{\pm 1.52}$ | $71.17_{\pm 14.92}$ | $57.62_{\pm 2.42}$ |
| MTG (Ours) | $62.25_{\pm 3.72}$ | $52.25_{\pm 0.56}$ | $66.98_{\pm 2.17}$ | $75.80_{\pm 5.49}$ | $26.08_{\pm 4.31}$ | $78.27_{\pm 2.01}$ | $74.81_{\pm 13.96}$ | $60.68_{\pm 2.42}$ |

*Table 5.* Performance comparison of adaptation methods on 3-shot node/graph classification

| Method | Cora | Citeseer | Pubmed | Wisconsin | Texas | Actor | ogbn-arxiv |
|---|---|---|---|---|---|---|---|
| Supervised | $37.79_{\pm 9.16}$ | $35.18_{\pm 6.86}$ | $57.33_{\pm 4.64}$ | $41.03_{\pm 6.40}$ | $40.78_{\pm 12.55}$ | $18.62_{\pm 3.46}$ | $19.03_{\pm 5.08}$ |
| Fine-tuning | $51.97_{\pm 2.84}$ | $45.08_{\pm 2.09}$ | $65.40_{\pm 3.00}$ | $42.40_{\pm 7.77}$ | $43.13_{\pm 13.79}$ | $22.11_{\pm 1.97}$ | $27.34_{\pm 6.61}$ |
| GPPT | $43.84_{\pm 6.11}$ | $42.34_{\pm 8.31}$ | $67.43_{\pm 2.96}$ | $34.29_{\pm 4.71}$ | $38.90_{\pm 8.86}$ | $21.65_{\pm 3.39}$ | $22.46_{\pm 4.05}$ |
| Gprompt | $63.78_{\pm 5.77}$ | $60.00_{\pm 6.18}$ | $66.68_{\pm 3.53}$ | $92.52_{\pm 5.38}$ | $39.00_{\pm 47.08}$ | $29.67_{\pm 2.53}$ | $73.92_{\pm 2.75}$ |
| All-in-one | $48.09_{\pm 4.83}$ | $48.09_{\pm 8.18}$ | $65.79_{\pm 5.79}$ | $89.62_{\pm 4.38}$ | $88.69_{\pm 1.08}$ | $24.23_{\pm 1.39}$ | $31.15_{\pm 2.25}$ |
| GPF | $34.84_{\pm 19.83}$ | $25.92_{\pm 12.30}$ | $71.20_{\pm 2.82}$ | $93.85_{\pm 3.71}$ | $95.47_{\pm 2.75}$ | $37.44_{\pm 3.43}$ | $59.67_{\pm 12.69}$ |
| GPF-plus | $56.38_{\pm 5.37}$ | $72.48_{\pm 5.63}$ | $70.85_{\pm 4.03}$ | $98.15_{\pm 0.73}$ | $97.66_{\pm 0.41}$ | $43.59_{\pm 4.52}$ | $64.63_{\pm 10.05}$ |
| MTG (Ours) | $66.11_{\pm 6.37}$ | $73.81_{\pm 8.56}$ | $71.38_{\pm 3.21}$ | $98.58_{\pm 0.93}$ | $98.17_{\pm 1.40}$ | $37.62_{\pm 4.72}$ | $76.01_{\pm 5.39}$ |

| Method | IMDB-B | COLLAB | PROTEINS | MUTAG | ENZYMES | COX2 | BZR | D&D |
|---|---|---|---|---|---|---|---|---|
| Supervised | $53.33_{\pm 6.61}$ | $50.77_{\pm 2.44}$ | $61.33_{\pm 2.89}$ | $59.47_{\pm 8.34}$ | $15.96_{\pm 1.64}$ | $65.15_{\pm 18.61}$ | $52.35_{\pm 8.12}$ | $59.77_{\pm 1.10}$ |
| Fine-tuning | $66.10_{\pm 0.70}$ | $56.10_{\pm 3.46}$ | $62.72_{\pm 2.39}$ | $59.87_{\pm 8.78}$ | $22.71_{\pm 0.86}$ | $69.97_{\pm 13.89}$ | $52.22_{\pm 10.64}$ | $59.70_{\pm 0.98}$ |
| GPPT | $59.48_{\pm 5.42}$ | $50.88_{\pm 6.31}$ | $64.74_{\pm 1.99}$ | $64.13_{\pm 18.31}$ | $19.12_{\pm 2.43}$ | $71.90_{\pm 14.28}$ | $70.93_{\pm 16.35}$ | $59.00_{\pm 6.34}$ |
| Gprompt | $64.35_{\pm 1.21}$ | $54.95_{\pm 9.47}$ | $64.94_{\pm 2.92}$ | $66.53_{\pm 14.84}$ | $22.08_{\pm 3.57}$ | $51.53_{\pm 13.08}$ | $54.63_{\pm 2.95}$ | $55.99_{\pm 7.53}$ |
| All-in-one | $65.67_{\pm 0.58}$ | $57.12_{\pm 1.99}$ | $69.84_{\pm 6.02}$ | $80.00_{\pm 5.67}$ | $23.96_{\pm 0.62}$ | $66.06_{\pm 18.23}$ | $61.98_{\pm 11.32}$ | $58.96_{\pm 5.93}$ |
| GPF | $65.97_{\pm 0.69}$ | $53.87_{\pm 3.44}$ | $63.35_{\pm 2.45}$ | $74.27_{\pm 1.55}$ | $23.87_{\pm 3.45}$ | $65.31_{\pm 19.45}$ | $74.38_{\pm 11.62}$ | $59.07_{\pm 0.65}$ |
| GPF-plus | $64.38_{\pm 2.30}$ | $56.50_{\pm 3.71}$ | $63.55_{\pm 1.85}$ | $75.20_{\pm 3.64}$ | $24.46_{\pm 2.27}$ | $65.25_{\pm 18.07}$ | $71.67_{\pm 14.87}$ | $59.51_{\pm 0.62}$ |
| MTG (Ours) | $66.95_{\pm 0.59}$ | $57.49_{\pm 2.52}$ | $70.49_{\pm 0.68}$ | $78.13_{\pm 6.36}$ | $29.71_{\pm 2.06}$ | $73.86_{\pm 9.74}$ | $74.65_{\pm 12.14}$ | $60.85_{\pm 6.39}$ |

*Table 6.* Performance comparison of adaptation methods on 5-shot node classification.

| Method | Cora | Citeseer | Pubmed | Wisconsin | Texas | Actor | ogbn-arxiv |
|---|---|---|---|---|---|---|---|
| Supervised | $50.25_{\pm 8.37}$ | $41.22_{\pm 6.30}$ | $67.88_{\pm 2.18}$ | $39.43_{\pm 5.86}$ | $43.91_{\pm 6.47}$ | $21.92_{\pm 1.86}$ | $22.38_{\pm 3.05}$ |
| Fine-tuning | $62.66_{\pm 3.55}$ | $39.54_{\pm 3.54}$ | $70.91_{\pm 4.87}$ | $42.97_{\pm 8.99}$ | $47.19_{\pm 7.37}$ | $22.92_{\pm 1.22}$ | $28.84_{\pm 3.11}$ |
| GPPT | $51.98_{\pm 3.43}$ | $45.77_{\pm 7.41}$ | $66.97_{\pm 3.70}$ | $37.00_{\pm 3.19}$ | $48.82_{\pm 5.15}$ | $21.58_{\pm 0.84}$ | $28.90_{\pm 1.64}$ |
| Gprompt | $69.03_{\pm 3.61}$ | $66.13_{\pm 1.64}$ | $67.87_{\pm 2.08}$ | $78.22_{\pm 37.33}$ | $39.32_{\pm 47.08}$ | $34.67_{\pm 1.28}$ | $85.40_{\pm 0.79}$ |
| All-in-one | $30.36_{\pm 13.48}$ | $27.93_{\pm 10.59}$ | $46.16_{\pm 15.83}$ | $87.16_{\pm 3.02}$ | $73.28_{\pm 9.91}$ | $21.49_{\pm 3.02}$ | $13.01_{\pm 6.29}$ |
| GPF | $35.43_{\pm 1.02}$ | $25.12_{\pm 3.01}$ | $68.96_{\pm 3.99}$ | $98.26_{\pm 1.19}$ | $98.42_{\pm 0.36}$ | $44.07_{\pm 3.94}$ | $71.83_{\pm 9.37}$ |
| GPF-plus | $66.22_{\pm 6.20}$ | $75.73_{\pm 2.19}$ | $69.59_{\pm 4.33}$ | $99.01_{\pm 1.43}$ | $99.12_{\pm 0.95}$ | $44.58_{\pm 5.95}$ | $66.88_{\pm 6.14}$ |
| MTG (Ours) | $71.81_{\pm 3.59}$ | $76.34_{\pm 6.18}$ | $70.84_{\pm 3.28}$ | $99.12_{\pm 0.95}$ | $98.76_{\pm 2.36}$ | $45.09_{\pm 3.26}$ | $85.94_{\pm 1.93}$ |

| Method | IMDB-B | COLLAB | PROTEINS | MUTAG | ENZYMES | COX2 | BZR | D&D |
|---|---|---|---|---|---|---|---|---|
| Supervised | $62.60_{\pm 4.01}$ | $55.23_{\pm 4.26}$ | $62.90_{\pm 5.03}$ | $73.47_{\pm 3.92}$ | $25.67_{\pm 0.48}$ | $64.99_{\pm 10.42}$ | $51.48_{\pm 2.29}$ | $63.59_{\pm 2.86}$ |
| Fine-tuning | $65.40_{\pm 3.33}$ | $60.72_{\pm 2.09}$ | $63.33_{\pm 4.13}$ | $75.33_{\pm 1.89}$ | $7.46_{\pm 1.29}$ | $73.19_{\pm 9.53}$ | $72.96_{\pm 11.98}$ | $64.71_{\pm 3.22}$ |
| GPPT | $66.37_{\pm 3.59}$ | $54.05_{\pm 4.58}$ | $58.27_{\pm 4.63}$ | $70.53_{\pm 3.90}$ | $22.17_{\pm 2.34}$ | $67.88_{\pm 17.34}$ | $69.63_{\pm 14.96}$ | $60.02_{\pm 3.24}$ |
| Gprompt | $66.70_{\pm 3.87}$ | $60.76_{\pm 5.08}$ | $62.94_{\pm 1.38}$ | $73.07_{\pm 2.13}$ | $21.46_{\pm 2.27}$ | $53.35_{\pm 7.75}$ | $59.38_{\pm 14.43}$ | $58.28_{\pm 2.18}$ |
| All-in-one | $63.62_{\pm 2.30}$ | $57.86_{\pm 5.88}$ | $71.37_{\pm 4.89}$ | $80.93_{\pm 1.96}$ | $26.71_{\pm 2.17}$ | $62.95_{\pm 8.57}$ | $62.78_{\pm 10.18}$ | $63.44_{\pm 1.35}$ |
| GPF | $67.80_{\pm 5.58}$ | $59.65_{\pm 6.25}$ | $63.37_{\pm 4.37}$ | $74.00_{\pm 3.65}$ | $27.00_{\pm 0.78}$ | $66.27_{\pm 14.57}$ | $61.05_{\pm 11.51}$ | $61.06_{\pm 2.63}$ |
| GPF-plus | $68.13_{\pm 3.31}$ | $60.68_{\pm 4.67}$ | $63.51_{\pm 2.89}$ | $73.87_{\pm 3.51}$ | $26.87_{\pm 1.89}$ | $72.87_{\pm 10.17}$ | $71.54_{\pm 14.81}$ | $64.80_{\pm 3.45}$ |
| MTG (Ours) | $69.15_{\pm 4.09}$ | $63.11_{\pm 1.88}$ | $70.10_{\pm 1.12}$ | $81.60_{\pm 4.53}$ | $35.08_{\pm 3.28}$ | $71.84_{\pm 2.75}$ | $76.37_{\pm 8.11}$ | $66.07_{\pm 2.39}$ |

*Table 7.* Performance comparison of Fine-tuning, GPF-plus, and MTG on 1-shot node classification. ↑/↓: positive/negative transfer vs. supervised learning baseline; **NTR** (Negative Transfer Rate): fraction of datasets with ↓ per daptation method.

| Pre-training | Adaptation | NTR | Cora | Citeseer | Pubmed | Wisconsin | Texas | Actor | ogbn-arxiv |
|---|---|---|---|---|---|---|---|---|---|
| - | Supervised | 0% | $26.56_{\pm5.55}$ (-) | $21.78_{\pm7.32}$ (-) | $39.37_{\pm16.34}$ (-) | $41.60_{\pm3.10}$ (-) | $37.97_{\pm5.80}$ (-) | $20.57_{\pm4.47}$ (-) | $10.99_{\pm3.19}$ (-) |
| DGI | Fine-tuning | 57% | $33.15_{\pm7.84}$ (↑) | $21.64_{\pm3.92}$ (↓) | $42.01_{\pm12.54}$ (↑) | $37.49_{\pm7.56}$ (↓) | $45.31_{\pm5.01}$ (↑) | $19.76_{\pm3.53}$ (↓) | $7.21_{\pm2.91}$ (↓) |
| | GPF-plus | 29% | $17.29_{\pm6.18}$ (↓) | $26.60_{\pm13.24}$ (↑) | $34.02_{\pm11.94}$ (↓) | $74.68_{\pm11.81}$ (↑) | $71.44_{\pm18.66}$ (↑) | $22.42_{\pm9.66}$ (↑) | $16.83_{\pm10.02}$ (↑) |
| | MTG | 0% | $49.48_{\pm4.82}$ (↑) | $62.31_{\pm18.90}$ (↑) | $46.18_{\pm7.32}$ (↑) | $67.72_{\pm10.19}$ (↑) | $62.96_{\pm16.80}$ (↑) | $25.48_{\pm7.33}$ (↑) | $25.06_{\pm10.57}$ (↑) |
| GraphMAE | Fine-tuning | 57% | $32.93_{\pm3.17}$ (↑) | $21.26_{\pm3.57}$ (↓) | $42.99_{\pm14.25}$ (↑) | $36.80_{\pm7.17}$ (↓) | $37.81_{\pm8.62}$ (↓) | $19.86_{\pm2.70}$ (↓) | $12.35_{\pm3.60}$ (↑) |
| | GPF-plus | 0% | $54.26_{\pm7.48}$ (↑) | $59.67_{\pm11.87}$ (↑) | $46.64_{\pm18.57}$ (↑) | $82.11_{\pm13.95}$ (↑) | $70.95_{\pm18.63}$ (↑) | $26.58_{\pm7.84}$ (↑) | $49.81_{\pm2.62}$ (↑) |
| | MTG | 0% | $46.27_{\pm6.66}$ (↑) | $49.21_{\pm12.95}$ (↑) | $46.98_{\pm10.02}$ (↑) | $83.32_{\pm12.46}$ (↑) | $71.59_{\pm18.67}$ (↑) | $29.44_{\pm7.31}$ (↑) | $36.44_{\pm9.59}$ (↑) |
| EdgePre-GPPT | Fine-tuning | 43% | $38.12_{\pm5.29}$ (↑) | $18.09_{\pm5.39}$ (↓) | $46.74_{\pm14.09}$ (↑) | $35.31_{\pm9.31}$ (↓) | $47.66_{\pm2.37}$ (↑) | $19.17_{\pm2.53}$ (↓) | $16.21_{\pm3.82}$ (↑) |
| | GPF-plus | 14% | $28.49_{\pm18.73}$ (↓) | $28.04_{\pm14.31}$ (↑) | $46.51_{\pm15.84}$ (↑) | $72.66_{\pm12.05}$ (↑) | $70.67_{\pm17.59}$ (↑) | $29.32_{\pm8.56}$ (↑) | $71.98_{\pm12.23}$ (↑) |
| | MTG | 0% | $46.68_{\pm2.66}$ (↑) | $33.22_{\pm12.52}$ (↑) | $44.85_{\pm9.75}$ (↑) | $73.80_{\pm9.56}$ (↑) | $71.11_{\pm17.13}$ (↑) | $20.96_{\pm2.93}$ (↑) | $75.97_{\pm4.29}$ (↑) |
| EdgePre-Gprompt | Fine-tuning | 14% | $35.57_{\pm5.83}$ (↑) | $22.28_{\pm3.80}$ (↑) | $41.50_{\pm7.54}$ (↑) | $40.69_{\pm4.13}$ (↓) | $40.62_{\pm7.95}$ (↑) | $20.74_{\pm4.16}$ (↑) | $14.83_{\pm2.38}$ (↑) |
| | GPF-plus | 0% | $55.77_{\pm10.30}$ (↑) | $49.43_{\pm8.21}$ (↑) | $42.79_{\pm18.18}$ (↑) | $78.76_{\pm13.63}$ (↑) | $68.75_{\pm16.51}$ (↑) | $22.68_{\pm3.64}$ (↑) | $57.44_{\pm6.95}$ (↑) |
| | MTG | 0% | $46.29_{\pm3.84}$ (↑) | $45.30_{\pm16.04}$ (↑) | $50.70_{\pm11.68}$ (↑) | $72.75_{\pm11.21}$ (↑) | $79.13_{\pm17.18}$ (↑) | $21.34_{\pm1.78}$ (↑) | $21.08_{\pm2.34}$ (↑) |
| GraphCL | Fine-tuning | 43% | $52.61_{\pm1.73}$ (↑) | $27.02_{\pm4.31}$ (↑) | $42.49_{\pm11.29}$ (↑) | $33.94_{\pm7.74}$ (↓) | $40.31_{\pm13.68}$ (↑) | $20.19_{\pm1.98}$ (↓) | $4.65_{\pm1.19}$ (↓) |
| | GPF-plus | 29% | $34.18_{\pm17.71}$ (↓) | $28.86_{\pm22.88}$ (↑) | $37.02_{\pm11.29}$ (↓) | $52.35_{\pm19.69}$ (↑) | $75.40_{\pm19.10}$ (↑) | $22.82_{\pm4.99}$ (↑) | $32.11_{\pm4.86}$ (↑) |
| | MTG | 0% | $58.54_{\pm7.89}$ (↑) | $50.96_{\pm16.40}$ (↑) | $40.00_{\pm7.80}$ (↑) | $48.41_{\pm16.10}$ (↑) | $69.71_{\pm16.42}$ (↑) | $24.77_{\pm8.45}$ (↑) | $38.96_{\pm6.82}$ (↑) |
| SimGRACE | Fine-tuning | 57% | $40.40_{\pm4.66}$ (↑) | $35.05_{\pm4.37}$ (↑) | $37.59_{\pm8.17}$ (↓) | $37.37_{\pm3.68}$ (↓) | $46.88_{\pm4.64}$ (↑) | $19.78_{\pm1.89}$ (↓) | $8.13_{\pm3.26}$ (↓) |
| | GPF-plus | 29% | $21.33_{\pm14.86}$ (↓) | $24.61_{\pm21.21}$ (↑) | $35.90_{\pm9.06}$ (↓) | $73.49_{\pm14.17}$ (↑) | $76.10_{\pm20.35}$ (↑) | $20.51_{\pm4.24}$ (↑) | $46.71_{\pm3.17}$ (↑) |
| | MTG | 0% | $45.93_{\pm7.67}$ (↑) | $57.60_{\pm9.01}$ (↑) | $43.29_{\pm10.80}$ (↑) | $72.98_{\pm9.75}$ (↑) | $73.17_{\pm16.68}$ (↑) | $22.03_{\pm3.59}$ (↑) | $37.90_{\pm5.83}$ (↑) |

*Table 8.* Performance comparison of Fine-tuning, All-in-one, and MTG on 1-shot graph classification.

| Pre-training | Adaptation | NTR | IMDB-B | COLLAB | PROTEINS | MUTAG | ENZYMES | COX2 | BZR | D&D |
|---|---|---|---|---|---|---|---|---|---|---|
| - | Supervised | 0% | $57.30_{\pm0.98}$ (-) | $47.23_{\pm0.61}$ (-) | $56.36_{\pm7.97}$ (-) | $65.20_{\pm6.70}$ (-) | $20.58_{\pm2.00}$ (-) | $27.08_{\pm11.94}$ (-) | $25.80_{\pm6.53}$ (-) | $55.33_{\pm6.22}$ (-) |
| DGI | Fine-tuning | 38% | $57.32_{\pm0.90}$ (↑) | $42.22_{\pm0.73}$ (↓) | $64.65_{\pm2.10}$ (↑) | $64.13_{\pm7.90}$ (↓) | $17.83_{\pm1.88}$ (↓) | $29.44_{\pm9.68}$ (↑) | $26.48_{\pm7.61}$ (↑) | $57.15_{\pm4.32}$ (↑) |
| | All-in-one | 13% | $60.07_{\pm4.81}$ (↑) | $39.56_{\pm5.00}$ (↓) | $62.58_{\pm7.07}$ (↑) | $73.87_{\pm6.13}$ (↑) | $23.96_{\pm1.45}$ (↑) | $50.72_{\pm9.93}$ (↑) | $64.38_{\pm9.32}$ (↑) | $55.97_{\pm6.52}$ (↑) |
| | MTG | 13% | $59.15_{\pm5.44}$ (↑) | $43.46_{\pm6.83}$ (↓) | $62.78_{\pm2.36}$ (↑) | $65.60_{\pm7.29}$ (↑) | $24.71_{\pm1.88}$ (↑) | $51.74_{\pm13.90}$ (↑) | $74.81_{\pm13.96}$ (↑) | $56.39_{\pm3.27}$ (↑) |
| GraphMAE | Fine-tuning | 0% | $57.70_{\pm1.13}$ (↑) | $48.10_{\pm0.23}$ (↑) | $63.57_{\pm3.57}$ (↑) | $65.20_{\pm5.00}$ (-) | $22.21_{\pm2.79}$ (↑) | $28.47_{\pm14.72}$ (↑) | $25.80_{\pm6.53}$ (-) | $57.54_{\pm4.41}$ (↑) |
| | All-in-one | 25% | $52.62_{\pm3.04}$ (↓) | $40.82_{\pm14.63}$ (↓) | $66.49_{\pm6.26}$ (↑) | $69.67_{\pm9.13}$ (↑) | $23.21_{\pm1.72}$ (↑) | $56.68_{\pm7.38}$ (↑) | $58.64_{\pm19.59}$ (↑) | $58.77_{\pm1.05}$ (↑) |
| | MTG | 0% | $58.10_{\pm5.72}$ (↑) | $48.24_{\pm9.56}$ (↑) | $59.62_{\pm6.41}$ (↑) | $66.93_{\pm7.03}$ (↑) | $22.71_{\pm2.58}$ (↑) | $58.93_{\pm12.05}$ (↑) | $54.07_{\pm18.34}$ (↑) | $58.01_{\pm5.85}$ (↑) |
| EdgePre-GPPT | Fine-tuning | 63% | $57.20_{\pm0.85}$ (↓) | $47.14_{\pm0.55}$ (↓) | $58.27_{\pm10.66}$ (↑) | $64.27_{\pm4.73}$ (↓) | $19.79_{\pm2.17}$ (↓) | $27.83_{\pm13.44}$ (↑) | $72.10_{\pm14.30}$ (↑) | $52.82_{\pm9.38}$ (↓) |
| | All-in-one | 13% | $59.12_{\pm0.77}$ (↑) | $42.74_{\pm4.65}$ (↓) | $65.71_{\pm5.49}$ (↑) | $75.20_{\pm6.33}$ (↑) | $20.92_{\pm2.04}$ (↑) | $60.27_{\pm16.97}$ (↑) | $59.69_{\pm9.90}$ (↑) | $56.24_{\pm2.46}$ (↑) |
| | MTG | 13% | $62.25_{\pm3.72}$ (↑) | $45.15_{\pm6.00}$ (↓) | $62.71_{\pm2.30}$ (↑) | $67.20_{\pm6.36}$ (↑) | $26.08_{\pm4.31}$ (↑) | $60.16_{\pm10.63}$ (↑) | $62.28_{\pm10.13}$ (↑) | $56.37_{\pm8.33}$ (↑) |
| EdgePre-Gprompt | Fine-tuning | 38% | $57.35_{\pm0.92}$ (↑) | $47.20_{\pm0.53}$ (↓) | $61.84_{\pm2.59}$ (↑) | $62.67_{\pm2.67}$ (↓) | $19.75_{\pm2.33}$ (↓) | $27.13_{\pm12.05}$ (↑) | $29.44_{\pm11.20}$ (↑) | $56.16_{\pm5.10}$ (↑) |
| | All-in-one | 25% | $53.78_{\pm2.82}$ (↓) | $42.87_{\pm6.19}$ (↓) | $61.82_{\pm7.53}$ (↑) | $68.27_{\pm3.88}$ (↑) | $21.88_{\pm0.56}$ (↑) | $49.06_{\pm5.53}$ (↑) | $32.65_{\pm10.08}$ (↑) | $57.60_{\pm4.37}$ (↑) |
| | MTG | 0% | $59.45_{\pm5.45}$ (↑) | $47.72_{\pm8.45}$ (↑) | $65.66_{\pm1.56}$ (↑) | $75.80_{\pm5.49}$ (↑) | $22.29_{\pm1.94}$ (↑) | $57.75_{\pm10.76}$ (↑) | $49.94_{\pm9.08}$ (↑) | $60.68_{\pm2.42}$ (↑) |
| GraphCL | Fine-tuning | 25% | $57.75_{\pm1.02}$ (↑) | $39.62_{\pm0.63}$ (↓) | $63.44_{\pm3.64}$ (↑) | $65.07_{\pm8.38}$ (↓) | $23.96_{\pm1.99}$ (↑) | $53.14_{\pm21.32}$ (↑) | $29.07_{\pm7.00}$ (↑) | $60.62_{\pm1.56}$ (↑) |
| | All-in-one | 13% | $58.75_{\pm0.80}$ (↑) | $51.66_{\pm2.60}$ (↑) | $66.00_{\pm8.79}$ (↑) | $66.00_{\pm8.79}$ (↑) | $19.46_{\pm2.85}$ (↓) | $52.55_{\pm13.51}$ (↑) | $42.65_{\pm14.43}$ (↑) | $59.72_{\pm1.52}$ (↑) |
| | MTG | 0% | $57.65_{\pm7.05}$ (↑) | $47.81_{\pm3.73}$ (↑) | $63.70_{\pm2.87}$ (↑) | $66.20_{\pm7.52}$ (↑) | $20.96_{\pm1.97}$ (↑) | $50.36_{\pm12.97}$ (↑) | $51.05_{\pm15.50}$ (↑) | $55.46_{\pm4.77}$ (↑) |
| SimGRACE | Fine-tuning | 38% | $57.33_{\pm0.96}$ (↑) | $46.89_{\pm0.42}$ (↓) | $60.07_{\pm3.21}$ (↑) | $65.47_{\pm5.89}$ (↑) | $19.71_{\pm1.76}$ (↓) | $76.19_{\pm5.41}$ (↑) | $28.48_{\pm6.49}$ (↑) | $53.23_{\pm9.71}$ (↓) |
| | All-in-one | 0% | $58.83_{\pm0.85}$ (↑) | $47.60_{\pm3.90}$ (↑) | $66.20_{\pm7.52}$ (↑) | $66.67_{\pm5.73}$ (↑) | $22.50_{\pm1.56}$ (↑) | $76.14_{\pm5.51}$ (↑) | $59.01_{\pm12.34}$ (↑) | $58.26_{\pm1.18}$ (↑) |
| | MTG | 0% | $61.82_{\pm3.49}$ (↑) | $52.25_{\pm0.56}$ (↑) | $66.98_{\pm2.17}$ (↑) | $68.87_{\pm5.01}$ (↑) | $21.33_{\pm1.92}$ (↑) | $78.27_{\pm2.01}$ (↑) | $65.68_{\pm16.41}$ (↑) | $57.26_{\pm2.01}$ (↑) |

*Table 9.* 1-shot node classification accuracy (%) on Wisconsin for various backbone models. Supervised learning baselines: GCN: $41.60_{\pm3.10}$, GAT: $34.51_{\pm18.02}$, GraphSAGE: $25.37_{\pm5.61}$, GIN: $28.91_{\pm11.51}$, GT: $20.91_{\pm7.07}$.

| | | | Fine-tuning | | | |
|---|---|---|---|---|---|---|
| Model | DGI | GraphMAE | EdgePreGPPT | EdgePreGprompt | GraphCL | SimGRACE |
| GCN | $37.49_{\pm5.13}$ | $36.80_{\pm7.17}$ | $35.31_{\pm9.31}$ | $\mathbf{40.69}_{\pm4.13}$ | $33.94_{\pm7.74}$ | $37.37_{\pm3.68}$ |
| GAT | $16.00_{\pm6.24}$ | $\mathbf{37.60}_{\pm10.69}$ | $20.00_{\pm3.82}$ | $33.37_{\pm4.76}$ | $18.86_{\pm1.88}$ | $28.00_{\pm9.40}$ |
| GraphSAGE | $40.69_{\pm9.46}$ | $\mathbf{43.77}_{\pm12.43}$ | $26.06_{\pm5.38}$ | $29.94_{\pm3.75}$ | $36.57_{\pm4.88}$ | $9.37_{\pm2.72}$ |
| GIN | $\mathbf{34.29}_{\pm10.40}$ | $26.29_{\pm7.81}$ | $25.14_{\pm7.70}$ | $33.49_{\pm7.69}$ | $22.63_{\pm8.62}$ | $16.46_{\pm4.53}$ |
| GT | $25.71_{\pm3.07}$ | $\mathbf{39.77}_{\pm8.42}$ | $23.20_{\pm2.65}$ | $28.23_{\pm6.64}$ | $11.77_{\pm1.06}$ | $14.51_{\pm5.08}$ |

| | | | GPPT | | | |
|---|---|---|---|---|---|---|
| Model | DGI | GraphMAE | EdgePreGPPT | EdgePreGprompt | GraphCL | SimGRACE |
| GCN | $29.94_{\pm10.40}$ | $29.83_{\pm9.34}$ | $23.89_{\pm5.40}$ | $\mathbf{30.40}_{\pm6.81}$ | $25.03_{\pm5.37}$ | $29.83_{\pm6.44}$ |
| GAT | $22.17_{\pm6.13}$ | $33.94_{\pm7.76}$ | $23.43_{\pm4.46}$ | $\mathbf{37.94}_{\pm7.11}$ | $26.86_{\pm6.12}$ | $29.83_{\pm8.04}$ |
| GraphSAGE | $26.51_{\pm8.00}$ | $\mathbf{30.51}_{\pm5.40}$ | $21.49_{\pm5.17}$ | $24.23_{\pm6.55}$ | $20.91_{\pm7.11}$ | $25.37_{\pm7.22}$ |
| GIN | $\mathbf{27.20}_{\pm5.34}$ | $24.00_{\pm3.29}$ | $21.14_{\pm1.84}$ | $20.46_{\pm2.79}$ | $19.09_{\pm14.19}$ | $19.54_{\pm15.85}$ |
| GT | $27.20_{\pm10.48}$ | $\mathbf{29.83}_{\pm5.80}$ | $28.00_{\pm6.01}$ | $23.31_{\pm3.01}$ | $27.66_{\pm0.69}$ | $25.03_{\pm5.43}$ |

| | | | Gprompt | | | |
|---|---|---|---|---|---|---|
| Model | DGI | GraphMAE | EdgePreGPPT | EdgePreGprompt | GraphCL | SimGRACE |
| GCN | $67.71_{\pm9.92}$ | $67.62_{\pm18.06}$ | $67.37_{\pm12.32}$ | $74.38_{\pm13.15}$ | $\mathbf{77.07}_{\pm5.93}$ | $65.38_{\pm13.70}$ |
| GAT | $58.25_{\pm13.83}$ | $67.77_{\pm15.91}$ | $\mathbf{94.17}_{\pm2.26}$ | $84.28_{\pm3.63}$ | $80.11_{\pm16.65}$ | $57.18_{\pm12.60}$ |
| GraphSAGE | $66.48_{\pm12.88}$ | $83.49_{\pm15.93}$ | $\mathbf{87.52}_{\pm3.79}$ | $82.16_{\pm2.64}$ | $65.50_{\pm6.48}$ | $72.61_{\pm5.97}$ |
| GIN | $45.47_{\pm9.62}$ | $37.72_{\pm15.00}$ | $58.36_{\pm15.10}$ | $59.29_{\pm12.72}$ | $59.03_{\pm19.98}$ | $\mathbf{71.80}_{\pm11.66}$ |
| GT | $56.03_{\pm7.33}$ | $73.50_{\pm9.72}$ | $76.97_{\pm13.39}$ | $\mathbf{80.07}_{\pm2.84}$ | $59.31_{\pm10.17}$ | $69.30_{\pm10.57}$ |

| | | | All-in-one | | | |
|---|---|---|---|---|---|---|
| Model | DGI | GraphMAE | EdgePreGPPT | EdgePreGprompt | GraphCL | SimGRACE |
| GCN | $56.02_{\pm13.12}$ | $57.54_{\pm10.66}$ | $\mathbf{66.29}_{\pm19.11}$ | $59.18_{\pm12.30}$ | $39.14_{\pm1.17}$ | $55.56_{\pm14.70}$ |
| GAT | $69.44_{\pm5.19}$ | $36.25_{\pm10.63}$ | $91.25_{\pm4.33}$ | $\mathbf{92.65}_{\pm3.75}$ | $42.85_{\pm9.16}$ | $36.61_{\pm14.86}$ |
| GraphSAGE | $74.88_{\pm19.77}$ | $87.55_{\pm3.78}$ | $98.60_{\pm0.87}$ | $\mathbf{99.12}_{\pm0.64}$ | $67.28_{\pm20.14}$ | $86.18_{\pm9.68}$ |
| GIN | $54.02_{\pm15.90}$ | $35.31_{\pm15.69}$ | $\mathbf{58.77}_{\pm13.43}$ | $57.07_{\pm12.51}$ | $45.94_{\pm9.52}$ | $25.30_{\pm14.83}$ |
| GT | $60.22_{\pm11.02}$ | $97.42_{\pm2.13}$ | $94.61_{\pm1.73}$ | $\mathbf{97.88}_{\pm2.24}$ | $51.33_{\pm15.56}$ | $83.26_{\pm16.29}$ |

| | | | GPF | | | |
|---|---|---|---|---|---|---|
| Model | DGI | GraphMAE | EdgePreGPPT | EdgePreGprompt | GraphCL | SimGRACE |
| GCN | $62.69_{\pm13.96}$ | $76.84_{\pm10.50}$ | $\mathbf{78.35}_{\pm4.07}$ | $75.20_{\pm13.22}$ | $51.60_{\pm20.06}$ | $60.81_{\pm26.52}$ |
| GAT | $65.14_{\pm11.94}$ | $74.39_{\pm16.46}$ | $\mathbf{94.96}_{\pm1.17}$ | $76.60_{\pm10.48}$ | $74.97_{\pm17.06}$ | $60.57_{\pm14.43}$ |
| GraphSAGE | $68.12_{\pm13.96}$ | $67.66_{\pm13.37}$ | $74.06_{\pm14.59}$ | $72.45_{\pm10.14}$ | $59.69_{\pm21.37}$ | $\mathbf{78.37}_{\pm14.84}$ |
| GIN | $47.11_{\pm11.28}$ | $49.47_{\pm14.94}$ | $\mathbf{66.99}_{\pm17.76}$ | $54.96_{\pm12.35}$ | $28.77_{\pm22.76}$ | $23.55_{\pm14.37}$ |
| GT | $39.85_{\pm4.83}$ | $71.26_{\pm14.43}$ | $72.67_{\pm13.36}$ | $\mathbf{81.33}_{\pm3.41}$ | $78.19_{\pm2.19}$ | $67.90_{\pm10.53}$ |

| | | | GPF-plus | | | |
|---|---|---|---|---|---|---|
| Model | DGI | GraphMAE | EdgePreGPPT | EdgePreGprompt | GraphCL | SimGRACE |
| GCN | $74.68_{\pm11.81}$ | $\mathbf{82.11}_{\pm13.95}$ | $72.66_{\pm12.05}$ | $78.76_{\pm13.63}$ | $52.35_{\pm19.69}$ | $73.49_{\pm14.17}$ |
| GAT | $93.34_{\pm6.13}$ | $83.28_{\pm12.20}$ | $\mathbf{95.24}_{\pm1.58}$ | $92.03_{\pm4.64}$ | $87.49_{\pm7.17}$ | $63.06_{\pm18.45}$ |
| GraphSAGE | $71.83_{\pm17.50}$ | $85.47_{\pm1.45}$ | $\mathbf{97.30}_{\pm1.68}$ | $80.35_{\pm14.37}$ | $50.35_{\pm8.91}$ | $71.95_{\pm9.43}$ |
| GIN | $57.55_{\pm16.90}$ | $66.88_{\pm14.55}$ | $\mathbf{82.79}_{\pm10.37}$ | $74.40_{\pm13.11}$ | $29.94_{\pm22.25}$ | $24.30_{\pm17.29}$ |
| GT | $72.41_{\pm11.37}$ | $\mathbf{95.19}_{\pm4.04}$ | $87.76_{\pm15.73}$ | $80.58_{\pm11.56}$ | $57.75_{\pm19.89}$ | $63.79_{\pm17.44}$ |

| | | | MTG (Ours) | | | |
|---|---|---|---|---|---|---|
| Model | DGI | GraphMAE | EdgePreGPPT | EdgePreGprompt | GraphCL | SimGRACE |
| GCN | $67.72_{\pm10.19}$ | $\mathbf{83.32}_{\pm12.46}$ | $73.80_{\pm9.56}$ | $72.75_{\pm11.21}$ | $48.41_{\pm16.10}$ | $72.98_{\pm9.75}$ |
| GAT | $59.87_{\pm9.77}$ | $82.16_{\pm11.33}$ | $\mathbf{95.84}_{\pm1.15}$ | $81.99_{\pm12.78}$ | $77.01_{\pm12.03}$ | $61.75_{\pm13.22}$ |
| GraphSAGE | $76.90_{\pm9.36}$ | $\mathbf{99.29}_{\pm1.41}$ | $87.23_{\pm4.91}$ | $72.63_{\pm10.16}$ | $62.44_{\pm19.82}$ | $63.50_{\pm17.62}$ |
| GIN | $59.93_{\pm13.84}$ | $69.96_{\pm10.90}$ | $78.87_{\pm16.32}$ | $\mathbf{83.57}_{\pm10.78}$ | $37.34_{\pm15.08}$ | $34.48_{\pm15.47}$ |
| GT | $58.35_{\pm10.12}$ | $\mathbf{98.85}_{\pm1.02}$ | $84.45_{\pm8.03}$ | $72.99_{\pm10.45}$ | $93.45_{\pm4.15}$ | $89.24_{\pm2.66}$ |

*Table 10.* 1-shot graph classification accuracy (%) on PROTEINS for various backbone models. Supervised learning baselines: GCN: $56.36_{\pm7.97}$, GAT: $48.34_{\pm9.96}$, GraphSAGE: $60.54_{\pm2.95}$, GIN: $59.66_{\pm1.12}$, GT: $61.44_{\pm2.48}$.

| | | | Fine-tuning | | | |
|---|---|---|---|---|---|---|
| Model | DGI | GraphMAE | EdgePreGPPT | EdgePreGprompt | GraphCL | SimGRACE |
| GCN | $60.00_{\pm4.48}$ | $62.40_{\pm1.494}$ | $58.27_{\pm10.66}$ | $61.84_{\pm2.59}$ | $\mathbf{63.44}_{\pm3.64}$ | $60.07_{\pm3.21}$ |
| GAT | $58.34_{\pm6.52}$ | $61.06_{\pm4.13}$ | $\mathbf{63.75}_{\pm3.71}$ | $54.09_{\pm4.03}$ | $60.04_{\pm3.06}$ | $58.65_{\pm6.71}$ |
| GraphSAGE | $60.70_{\pm4.08}$ | $60.56_{\pm5.12}$ | $61.60_{\pm1.78}$ | $\mathbf{63.21}_{\pm1.80}$ | $61.80_{\pm3.77}$ | $58.56_{\pm1.84}$ |
| GIN | $59.71_{\pm1.16}$ | $59.75_{\pm1.22}$ | $64.83_{\pm3.56}$ | $\mathbf{65.35}_{\pm2.36}$ | $58.52_{\pm0.77}$ | $58.49_{\pm0.80}$ |
| GT | $53.87_{\pm4.81}$ | $60.00_{\pm3.99}$ | $\mathbf{64.92}_{\pm3.19}$ | $56.58_{\pm3.28}$ | $62.88_{\pm1.82}$ | $60.00_{\pm1.60}$ |
| | | | GPPT | | | |
| Model | DGI | GraphMAE | EdgePreGPPT | EdgePreGprompt | GraphCL | SimGRACE |
| GCN | $60.81_{\pm1.55}$ | $60.72_{\pm1.70}$ | $\mathbf{60.92}_{\pm2.47}$ | $57.03_{\pm4.55}$ | $59.24_{\pm1.01}$ | $55.42_{\pm8.81}$ |
| GAT | $57.71_{\pm8.98}$ | $57.80_{\pm10.55}$ | $\mathbf{58.04}_{\pm9.92}$ | $54.97_{\pm7.45}$ | $52.29_{\pm7.83}$ | $55.15_{\pm9.84}$ |
| GraphSAGE | $56.56_{\pm6.73}$ | $57.73_{\pm7.95}$ | $\mathbf{58.63}_{\pm11.78}$ | $56.94_{\pm5.67}$ | $58.00_{\pm7.80}$ | $54.74_{\pm6.59}$ |
| GIN | $\mathbf{62.27}_{\pm2.54}$ | $52.13_{\pm11.00}$ | $52.52_{\pm6.97}$ | $55.53_{\pm8.92}$ | $55.78_{\pm7.22}$ | $55.78_{\pm7.22}$ |
| GT | $53.08_{\pm7.56}$ | $57.35_{\pm8.58}$ | $\mathbf{60.27}_{\pm3.92}$ | $55.51_{\pm7.68}$ | $56.18_{\pm5.79}$ | $55.87_{\pm7.69}$ |
| | | | Gprompt | | | |
| Model | DGI | GraphMAE | EdgePreGPPT | EdgePreGprompt | GraphCL | SimGRACE |
| GCN | $56.61_{\pm7.93}$ | $57.66_{\pm12.56}$ | $\mathbf{59.17}_{\pm11.26}$ | $55.55_{\pm8.17}$ | $55.51_{\pm10.73}$ | $57.53_{\pm11.05}$ |
| GAT | $61.08_{\pm6.19}$ | $63.03_{\pm2.61}$ | $\mathbf{64.47}_{\pm4.30}$ | $61.48_{\pm3.34}$ | $59.12_{\pm6.84}$ | $58.13_{\pm7.27}$ |
| GraphSAGE | $61.35_{\pm2.21}$ | $59.48_{\pm9.19}$ | $60.92_{\pm3.16}$ | $\mathbf{63.30}_{\pm1.43}$ | $55.26_{\pm2.61}$ | $63.21_{\pm2.66}$ |
| GIN | $54.36_{\pm5.18}$ | $46.97_{\pm11.45}$ | $55.82_{\pm5.35}$ | $\mathbf{57.84}_{\pm10.75}$ | $56.92_{\pm11.77}$ | $46.16_{\pm10.78}$ |
| GT | $56.65_{\pm5.81}$ | $60.99_{\pm1.62}$ | $\mathbf{61.87}_{\pm5.60}$ | $55.33_{\pm3.69}$ | $54.81_{\pm7.62}$ | $58.97_{\pm1.16}$ |
| | | | All-in-one | | | |
| Model | DGI | GraphMAE | EdgePreGPPT | EdgePreGprompt | GraphCL | SimGRACE |
| GCN | $62.58_{\pm7.07}$ | $\mathbf{66.49}_{\pm6.26}$ | $65.71_{\pm5.49}$ | $61.82_{\pm7.53}$ | $64.36_{\pm7.30}$ | $61.17_{\pm1.73}$ |
| GAT | $60.04_{\pm3.84}$ | $60.00_{\pm6.04}$ | $62.11_{\pm2.85}$ | $\mathbf{63.21}_{\pm2.22}$ | $58.36_{\pm4.93}$ | $59.37_{\pm5.59}$ |
| GraphSAGE | $59.53_{\pm4.94}$ | $60.70_{\pm4.89}$ | $63.12_{\pm1.59}$ | $59.98_{\pm8.46}$ | $62.22_{\pm3.81}$ | $62.04_{\pm2.07}$ |
| GIN | $\mathbf{61.55}_{\pm3.02}$ | $60.72_{\pm4.32}$ | $59.78_{\pm3.28}$ | $58.29_{\pm12.13}$ | $40.81_{\pm1.04}$ | $59.19_{\pm1.04}$ |
| GT | $57.39_{\pm3.66}$ | $58.92_{\pm6.61}$ | $62.61_{\pm4.08}$ | $60.20_{\pm7.55}$ | $\mathbf{62.81}_{\pm1.63}$ | $50.52_{\pm6.17}$ |
| | | | GPF | | | |
| Model | DGI | GraphMAE | EdgePreGPPT | EdgePreGprompt | GraphCL | SimGRACE |
| GCN | $59.17_{\pm3.63}$ | $58.65_{\pm8.49}$ | $62.54_{\pm2.55}$ | $61.82_{\pm2.61}$ | $\mathbf{63.91}_{\pm3.26}$ | $63.35_{\pm3.69}$ |
| GAT | $\mathbf{63.01}_{\pm1.22}$ | $59.62_{\pm5.38}$ | $47.53_{\pm9.42}$ | $47.71_{\pm7.14}$ | $56.65_{\pm5.15}$ | $57.91_{\pm3.10}$ |
| GraphSAGE | $52.72_{\pm6.43}$ | $59.17_{\pm2.22}$ | $61.73_{\pm2.59}$ | $\mathbf{64.54}_{\pm3.73}$ | $62.27_{\pm2.60}$ | $58.00_{\pm3.81}$ |
| GIN | $\mathbf{61.19}_{\pm3.39}$ | $54.34_{\pm8.61}$ | $60.58_{\pm6.80}$ | $62.34_{\pm1.19}$ | $59.19_{\pm1.04}$ | $59.19_{\pm1.04}$ |
| GT | $\mathbf{65.80}_{\pm7.42}$ | $60.16_{\pm5.81}$ | $64.54_{\pm7.18}$ | $61.21_{\pm2.91}$ | $58.74_{\pm5.51}$ | $59.57_{\pm2.93}$ |
| | | | GPF-plus | | | |
| Model | DGI | GraphMAE | EdgePreGPPT | EdgePreGprompt | GraphCL | SimGRACE |
| GCN | $61.26_{\pm3.06}$ | $62.49_{\pm2.05}$ | $\mathbf{63.06}_{\pm2.55}$ | $61.33_{\pm2.81}$ | $59.75_{\pm7.95}$ | $62.92_{\pm2.78}$ |
| GAT | $56.20_{\pm12.87}$ | $57.35_{\pm11.28}$ | $56.25_{\pm8.61}$ | $53.24_{\pm4.79}$ | $57.48_{\pm11.74}$ | $\mathbf{57.48}_{\pm9.63}$ |
| GraphSAGE | $56.22_{\pm9.08}$ | $57.55_{\pm10.56}$ | $56.31_{\pm9.26}$ | $\mathbf{57.71}_{\pm9.60}$ | $53.89_{\pm9.47}$ | $55.89_{\pm4.30}$ |
| GIN | $62.22_{\pm2.49}$ | $61.75_{\pm3.58}$ | $57.33_{\pm9.24}$ | $\mathbf{64.99}_{\pm0.82}$ | $59.19_{\pm1.04}$ | $59.19_{\pm1.04}$ |
| GT | $53.39_{\pm5.23}$ | $57.37_{\pm10.95}$ | $57.39_{\pm11.88}$ | $52.61_{\pm5.30}$ | $\mathbf{57.62}_{\pm12.27}$ | $56.16_{\pm5.07}$ |
| | | | MTG (Ours) | | | |
| Model | DGI | GraphMAE | EdgePreGPPT | EdgePreGprompt | GraphCL | SimGRACE |
| GCN | $62.78_{\pm2.36}$ | $59.62_{\pm6.41}$ | $62.71_{\pm2.30}$ | $\mathbf{65.66}_{\pm1.56}$ | $63.70_{\pm2.87}$ | $66.98_{\pm2.17}$ |
| GAT | $61.48_{\pm2.14}$ | $60.38_{\pm4.81}$ | $53.46_{\pm7.80}$ | $\mathbf{63.53}_{\pm1.25}$ | $49.12_{\pm6.49}$ | $52.63_{\pm4.14}$ |
| GraphSAGE | $61.98_{\pm2.03}$ | $58.85_{\pm1.66}$ | $65.24_{\pm1.83}$ | $\mathbf{65.88}_{\pm0.58}$ | $62.20_{\pm3.35}$ | $60.94_{\pm9.92}$ |
| GIN | $61.55_{\pm1.47}$ | $60.52_{\pm3.27}$ | $\mathbf{65.64}_{\pm6.34}$ | $63.10_{\pm0.39}$ | $60.19_{\pm1.04}$ | $59.69_{\pm3.21}$ |
| GT | $61.83_{\pm6.86}$ | $57.19_{\pm8.75}$ | $63.64_{\pm5.55}$ | $59.91_{\pm5.85}$ | $\mathbf{66.08}_{\pm2.70}$ | $61.72_{\pm0.83}$ |

*Table 11.* Performance comparison of adaptation methods on deep backbone models.

| Method | Cora (1-shot) | | | | BZR (1-shot) | | | |
|---|---|---|---|---|---|---|---|---|
| | $L = 4$ | $L = 8$ | $L = 12$ | $L = 16$ | $L = 4$ | $L = 8$ | $L = 12$ | $L = 16$ |
| Fine-tuning | $38.88_{\pm 6.74}$ | $36.84_{\pm 4.10}$ | $33.78_{\pm 6.05}$ | $30.70_{\pm 4.18}$ | $70.06_{\pm 18.37}$ | $56.17_{\pm 28.60}$ | $67.41_{\pm 23.52}$ | $71.11_{\pm 15.81}$ |
| GPPT | $30.68_{\pm 5.78}$ | $33.82_{\pm 1.99}$ | $33.89_{\pm 8.06}$ | $24.32_{\pm 4.60}$ | $68.95_{\pm 8.69}$ | $77.90_{\pm 23.15}$ | $67.59_{\pm 12.99}$ | $69.20_{\pm 14.51}$ |
| Gprompt | $38.53_{\pm 5.57}$ | $43.55_{\pm 4.87}$ | $41.42_{\pm 8.95}$ | $33.74_{\pm 4.10}$ | $67.04_{\pm 12.70}$ | $71.67_{\pm 7.01}$ | $76.60_{\pm 7.37}$ | $72.65_{\pm 7.40}$ |
| All-in-one | $29.42_{\pm 4.09}$ | $29.68_{\pm 6.53}$ | $26.02_{\pm 4.38}$ | $30.93_{\pm 4.38}$ | $61.23_{\pm 7.94}$ | $62.53_{\pm 10.26}$ | $69.32_{\pm 9.94}$ | $77.04_{\pm 2.93}$ |
| GPF | $33.84_{\pm 9.28}$ | $36.82_{\pm 13.61}$ | $28.12_{\pm 2.39}$ | $30.68_{\pm 3.43}$ | $75.74_{\pm 7.02}$ | $73.95_{\pm 10.60}$ | $72.59_{\pm 8.94}$ | $78.83_{\pm 0.75}$ |
| GPF-plus | $43.67_{\pm 9.52}$ | $41.34_{\pm 6.52}$ | $39.23_{\pm 7.88}$ | $36.08_{\pm 5.54}$ | $73.70_{\pm 3.14}$ | $76.79_{\pm 1.08}$ | $75.86_{\pm 6.77}$ | $71.73_{\pm 9.99}$ |
| MTG (Ours) | $\mathbf{47.80}_{\pm 7.07}$ | $\mathbf{45.10}_{\pm 7.90}$ | $\mathbf{43.36}_{\pm 6.54}$ | $\mathbf{37.00}_{\pm 5.66}$ | $\mathbf{77.04}_{\pm 9.93}$ | $\mathbf{79.26}_{\pm 0.45}$ | $\mathbf{79.26}_{\pm 0.45}$ | $\mathbf{79.26}_{\pm 0.45}$ |

*Table 12.* Computational efficiency comparison of graph prompt methods and message tuning.

| Method | ogbn-arxiv (1-shot) | | COLLAB (1-shot) | |
|---|---|---|---|---|
| | Time (s) | Memory (MB) | Time (s) | Memory (MB) |
| GPPT | 0.6032 | 3499 | 0.0204 | 32357 |
| Gprompt | 0.0326 | 10987 | 0.0081 | 3517 |
| All-in-one | 0.0559 | 11023 | 0.0147 | 3767 |
| GPF | 0.0067 | 10963 | 0.0045 | 3515 |
| GPF-plus | 0.0074 | 10983 | 0.0057 | 3517 |
| MTG (Ours) | **0.0053** | **10963** | **0.0036** | **3515** |

*Table 13.* Performance sensitivity to the number of message prototypes $m$ in MTG.

| Dataset | $m = 3$ | $m = 5$ | $m = 10$ | $m = 20$ | $m = 30$ |
|---|---|---|---|---|---|
| Cora (1-shot) | $51.28_{\pm 7.29}$ | $54.06_{\pm 4.49}$ | $58.54_{\pm 7.89}$ | $56.73_{\pm 5.51}$ | $56.21_{\pm 7.02}$ |
| BZR (1-shot) | $73.15_{\pm 15.26}$ | $77.84_{\pm 2.22}$ | $74.81_{\pm 13.96}$ | $77.53_{\pm 2.39}$ | $72.78_{\pm 17.86}$ |

