# OpenReview forum: "Message Tuning Outshines Graph Prompt Tuning: A Prismatic Space Perspective"
_ICML.cc/2026/Conference — ICML 2026 regular_

### Official Review · Reviewer_MNhi · 2026-03-10

**Soundness:** 2
**Presentation:** 3
**Significance:** 2
**Originality:** 3
**Overall Recommendation:** 4
**Confidence:** 4

**Summary:**

To address the open problem of how to rigorously characterize and measure the task-specific adaptation capacity of frozen GFMs, this paper proposes Prismatic Space Theory, which provides a geometric framework to formally analyze the expressive limitations of input-level graph prompt tuning and establishes a theoretical upper bound on its adaptation capability. Building upon this analysis, the authors introduce MTG, a parameter-efficient adaptation strategy that injects learnable prototype messages into each layer’s message-passing process, thereby directly intervening in the internal information flow and overcoming the representational constraints imposed by methods that operate solely in the input space. Experimental results demonstrate that MTG consistently outperforms existing graph prompt tuning approaches across multiple datasets and backbone models while maintaining favorable computational efficiency.

**Compliance With Llm Reviewing Policy:**

Affirmed.

**Final Justification:**

The authors’ response has addressed my concerns, and I will raise my score.

**Key Questions For Authors:**

1) The theoretical framework relies on the assumption that frozen GFM layers induce progressive contraction characterized by Jacobian rank reduction or intrinsic dimension decay. Could the authors provide empirical evidence to verify whether this contraction phenomenon indeed occurs in real backbone models?
2) MTG adopts a layer-wise parameter injection strategy that resembles adapter-style or prefix-style tuning paradigms. Could the authors elaborate on the structural and functional differences between MTG and existing PEFT approaches, both theoretically and empirically?
3) The experimental section demonstrates consistent improvements over prompt-based baselines, but it does not include comparisons with more recent or competitive adaptation strategies. Could the authors expand the evaluation to include additional state-of-the-art methods or more diverse tasks?
4) How does the theoretical analysis account for residual, normalization, and attention modules? Do the contraction results remain valid in such architectures?

**Limitations:**

The paper does not include an explicit discussion of limitations or broader impact. It would be beneficial for the authors to clarify the scope of the theoretical assumptions, and reflect on the practical boundaries where the intrinsic-dimension framework may not fully explain adaptation behavior.

**Strengths And Weaknesses:**

Strengths:
1) The work tackles adapting pre-trained graph foundation models, highlighting the limits of prompt-based adaptation and how message tuning can overcome them. It provides a novel theoretical perspective explaining why traditional input-level prompts are constrained under frozen backbones, which is inspiring for the design of message-level tuning strategies.
2) The manuscript is clearly written and well-organized, with a logical flow from theoretical foundations to method design and experimental evaluation.

Weaknesses:
1) The theoretical analysis relies heavily on modeling each layer as a piecewise linear map and characterizing contraction via Jacobian rank. However, modern GFMs typically include residual connections, LayerNorm and other nonlinear components. The paper does not clearly specify whether these elements are incorporated into the formal analysis, nor does it empirically verify the assumed “layer-wise contraction” phenomenon in real backbone models. This creates a potential gap between theory and practice.
2) The MTG mechanism appears to be an incremental architectural extension, as its layer-wise parameter injection strategy resembles adapter-style or prefix-style parameter-efficient tuning approaches.
3) The experimental evaluation is limited and insufficient, and it does not include comparisons with the latest methods.
4) The theoretical analysis emphasizes layer-wise contraction and the decay of prompt influence; however, the experimental section does not measure the Jacobian spectra, analyze how prompt effects vary across different network depths, or verify whether the theoretically predicted phenomena manifest in practice.

---

> ### Author Rebuttal · Authors · 2026-03-31
>
> Dear Reviewer MNhi,
>
> **Thank you so much for your careful review and valuable suggestions on our work.** We appreciate your recognition that the theoretical perspective of PS-Theory is novel.
>
> Your concerns have been carefully addressed below.
>
> > W1: modern GFMs typically include residual connections, LayerNorm and other nonlinear components. The paper does not clearly specify whether these elements are incorporated into the formal analysis.
>
> A: Our analysis targets GNN-based GFMs that can be abstracted by the unified layer in Section 3.1 (attention/message fusion/update).We have clarified in lines 133–136 that this abstraction only captures the core structure rather than all architectural details. Residual connections are linear additions and thus do not conflict with the piecewise-linear composition view. LayerNorm and other nonlinear components can be incorporated into the theoretical framework using the approximation method mentioned in Remark B.8. We also note that existing theoretical analyses of graph prompt tuning rely on assumptions [1]; compared with them, we formalize a unified GNN-based GFM layer and introduce PS-Theory to quantify adaptation capacity via intrinsic dimension, Hausdorff measure, and diameter of the induced prismatic space, which enables an explicit prompt efficacy bound. Our work tries to advance the theoretical understanding of graph prompt tuning in a more rigorous direction.
>
> > W2: The MTG mechanism appears to be an incremental architectural extension, as its layer-wise parameter injection strategy resembles adapter-style or prefix-style parameter-efficient tuning approaches.
>
> A: We have discussed the key difference between MTG and prefix-tuning in lines 280-297. Structurally, MTG injects learnable prototypes into each layer’s message pathway and fuses them with native messages before message passing, while keeping all pretrained backbone parameters frozen. Functionally, this creates layer-wise controllability over message fusion, rather than adding generic transformation capacity on hidden states (adapters) or adding input-like context that must be propagated by a fixed attention mechanism (prefix-style tuning).
>
> > W3: The experimental evaluation is limited and insufficient, and it does not include comparisons with the latest methods.
>
> A: ProG (we use) is currently the most influential benchmark in the field of graph prompt, and its GitHub repository has garnered over 500 stars (https://github.com/sheldonresearch/ProG). We conduct extra 5-shot evaluations on six representative datasets (Cora, Citeseer, Pubmed, ENZYMES, PROTEINS, BZR), comparing MTG against EdgePrompt [2], EdgePrompt-Plus [2], and an adapted adapter baseline. We will include these additional results in the revision.
>
> | Method / Dataset (5-shot) | Cora | Citeseer | Wisconsin | ENZYMES | PROTEINS | BZR |
> |---|---:|---:|---:|---:|---:|---:|
> | EdgePrompt | 68.21±2.99 | 71.24±3.84 | 93.29±2.07 | 30.28±4.26 | 68.24±1.49 | 73.46±7.42|
> | EdgePrompt-Plus| 66.84±2.56 | 73.43±2.86 | 97.72±1.51 | 31.73±2.88 | 69.58±1.05 | 71.35±8.23|
> |Adapter (adapted by us) | 64.33±5.29 | 72.19±3.45 | 87.09±5.74 | 29.32±5.10 | 66.37±1.65 | 70.33±7.42|
> | MTG (ours) | 71.81±3.59 | 76.34±6.18 | 99.12±0.95 | 35.08±3.28 | 70.10±1.12 | 76.37±8.11|
>
> > W4: the experimental section does not measure the Jacobian spectra, analyze how prompt effects vary across different network depths.
>
> A4: The Jacobian spectra in the PS-Theory is a theoretical analysis technique that cannot be accurately calculated in actual models. We have analyzed in the Appendix F.2 and Table 11 how the prompt effects vary across different network depths through experiments.
>
> > Q1: empirical evidence to verify whether this contraction phenomenon indeed occurs in real backbone models?
>
> A: The details in W4-A might address your concerns.
>
> > Q2: the structural and functional differences between MTG and existing PEFT approaches?
>
> A: The details in W2-A might address your concerns.
>
> > Q3: additional state-of-the-art methods or more diverse tasks?
>
> A: The details in W3-A might address your concerns.
>
> > Q4: theoretical analysis account for residual, normalization, and attention modules?
>
> A: The details in W1-A might address your concerns.
>
> > Limitations: clarify the scope of the theoretical assumptions.
>
> A: Thank you for this helpful suggestion. We will add a Limitations subsection in the revision, which clarifies that PS-Theory is more of a theoretical perspective that serves to inspire and guide the design of adaptation methods.
>
> **Once again, we deeply appreciate your valuable comments and thoughtful efforts.** We sincerely hope our clarification has addressed your concerns and look forward to your response.
>
> [1] Qunzhong Wang, Xiangguo Sun, and Hong Cheng. Does graph prompt work? A data operation perspective with theoretical analysis. In ICML, 2025.
>
> [2] Xingbo Fu, Yinhan He, and Jundong Li. Edge Prompt Tuning for Graph Neural Networks. In ICLR, 2025.

---

> > ### Author Rebuttal · Reviewer_MNhi · 2026-04-01
> >
> > The authors’ response has addressed my concerns, and I will raise my score.

---

> > > ### Author Response · Authors · 2026-04-01
> > >
> > > Thank you for your encouraging feedback! We are pleased to hear that our rebuttal has effectively addressed your concerns. We thank you for taking the time to reassess the manuscript and adjust the score accordingly.

---

### Official Review · Reviewer_JsU5 · 2026-03-10

**Soundness:** 2
**Presentation:** 2
**Significance:** 2
**Originality:** 2
**Overall Recommendation:** 2
**Confidence:** 4

**Summary:**

This paper studies parameter-efficient adaptation for GNN-based graph foundation models. It argues that, while graph prompt tuning has become a dominant adaptation strategy, its adaptation capacity has not been rigorously characterized. To address this, the paper proposes Prismatic Space Theory (PS-Theory), a geometric framework intended to quantify the adaptation capacity of prompt-based methods and to establish an upper bound for graph prompt tuning. Motivated by this analysis, the paper further introduces Message Tuning for GFMs (MTG), which injects a small set of learnable message prototypes into each layer of the frozen GNN backbone to guide message fusion without updating pretrained weights. The paper provides theoretical arguments that MTG can exceed the adaptation capacity bound of graph prompt tuning, and empirically evaluates the method across multiple few-shot downstream graph tasks, showing competitive performance relative to graph prompt baselines and full fine-tuning.

**Compliance With Llm Reviewing Policy:**

Affirmed.

**Final Justification:**

I appreciate the authors' further clarification regarding the GFM-specific design. However, I remain of the view that the layer-wise message injection mechanism does not represent a sufficient conceptual leap in methodological novelty, as my core concerns detailed in Weakness 3 and Key Question 4 persist. Since this is a fundamental judgment on the work’s distinctiveness rather than a matter of research scope, it is not something that can be resolved through further discussion, and I will maintain my original assessment.

**Key Questions For Authors:**

1. How essential is the PS-Theory framework to the empirical success of MTG?
The paper presents MTG as being theoretically motivated by PS-Theory, but the empirical section mainly shows that MTG performs well. Can the authors provide stronger evidence that the theory is not just a post hoc interpretation, but actually explains why MTG works better than graph prompt tuning in practice?

2. How robust is MTG to architectural and hyperparameter choices?
Since MTG injects message prototypes into each layer, its behavior may depend on the number of prototypes, insertion strategy, fusion rule, and backbone depth. The paper includes some sensitivity analysis, but could the authors clarify whether MTG remains consistently strong under a wider range of realistic settings, especially when model scale increases?

3. How should readers interpret the practical meaning of the proposed "adaptation capacity" metric?
The paper uses geometric quantities such as measure, intrinsic dimension, and diameter of representation spaces to characterize adaptation capacity. However, the connection between these quantities and downstream generalization or optimization quality is not fully clear. Can the authors clarify in what sense these quantities should be viewed as predictive of real adaptation performance, rather than as a stylized theoretical proxy?

4. Why should MTG be viewed as a distinct adaptation paradigm rather than an intermediate-layer variant of existing parameter-efficient tuning ideas?
The method is intuitive and practically appealing, but conceptually it seems related to inserting trainable intermediate representations while keeping the backbone frozen. Can the authors better articulate what makes MTG fundamentally different from prior lightweight adaptation mechanisms beyond graph prompt tuning?

5. How general are the conclusions beyond the specific benchmark setting used in the paper?
The evaluation is mainly conducted in the ProG-style few-shot benchmark setting for node and graph classification. Can the authors discuss how confidently the conclusions should transfer to other graph tasks, larger-scale datasets, or other classes of graph foundation models?

**Limitations:**

- The paper overclaims in places.
Phrases such as "rigorous mathematical framework", "upper bound", and "outshines graph prompt tuning" are presented quite strongly, but the actual evidence feels less definitive than the wording suggests. In particular, the theory appears to provide a stylized geometric view rather than a clearly validated or tight bound for real-world adaptation behavior.

- The theory relies on strong assumptions that should be acknowledged more explicitly.
The analysis depends on assumptions such as compact smooth input manifolds, piecewise linearity, local injectivity on partition cells, and geometric quantities derived from Jacobians. These assumptions may be mathematically convenient but are far from obviously satisfied in realistic graph learning settings, and the paper should state this more clearly as a limitation.

- The empirical evaluation emphasizes best-case performance more than rigorous statistical comparison.
Section 5.2 explicitly frames results as "upper bound performance" by selecting the best results over different pre-training strategies. While this is one way to summarize performance ceilings, it can also make comparisons look stronger than they are in a more controlled apples-to-apples setting. I would have preferred more emphasis on consistent gains under fixed settings and stronger discussion of variance/significance beyond averages over five runs.

- The scalability and implementation trade-offs are under-discussed.
Although the paper claims efficiency and provides some efficiency analysis, the method introduces learnable parameters at every layer and evaluates across many settings. A more explicit discussion of computational and memory overhead under larger backbones or larger-scale deployment would strengthen the paper.

**Strengths And Weaknesses:**

Strengths

- The paper tackles an important problem in graph foundation model adaptation.
The work focuses on parameter-efficient adaptation for GNN-based graph foundation models, which is a relevant topic given the growing interest in graph pre-training and lightweight downstream adaptation. The paper is well motivated by the limitations of full fine-tuning and by the current prominence of graph prompt tuning.

- The proposed method is conceptually simple and practically lightweight.
MTG injects a small number of learnable message prototypes into each layer while freezing pretrained backbone weights. This is an intuitive design that is easy to understand at a high level and appears broadly compatible with different GNN-based backbones.

- The paper makes an effort to provide theoretical grounding.
A substantial portion of the paper is devoted to proposing PS-Theory as a framework for reasoning about the adaptation capacity of graph prompt tuning and message tuning. Even if one may debate the usefulness or tightness of the framework, the authors clearly aim to go beyond purely empirical method design.

Weaknesses

- The central theoretical framework is ambitious but not fully convincing.
The paper introduces PS-Theory as a rigorous framework for quantifying adaptation capacity, but many of the assumptions and abstractions seem quite strong relative to the actual models and datasets used in practice. For example, the analysis depends heavily on piecewise linear formulations, manifold assumptions, Jacobian singular values, and injectivity conditions on partition cells. I am not convinced that this framework yields a practically meaningful or tight characterization of adaptation capacity for real graph foundation models.

- The notion of “adaptation capacity” remains somewhat abstract and loosely connected to downstream performance.
The paper argues that larger representation-space measure, intrinsic dimension, or diameter corresponds to stronger adaptation capacity, and uses this to justify the superiority of MTG over graph prompt tuning. However, the link between these geometric quantities and actual task generalization or optimization outcomes is not sufficiently established. As a result, the main theoretical claim feels more suggestive than definitive.

- The novelty of the method itself feels moderate.
Once the theoretical framing is set aside, MTG can be viewed as injecting trainable message-level parameters into intermediate layers while keeping pretrained weights frozen. This is a reasonable and potentially useful design, but it does not strike me as highly surprising at the method level, especially given the analogy the paper itself draws to prefix-style adaptation at multiple layers.

- It is unclear whether the theory meaningfully explains the empirical gains.
The paper claims that MTG outperforms graph prompt tuning because it exceeds the theoretical upper bound of graph prompt tuning’s adaptation capacity. However, the experiments mainly show empirical improvement, not a direct validation that the proposed geometric quantities are what drive those gains. The theory and experiments feel somewhat parallel rather than tightly integrated.

---

> ### Author Rebuttal · Authors · 2026-03-31
>
> Dear Reviewer JsU5,
>
> **Thank you very much for your review and questions.** We sincerely hope that the following clarification will be helpful in resolving any potential misunderstandings.
>
> > W1 &W2 & L1 & L2: Concerns regarding the theoretical framework.
>
> A: Thank you for raising these concerns regarding the theory. Both reviewers xcpy and MNhi have recognized the novelty of PS-Theory. First, theoretical analysis in graph prompt tuning necessarily operates under explicit abstractions/assumptions to yield tractable insights. Our analysis focuses on GNN-based GFMs that can be abstracted by the unified layer described in Section 3.1 (attention/message fusion/update); accordingly, some modeling assumptions are required. Second, the problems that PS-Theory aims to address are rigorously formalized in Section 2, and our analysis builds upon prior theoretical studies of graph prompt tuning [1]. In the revision, we will further clarify the scope and assumptions and better connect them to practical architectures. We also note that existing theoretical analyses of graph prompt tuning rely on some assumptions [1]. Our work tries to advance the theoretical understanding of graph prompt tuning in a more rigorous direction.
>
> > W3 & Q4: Novelty of the method.
>
> A: Guided by theoretical analysis, the design concept of our method is lightweight, user-friendly and efficient. We acknowledge that prefix-tuning has provided us with some inspiration for our method design, which is elaborated in Section 4.1. But we also have discussed the key difference between MTG and prefix-tuning in lines 280-297. MTG is a plug-and-play method specifically designed for GNN backbones, not a simple transfer of prefix-tuning from LLMs to GNN-based GFMs.
>
> > W4 & Q1: The relationship between theory and method.
>
> A: MTG is motivated by PS-Theory, which suggests that input-space prompt perturbations are fundamentally constrained by the compositional prismatic effect of a frozen backbone. Building on the design rationale in Section 4, MTG parameterizes adaptation by introducing a small set of learnable message prototypes at each layer and learning a dynamic fusion mechanism that modulates the message-passing across layers, while keeping all pretrained backbone parameters frozen. This is substantially different from transformer prefix-tuning: rather than prepending fixed virtual tokens and relying on attention to propagate their influence, MTG explicitly augments the graph message-passing operator by lightweight, layer-wise learnable components.
>
> > Q2: How robust is MTG to architectural and hyperparameter choices?
>
> A: We evaluate MTG across multiple backbones and depths (Appendix F.2), and we include sensitivity analysis with respect to the number of message prototypes (m) (Appendix F.4). MTG shows stable performance and strong robustness, including markedly reduced negative transfer (Section 5.4). We will make the robustness/sensitivity experiments more explicit in the revision.
>
> > Q3: How should readers interpret the practical meaning of the proposed "adaptation capacity" metric?
>
> A: In PS-Theory, “adaptation capacity” is defined as a geometric characterization of the reachable final representation space under a frozen backbone, using intrinsic dimension, Hausdorff measure, and diameter of the induced prismatic space (Section 3.2-3.3). These quantities are intended to quantify the expressive range that an adaptation method can induce in the frozen model’s output space, rather than to directly predict task loss or generalization. This is not a metric during the actual model training process. Instead, it is used in theoretical analysis to measure the output space (RQ3 in Section 2).
>
> > Q5: Concerns regarding the benchmark.
>
> A: ProG (we use) is currently the most influential benchmark in the field of graph prompt, and its GitHub repository has garnered over 500 stars (https://github.com/sheldonresearch/ProG). Our experimental setup has been aligned with other research works in the same field.
>
> > L3 & L4: Concerns regarding the experimental evaluation.
>
> A: All our evaluations are in line with those in ProG, including the best results obtained through different pre-training strategies. We repeat sampling five times and report average and standard deviation over these five results. Due to the page limit of the main text, the variance is not shown in Table 1. The detailed experimental results (mean and variance) are presented in Tables 4-6. Tables 7 and 8 also present the impact of different pre-training strategies on MTG and their comparison with the baselines of graph prompt tuning.
>
> [1] Qunzhong Wang, Xiangguo Sun, and Hong Cheng. Does graph prompt work? A data operation perspective with theoretical analysis. In ICML, 2025.
>
> **Once again, we deeply appreciate your valuable comments and thoughtful efforts.** We sincerely hope our clarification has addressed your concerns and look forward to your response.

---

> > ### Author Rebuttal · Reviewer_JsU5 · 2026-04-03
> >
> > Thank you for the detailed and thoughtful rebuttal. While I appreciate the clarifications provided, I intend to maintain my score. My primary reservation regarding the methodological novelty, as detailed in Weaknesses 3, Key Question 4, and Limitation 1, remains. As this concern relates to the core nature of the method itself, I believe it is not something that can be resolved within the scope of a short rebuttal. Therefore, I will maintain my original assessment and score.

---

> > > ### Author Response · Authors · 2026-04-04
> > >
> > > Dear Reviewer JsU5,
> > >
> > > Thank you very much for your reply. We further clarify the methodological novelty below, making our best effort to address your concerns.
> > >
> > > We respectfully suggest that your concerns regarding the methodological novelty may stem from a potential confusion between the adaptation methods of GFMs and existing parameter-efficient fine-tuning methods for large language models (LLMs). Our method MTG is specifically designed for GNN-based GFMs, and the graph prompt tuning methods we compare against are also the same. **This is not within the same research scope as the PEFT of LLMs.** We provide a detailed explanation from the following three aspects.
> > >
> > > 1. **Grounded in a profound geometric theoretical insight.** Unlike existing graph prompt tuning methods that rely on empirical design or simplistic data operation analysis, MTG is grounded in the newly proposed Prismatic Space Theory (PS-Theory). For the first time, it proves from a geometric measure theory perspective that graph prompt tuning has a theoretical upper bound on adaptation capacity, whereas layer-wise message injection can surpass this bound, providing rigorous theoretical guarantees for the method.
> > >
> > > 2. **A novel layer-wise message tuning mechanism specifically designed for GNN-based GFMs.** Graph prompt tuning only adds perturbations in the input space and cannot intervene in the model's internal message passing. In contrast, MTG injects learnable message prototypes at every layer and dynamically fuses them with the model's native messages via an attention mechanism, directly acting on the message passing process. This effectively alleviates the information contraction and rank collapse caused by frozen GFM layers.
> > >
> > > 3. **Lightweight and backbone-agnostic plug-and-play design.** MTG introduces only a small number of trainable parameters per layer (message prototypes and a projection matrix) without updating any pre-trained weights, enabling efficient plug-and-play adaptation. It is compatible with various backbone architectures, including MPNNs and Graph Transformers, and experiments demonstrate its robustness, high computational efficiency, and ability to mitigate negative transfer.
> > >
> > > Therefore, MTG is not only different from **graph prompt tuning** (which only adds learnable perturbations to the input space), but also differs from **adapter-style** (which inserts trainable modules between frozen layers) or **prefix-style** (which prepends learnable context vectors to each layer’s input) parameter-efficient tuning approaches. Furthermore, the contribution of our work goes beyond merely introducing new adaptation methods. Instead, our work provides **a novel theoretical analysis framework** and **a deeper theoretical understanding of graph prompt tuning**.
> > >
> > > In short, the novelty of MTG lies in **(1) introducing a GFM-specific, layer-wise, dynamic message-fusion mechanism motivated by PS-Theory**, and **(2) demonstrating both theoretically and empirically that this mechanism defines a strictly more expressive adaptation paradigm than graph prompt tuning**.
> > >
> > > We hope our clarifications are helpful, and we sincerely thank you again for the time and effort you have devoted to reviewing our manuscript.

---

### Official Review · Reviewer_xcpy · 2026-03-13

**Soundness:** 3
**Presentation:** 3
**Significance:** 3
**Originality:** 3
**Overall Recommendation:** 5
**Confidence:** 4

**Summary:**

This paper addresses the problem of measuring the adaptation capacity of graph prompt tuning for graph foundation models (GFMs). The authors introduce a mathematical framework Prismatic Space Theory (PS-Theory) to quantify the capacity of adaptation methods and propose Message Tuning for GFMs (MTG) by adding small learnable message prototypes at each GNN layer. The authors prove both theoretically and experimentally that MTG adapts better than graph prompt tuning.

**Compliance With Llm Reviewing Policy:**

Affirmed.

**Final Justification:**

I maintain my score after the author rebuttal.

**Key Questions For Authors:**

1. Theoretical analysis involves certain assumptions and approximations. What are the limitations on the applicable range of PS-Theory?

2. Is MTG plug-and-play? Are there any limitations on the types of graph models MTG can work with?

3. In section 5.4, the authors conducted experiments related to negative transfer. Can PS-Theory provide a theoretical explanation for negative transfer?

**Limitations:**

It is recommended that the authors add a section discussing the limitations of PS-Theory and MTG. This addition would improve the overall coherence of the paper.

**Strengths And Weaknesses:**

Strengths

1. PS-Theory appears novel, addressing a critical theoretical problem in the field of graph foundation models. It provides theoretical insights that will inspire the design of future adaptation methods.

2. Introducing analytical techniques from learning theory into the domain of graph foundation models is a promising endeavor, offering a new perspective for theoretical advancements in this area.

3. the design of MTG is elegant, lightweight, and efficient. Inspired by PS-Theory, MTG benefits from theoretical guarantees while also demonstrating strong empirical results.

4. the paper is well-written and includes complete code for reproducibility.

Weaknesses

1. The mathematical definitions, concepts and symbols introduced in the PS-Theory are too numerous, making the reading somewhat obscure. The authors should further optimize the theoretical part to lower the reading barrier.

2. PS-Theory and MTG are both limited to GNN-based GFMs and have difficulty covering a wider range of GFMs (such as those with LLM as the backbone).

3. Some of the Figures (such as Figure 3) have small fonts, which should be optimized.

---

> ### Author Rebuttal · Authors · 2026-03-31
>
> Dear Reviewer xcpy,
>
> **We sincerely appreciate your time and expertise in evaluating our work.** We appreciate your recognition of the novelty of PS-Theory and the elegance of MTG.
>
> Your concerns have been carefully addressed below.
>
> > W1: The mathematical definitions, concepts and symbols introduced in the PS-Theory are too numerous, making the reading somewhat obscure. The authors should further optimize the theoretical part to lower the reading barrier.
>
> A: Thank you for this feedback. We provided a Reading Guideline and a notation table in appendix to support readers. In the revised version, we will add additional theoretical background and more detailed explanations of symbols to further improve readability.
>
> > W2: PS-Theory and MTG are both limited to GNN-based GFMs and have difficulty covering a wider range of GFMs (such as those with LLM as the backbone).
>
> A: We agree that PS-Theory and MTG are developed for GNN-based GFMs and do not aim to cover LLM-backbone GFMs in this work. We state this focus explicitly in the second sentence of the Introduction (“As an important category of GFMs, GNN-based GFMs …”). More concretely, our theoretical study follows the graph prompt analysis of Wang et al. [1]. We will make this scope clearer in the revision.
>
> > W3: Some of the Figures (such as Figure 3) have small fonts, which should be optimized.
>
> A: Thank you very much for pointing this out. We will optimize the charts in the revised version to make them clearer and more attractive.
>
> > Q1: Theoretical analysis involves certain assumptions and approximations. What are the limitations on the applicable range of PS-Theory?
>
> A: The applicable range of the PS-Theory encompasses GNN-based GFMs that can perform modeling abstraction as described in Section 3.1 and Appendix B.2. We have clarified in lines 133–136 that this abstraction only captures the core structure rather than all architectural details. We further discuss the required approximations of the proof of Proposition 3.3 in Remark B.8. The problems in Section 2 (RQ1-RQ3) are inherently complex and difficult, so these assumptions and approximations are necessary compromises under the current theoretical techniques. We also note that existing theoretical analyses of graph prompt tuning rely on assumptions [1]; compared with them, we formalize a unified GNN-based GFM layer and introduce PS-Theory to quantify adaptation capacity via intrinsic dimension, Hausdorff measure, and diameter of the induced prismatic space, which enables an explicit prompt efficacy bound. Our work tries to advance the theoretical understanding of graph prompt tuning in a more rigorous direction. We will add a paragraph to explain these in the revision.
>
> > Q2: Is MTG plug-and-play? Are there any limitations on the types of graph models MTG can work with?
>
> A: Yes. Guided by theoretical analysis, the design concept of our method is lightweight, user-friendly and efficient. Detailed design insights can be found in Section 4.2. As long as there are layer-by-layer accessible feature interfaces, MTG can be used as a plugin. The GNN-based GFMs discussed in this article, whether it is MPNN or GT backbone, can all be used by MTG. Detailed experiments are presented in Appendix F.2. For more complex graph models, the access of MTG may need to be slightly modified in combination with the specific architecture.
>
> > Q3: In section 5.4, the authors conducted experiments related to negative transfer. Can PS-Theory provide a theoretical explanation for negative transfer?
>
> A: In the Appendix C.2, we give a manifold-based definition of negative transfer under the frozen GFM map, and a corollary that relates mitigation to larger intrinsic dimension, Hausdorff measure, and diameter of the adapted representation space compared to graph prompt tuning. This issue is also quite significant within the field, but it is not the focus of our work. PS-Theory can offer a theoretical perspective and technical approach to analyze this problem, but it will not be specifically discussed in this paper, it is presented as a future research direction.
>
> > Limitations: It is recommended that the authors add a section discussing the limitations of PS-Theory and MTG. This addition would improve the overall coherence of the paper.
>
> A: Thank you very much for this helpful suggestion. We will add a dedicated Limitations subsection in the revision to clearly state the current scope of PS-Theory and MTG (GNN-based GFMs under our unified layer formulation), and to outline concrete directions for extending the framework to broader GFM architectures, including LLM-related models.
>
> **Once again, we deeply appreciate your valuable comments and thoughtful efforts.** We sincerely hope our clarification has addressed your concerns and look forward to your response.
>
> [1] Qunzhong Wang, Xiangguo Sun, and Hong Cheng. Does graph prompt work? A data operation perspective with theoretical analysis. In ICML, 2025.

---

> > ### Author Rebuttal · Reviewer_xcpy · 2026-04-04
> >
> > Thank you for the rebuttal. I maintain my positive score.

---

> > > ### Author Response · Authors · 2026-04-04
> > >
> > > We are very pleased to receive your feedback! Once again, we would like to express our gratitude for the time and effort you have devoted to our manuscript.

---

### Official Review · Reviewer_pjmy · 2026-04-10

**Soundness:** 3
**Presentation:** 2
**Significance:** 4
**Originality:** 3
**Overall Recommendation:** 5
**Confidence:** 3

**Summary:**

To address the problem of quantifying the adaptation capacity of graph prompt tuning, the authors introduce a novel Prismatic Space Theory (PS-Theory), which models GFM layers as piecewise linear maps and uses geometric analysis to establish an upper bound of the adaptation capacity for graph prompt tuning. Based on this theoretical insight, they propose message tuning (MTG) and prove both theoretically and experimentally that it surpasses graph prompt methods.

**Compliance With Llm Reviewing Policy:**

Affirmed.

**Final Justification:**

See strengths and weaknesses.

**Key Questions For Authors:**

1. In Equation (1), the optimization objective uses the "optimal representation" of a graph. How to understand this optimal representation? Does it depend on the downstream task, or is it some fixed ideal representation?

2. Definition 3.5 models the input space as a compact smooth manifold. Why does treating discrete graph samples as samples from a continuous smooth manifold help characterize boundedness properties of the data?

**Limitations:**

It is suggested that the authors include a section discussing the limitations of the theory and method.

**Strengths And Weaknesses:**

Strengths

1. PS-Theory is novel, establishing a geometric framework to quantify the adaptation capacity of graph prompt tuning. The idea is clear and insightful. Compared with the existing theoretical work on graph prompt, this analytical perspective seems to be more reasonable and has greater potential for development.

2. MTG is a clean and effective design. By injecting learnable message prototypes into each layer, it adaptively guides message fusion without updating pre-trained weights. This design appears to be more expressive compared to graph prompt tuning.

3. The experimental evaluation is solid. Experimental results show that MTG performs well, which is consistent with the theoretical analysis. Ablation and sensitivity analyses also provide good evidence of the effectiveness of the method.

4. The writing details of the paper is good, with a clear overall structure. There are numerous indexes corresponding to the appendices in the main text, as well as some problem definitions and experimental questions serving as guidance.

Weaknesses

1. More detailed explanations and relevant theoretical citations are needed to make the geometric intuition of prismatic space more clear. The related work on the piecewise linear maps theory is placed in the appendix; this should be mentioned in the main text.

2. Readability of the theoretical part needs to be improved. The symbol definitions in Table 2 should be more detailed, and a table of contents for the appendix would be helpful. Although the Reading Guideline in Appendix B helps, the overall logical flow of the theory should be stated more clearly.

3. The layout of the experimental section is somewhat crowded due to the large number of figures; the organization of this part needs to be optimized. The description of the experimental setup should be more detailed.

4. The application scope of both the theory and the method is not clearly defined. The assumptions on which the theoretical analysis is based are scattered throughout Section 3.2; they should be stated in a unified manner.

---

### Decision · Program_Chairs · 2026-04-30

**Decision:**

Accept (regular)

**Comment:**

The main contribution of this paper lies in its introduction of Prismatic Space Theory (PS-Theory), a theoretical framework for measuring the adaptability of Graph Prompt Tuning, and the proposed MTG method derived from this analysis. MTG introduces small learnable message prototypes at each GNN layer to adaptively guide message aggregation. Extensive experiments further support the effectiveness of the proposed method.

The reviews for this submission are mixed. Accordingly, the AC invited an additional reviewer, pjmy. Among the four reviewers, three recommend acceptance, while one recommends rejection. The three supportive reviews are broadly aligned in recognizing the paper’s clear motivation, rigorous theoretical analysis, method novelty, and strong, convincing empirical results. The  reviewer JsU5 who recommends to reject, raised concerns primarily about the assumptions underlying the theory and the method novelty. After the rebuttal, JsU5 maintained reservations regarding the novelty of the method and kept the original score of 2 (reject).

Overall, I find that this paper offers a solid theoretical perspective on measuring the adaptability of Graph Prompt Tuning and, based on that perspective, proposes a lightweight method with practical value. While MTG shares certain similarities with some forms of prefix-tuning, the authors have explained the distinctions and their technical contributions in sufficient detail both in the paper and in the rebuttal. Notably, the reviewers generally agree on the originality and value of the theoretical contribution. On balance, I recommend weak accept.